# Overparameterization Improves Robustness to Covariate Shift in High Dimensions

**Nilesh Tripuraneni**[*]
U.C. Berkeley[†]
nilesh_tripuraneni@berkeley.edu

**Ben Adlam**[*]
Brain Team, Google Research
adlam@google.com

**Jeffrey Pennington**[*]
Brain Team, Google Research
jpennin@google.com

## Abstract

A significant obstacle in the development of robust machine learning models is *covariate shift*, a form of distribution shift that occurs when the input distributions of the training and test sets differ while the conditional label distributions remain the same. Despite the prevalence of covariate shift in real-world applications, a theoretical understanding in the context of modern machine learning has remained lacking. In this work, we examine the exact high-dimensional asymptotics of random feature regression under covariate shift and present a precise characterization of the limiting test error, bias, and variance in this setting. Our results motivate a natural partial order over covariate shifts that provides a sufficient condition for determining when the shift will harm (or even help) test performance. We find that overparameterized models exhibit enhanced robustness to covariate shift, providing one of the first theoretical explanations for this ubiquitous empirical phenomenon. Additionally, our analysis reveals an exact linear relationship between the in-distribution and out-of-distribution generalization performance, offering an explanation for this surprising recent observation.

## 1 Introduction

Theoretical justification for almost all machine learning methods relies upon the equality of the distributions from which the training and test data are drawn. Nevertheless, in many real-world applications, this equality is violated—naturally-occurring distribution shift between the training data and the data encountered during deployment is the rule, not the exception [31]. Even *non-adversarial* changes in distributions can uncover the surprising fragility of modern machine learning models [55, 56, 43, 26, 12, 50]. Such shifts are distinct from adversarial examples, which require explicit poisoning attacks [22]; rather, they can result from mild corruptions, ranging from changes of camera angle or blur [26], to subtle, unintended changes in data acquisition procedures [56]. Moreover, this fragility limits the application of deep learning in certain safety-critical areas [31].

Empirical studies of distribution shift have observed several intriguing phenomena, including linear trends between model performance on shifted and unshifted test distributions [56, 26, 31], dramatic degradation in calibration [50], and surprising spurious inductive biases [12]. Theoretical understanding of why such patterns occur across a variety of real-world domains is scant. Even basic questions such as what makes a certain distribution shift likely to hurt (or help) a model's performance,

---

[*]Equal contribution.
[†]Work performed while the author was at Google.

35th Conference on Neural Information Processing Systems (NeurIPS 2021).

and by how much, are not understood. One reason that these phenomena have eluded theoretical understanding is that there is often a strong coupling between model and distribution, implying that the effect of a given shift cannot usually be understood in a model-agnostic way. Another reason that satisfactory explanations have remained lacking is that the go-to formalism for studying generalization in classical models, namely uniform convergence theory (see e.g. [66]), may be insufficient to explain the behavior of modern deep learning methods (even in the absence of distribution shift) [47, 68]. Indeed, classical measures of model complexity, such as various norms of the parameters, have been found to lead to ambiguous conclusions [49].

In this paper, we follow a different approach: instead of focusing on worst-case bounds for generic distributions, we study average-case behavior for narrowly specified distributions. While this change in perspective sacrifices generality, it allows us to derive more precise predictions, which we believe are necessary to fully capture the relevant phenomenology. We study a specific type of distribution shift called covariate shift, in which the distributions of the training and test covariates differ, while the conditional distribution of the labels given the covariates remains fixed. Using random matrix theory, we perform an asymptotically exact computation of the generalization error of random feature regression under covariate shift. The random feature model provides a useful testbed to (1) investigate the interplay between various factors such as model complexity, label noise, bias, variance, and covariate shift; (2) rigorously define a *model-agnostic* notion for the strength of covariate shift; and (3) provide a theoretical explanation for the linear relationships recently observed between in-distribution and out-of-distribution generalization performance [55, 56, 27].

## 1.1 Contributions

Our primary contributions are to:

1. Provide a *model-agnostic* partial order over covariate shifts that is sufficient to determine when a shift will increase or decrease the test error in random feature regression (see Def. 4.1);

2. Compute the test error, bias, and variance of random feature regression for general multivariate Gaussian covariates under covariate shift in the high-dimensional limit (see Sec. 5.1);

3. Prove that overparameterization enhances robustness to covariate shift, and that the error, bias, and variance are nonincreasing functions of the number of excess parameters (see Sec. 5.3);

4. Deduce an exact linear relationship between in-distribution and out-of-distribution generalization performance, offering an explanation for this surprising recent empirical observation (see Sec. 5.4).

## 1.2 Related work

There is extensive literature on the empirical analysis of distribution shift in all of its myriad forms, ranging from domain adaptation [62, 20, 8, 70, 69, 37, 36] to defenses against adversarial attacks [39, 60] to distributionally robust optimization [59, 15, 16], among many others. Interestingly, for naturally occurring distribution shifts [31, 26], standard robustness interventions provide little protection [56, 63]. Indeed, empirical risk minimization on clean, unshifted training data often performs better on out-of-distribution benchmarks than more sophisticated methods [31]. One of the most striking observations in the context of natural distribution shifts is that model robustness improves with the classifier's accuracy [56, 63, 26, 43]. For example, if a classifier's accuracy increases by 1.0% on the unshifted CIFAR-10 test set, this tends to increase its accuracy by 1.7% on the CIFAR-10.1 dataset (a dataset with natural distribution shift) [56]. Moreover, such linear trends between the unshifted and shifted measures of error have now been observed in several contexts [56, 63, 43, 40, 44].

The number of theoretical works studying the impact of distribution shift on generalization is far smaller. One pioneering work provides VC-dimension-based error bounds for classification that are augmented by a discrepancy measure between source and target domains [10], while another demonstrates a similar class of uniform convergence-based results in the setting of kernel regression [11]. Recent work shows that learning domain-invariant features is insufficient to guarantee generalization when the class-conditional distributions of features may shift [69]. When the source domain gradually shifts toward the target domain, non-vacuous margin-based bounds for self-training can be established [32]. In [40], assumptions based on model similarity are used to help explain why classifiers exhibit linear trends between their accuracies on shifted and unshifted test sets [56, 43].

Our technical tools build on a series of works that have studied the exact high-dimensional limit of the test error for a growing class of model families and data distributions. In the context of linear models, recent work analyzes ridge regression for general covariances and a general non-isotropic source condition on the parameters which generate the targets [57], extending earlier work studying minimum-norm interpolated least squares and ridge regression in the random design setting [9, 14, 24]. The non-isotropy of source parameter effectively induces a shift on the bias term of this model, but the phenomenon is distinct from the covariate shifts we study here. Beyond linear regression, random feature models provide a rich but tractable class of models to gain further insight into generalization phenomena [2, 3, 41, 35]. These methods are of particular interest because of their connection to neural networks, with the number of random features corresponding to the network width (or model complexity) [48, 33, 29], and because they serve as a practical method for data analysis in their own right [54, 61]. In this context, a precise characterization of the gaps between uniform convergence and the (asymptotic) exact test error as a function of the sample size and number of random features can be derived [68]. Since this paper's publication, we released follow-up work considering unequal scales in the training and test distributions and optimal regularization [64]. From the technical perspective, our analytic techniques build upon these works and a series of recent results stemming from the literature on random matrix theory and free probability [53, 52, 1, 2, 38, 51, 19, 45].

## 2 Preliminaries

### 2.1 Problem setup and notation

As in prior work studying random feature regression [24, 41, 2, 1], we compute the test error in the high-dimensional, proportional asymptotics where the dataset size $m$, input feature dimension $n_0$, and hidden layer size $n_1$ all tend to infinity at the same rate, with $\phi := n_0/m$ and $\psi := n_0/n_1$ held fixed. We refer to $\phi/\psi$ as the *overparameterization ratio*, which is the limit of $n_1/m$ and characterizes the normalized complexity of the (random) feature model.

Interestingly, in this high-dimensional limit, the conditional distribution of a linear labeling function is asymptotically equivalent to a wide class of nonlinear teacher functions (see [41, 2] for more details). With this in mind, we consider the task of learning an unknown function from $m$ i.i.d. samples $(\mathbf{x}_i, y_i) \in \mathbb{R}^{n_0} \times \mathbb{R}$ for $i \in \{1, \ldots, m\}$, where the covariates are Gaussian, $\mathbf{x}_i \sim \mathcal{N}(0, \Sigma)$ with positive definite covariance matrix $\Sigma$, and the labels are generated by a linear function parameterized by $\beta \in \mathbb{R}^{n_0}$, drawn from $\mathcal{N}(0, I_{n_0})$. In particular

$$y(\mathbf{x}_i) = \beta^\top \mathbf{x}_i / \sqrt{n_0} + \epsilon_i, \tag{1}$$

where $\epsilon_i \sim \mathcal{N}(0, \sigma_\epsilon^2)$ is additive label noise on the training points.

We study the class of prediction models defined by kernel ridge regression using unstructured random feature maps [54]. The random features are given by a single-layer, fully-connected neural network with random weights. Given a set of training data $X = [\mathbf{x}_1, \ldots, \mathbf{x}_m]$ and a prospective test point $\mathbf{x}$, the random features embeddings of the training and test data are given by

$$F := \sigma(WX/\sqrt{n_0}) \quad \text{and} \quad f := \sigma(W\mathbf{x}/\sqrt{n_0}), \tag{2}$$

for a random weight matrix $W \in \mathbb{R}^{n_1 \times n_0}$ with i.i.d. standard Gaussian entries and an activation function $\sigma : \mathbb{R} \to \mathbb{R}$ applied elementwise. The induced kernel is

$$K(\mathbf{x}_1, \mathbf{x}_2) := \frac{1}{n_1} \sigma(W\mathbf{x}_1/\sqrt{n_0})^\top \sigma(W\mathbf{x}_2/\sqrt{n_0}), \tag{3}$$

and the model's predictions are given by $\hat{y}(\mathbf{x}) = YK^{-1}K_\mathbf{x}$, where $Y := [y(\mathbf{x}_1), \ldots, y(\mathbf{x}_m)]$, $K := K(X, X) + \gamma I_m$, $K_\mathbf{x} := K(X, \mathbf{x})$, and $\gamma \geq 0$ is a ridge regularization constant[1]. Owing to the implicit regularization effect of the nonlinear feature maps [7], in low noise settings the optimal value of $\gamma$ can sometimes be negative [30]. For simplicity, we nevertheless make the standard assumption that $\gamma \geq 0$, though we emphasize our techniques readily accommodate negative values.

---

[1]We overload the definition of $K$ to include the additive regularization whenever no arguments are present. Also, if $\gamma = 0$ and $K$ is not full-rank, $K^{-1}$ should be understood as the Moore–Penrose pseudoinverse.

Our central object of study is the expected test loss for a datapoint $\mathbf{x} \sim \mathcal{N}(0, \Sigma^*)$ where $\Sigma^*$ may be different from the training covariance $\Sigma$. The test error (without label noise on the test point) is

$$E_{\Sigma^*} = \mathbb{E}[(\beta^\top \mathbf{x}/\sqrt{n_0} - YK^{-1}K_\mathbf{x})^2] \tag{4}$$

$$= \underbrace{\mathbb{E}_{\mathbf{x},\beta}[(\mathbb{E}[\hat{y}(\mathbf{x})] - y(\mathbf{x}))^2]}_{B_{\Sigma^*}} + \underbrace{\mathbb{E}_{\mathbf{x},\beta}[\mathbb{V}[\hat{y}(\mathbf{x})]]}_{V_{\Sigma^*}}, \tag{5}$$

where the inner expectations defining the bias and variance are computed over $W$, $X$, and $Y$. We decompose the training and test covariance matrices into eigenbases as $\Sigma = \sum_{i=1}^{n_0} \lambda_i \mathbf{v}_i \mathbf{v}_i^\top$ and $\Sigma^* = \sum_{i=1}^{n_0} \lambda_i^* \mathbf{v}_i^* \mathbf{v}_i^{*\top}$, where the eigenvalues are in nondecreasing magnitude, i.e. $\lambda_1 \leq \lambda_2 \leq \ldots \leq \lambda_{n_0}$ and $\lambda_1^* \leq \lambda_2^* \leq \ldots \leq \lambda_{n_0}^*$. We define the *overlap coefficients*

$$r_i := \mathbf{v}_i^\top \Sigma^* \mathbf{v}_i = \sum_{j=1}^{n_0} (\mathbf{v}_j^* \cdot \mathbf{v}_i)^2 \lambda_j^* \tag{6}$$

to measure the alignment of $\Sigma^*$ with the $i$th eigendirection of $\Sigma$. In particular, $r_i$ is the induced norm of $\mathbf{v}_i$ with respect to $\Sigma^*$. We use $\bar{\mathrm{tr}}$ to denote the dimension-normalized trace: for a matrix $A \in \mathbb{R}^{n \times n}$, $\bar{\mathrm{tr}}(A) = \frac{1}{n}\mathrm{tr}(A)$. We use $\|A\|_\infty$ and $\|A\|_F$ to denote the operator norm and Frobenius norm of matrix $A$ respectively. Finally, we use $\delta_\mathbf{x}$ to denote the Dirac delta function centered at $\mathbf{x}$.

## 2.2 Assumptions

Regularity assumptions on the spectra of $\Sigma$ and $\Sigma^*$ are necessary to state the limiting behavior of this system. As in [67], it is not sufficient to consider the spectra of these matrices individually; they must be considered jointly. We do this in an eigenbasis of $\Sigma$.

**Assumption 1.** *We define the empirical joint spectral distribution (EJSD) as*

$$\mu_{n_0} := \frac{1}{n_0} \sum_{i=1}^{n_0} \delta_{(\lambda_i, r_i)} \tag{7}$$

*and assume it converges in distribution to some $\mu$, a distribution on $\mathbb{R}_+^2$ as $n_0 \to \infty$. We refer to $\mu$ as the limiting joint spectral distribution (LJSD), and emphasize that this defines the relevant limiting properties of the train and test distributions[2]. Additionally, we require that $\limsup_{n_0} \max(\|\Sigma\|_\infty, \|\Sigma^*\|_\infty) \leq C$ for a constant $C$.*

Often we use $(\lambda, r)$ for random variables sampled jointly from $\mu$ and denote the marginal of $\lambda$ under $\mu$ with $\mu_{\mathrm{train}}$. The conditional expectation $\mathbb{E}[r|\lambda]$ is an important object in our study. We frequently overload the notation $\mathbb{E}[r|\lambda]$ to view it as a function of $\lambda$, and we assume the following for simplicity.

**Assumption 2.** *$\mu$ is either absolutely continuous or a finite sum of delta masses. Moreover, the expectations of $\lambda$ and $r$ are finite.*

When the eigenspaces of $\Sigma$ and $\Sigma^*$ are aligned and $r_i = \lambda_i^* = \Phi(\lambda_i)$ for some smooth function $\Phi$, the support of the LJSD degenerates. Here, Assump. 1 is essentially equivalent to assuming the empirical spectral distribution of $\Sigma$ converges in distribution to some $\mu_{\mathrm{train}}$, which is a standard assumption in the regression literature [14, 41]. One special case of note is when there is no shift, i.e. $\Phi$ is the identity, in which case the LJSD degenerates to $\mu_\emptyset$ defined by

$$\mu_\emptyset(\lambda, r) := \mu_{\mathrm{train}}(\lambda)\delta_\lambda(r), \quad i.e. \quad (\lambda, \lambda) \sim \mu_\emptyset \text{ for } \lambda \sim \mu_{\mathrm{train}}. \tag{8}$$

As our analysis will eventually take place in the high-dimensional limit, we further define the asymptotic scales of the training and test covariances as $s := \lim_{n_0 \to \infty} \bar{\mathrm{tr}}(\Sigma) = \mathbb{E}_\mu[\lambda]$ and $s_* := \lim_{n_0 \to \infty} \bar{\mathrm{tr}}(\Sigma^*) = \mathbb{E}_\mu[r]$ under the limiting behavior specified in Assump. 1.

Throughout this paper, we also enforce the following standard regularity assumptions on the activation functions to ensure the existence of the moments and derivatives we compute.

**Assumption 3.** *The activation function $\sigma : \mathbb{R} \to \mathbb{R}$ is assumed to be differentiable almost everywhere. We assume that, $|\sigma(x)|, |\sigma'(x)| \leq c_0 \exp(c_1 x)$ for constants $c_0, c_1$.*

---

[2]Note that the EJSD depends not only on $\Sigma$ and $\Sigma^*$ but also on a choice of eigendecomposition for $\Sigma$ when it has repeated eigenvalues. However, all possible choices for the EJSD form an equivalence class, and the ambiguity does not affect later definitions and conclusions. See Sec. A3.

## 2.3 A simple family of diatomic distributions

As the above assumptions allow such a general class of covariance structures, it is useful to consider our results in the context of a simple family of distributions that readily admits a simple interpretation.

**Definition 2.1.** *For $\alpha \geq 1$ and $\theta \in \mathbb{R}$, we define the family of $(\alpha, \theta)$-diatomic LJSDs with $\theta$-power-law shifts as*

$$\mu_{\alpha,\theta}^{diatomic} := \frac{1}{\alpha+1}\delta_{(\alpha, C\alpha^\theta)} + \frac{\alpha}{\alpha+1}\delta_{(\alpha^{-1}, C\alpha^{-\theta})}, \tag{9}$$

*where $C$ is a normalization constant chosen so that $\mathbb{E}_{\mu_{\alpha,\theta}^{diatomic}}[r] = 1$. Note that $\mu_{\alpha,\theta}^{diatomic}$ is the limit of*

$$\Sigma_{ij} := \begin{cases} \alpha & \text{if } i = j \text{ and } i \leq \lfloor \frac{n_0}{1+\alpha} \rfloor \\ \alpha^{-1} & \text{if } i = j \text{ and } i > \lfloor \frac{n_0}{1+\alpha} \rfloor \\ 0 & \text{if } i \neq j \end{cases} \quad \text{and} \quad \Sigma^* := \frac{1}{\bar{\mathrm{tr}}(\Sigma^\theta)}\Sigma^\theta. \tag{10}$$

This simple two-parameter family of distributions captures the fast eigenvalue decay observed in many datasets in machine learning, for which the covariance spectra are often dominated by several large eigenvalues and exhibit a long tail of many small eigenvalues [34]. Note that the trivial case of $\alpha = 1$ yields an identity covariance with no shift. For the nontrivial setting $\alpha > 1$, the exponent $\theta$ parameterizes the strength of the shift in an intuitive way: when $\theta = 1$, there is no shift; when $\theta < 1$, $\alpha^\theta < \alpha$, so the large eigendirections of the training distribution are suppressed in the test distribution, suggesting that the shift makes learning harder; when $\theta > 1$, $\alpha^\theta > \alpha$, so the large eigendirections of the training distribution are further emphasized in the test distribution, suggesting that the shift makes learning easier. We will return to the notion of shift strength in Secs. 3 and 4.

# 3 Motivating example: linear regression

We first consider the relatively simple case of ridgeless linear regression (LR), which will help build some intuition for the more general analysis of random feature regression in Sec. 5.1. Assuming the labels are generated by the linear model defined above, i.e. $y_i = \beta^\top \mathbf{x}_i / \sqrt{n_0} + \varepsilon_i$, the estimator is given by $\hat{\beta} = (XX^\top)^{-1}XY$, and the test risk (see Eq. (4)) has the following simple form.

**Proposition 3.1.** *For fixed dimension $n_0$ and sample size $m > n_0 + 1$, the test error of LR is given by*

$$E_{\Sigma^*}^{LR} = \sigma_\epsilon^2 \frac{n_0}{m - n_0 - 1}\bar{\mathrm{tr}}(\Sigma^*\Sigma^{-1}) = \sigma_\epsilon^2 \frac{n_0}{m - n_0 - 1}\frac{1}{n_0}\sum_{i=1}^{n_0}\frac{r_i}{\lambda_i}. \tag{11}$$

*Under Assump. 1, as $n_0, m \to \infty$ with $\phi = n_0/m$ fixed, $E_{\Sigma^*}^{LR} \to E_\mu^{LR} = \sigma_\epsilon^2 \phi/(1-\phi)\mathbb{E}_\mu[r/\lambda]$.*

One immediate question is whether a given shift will increase or decrease the test error relative to $E_\Sigma^{\mathrm{LR}}$. While the precise answer is of course determined by the value of $\bar{\mathrm{tr}}(\Sigma^*\Sigma^{-1})$, it useful for the subsequent analysis to develop an understanding of the individual contributions to this term. Similar decompositions of the test error into eigenspaces have proved useful in a variety of other contexts, e.g. [42, 4]. We begin with a specific example in the setting of the finite-dimensional analog of Def. 2.1.

**Example 3.1.** *For the finite form of the $(\alpha, \theta)$-diatomic density defined in Eq. (10), the test error of LR is given by*

$$E_{\Sigma^*}^{LR} = \sigma_\epsilon^2 \frac{n_0}{m - n_0 - 1}\frac{\alpha + w(\alpha^{2\theta-1} - \alpha)}{1 + w(\alpha^{2\theta} - 1)} \quad \text{for} \quad w = \frac{1}{n_0}\left\lfloor \frac{n_0}{1+\alpha} \right\rfloor, \tag{12}$$

*and so $\frac{\partial}{\partial\theta}E_{\Sigma^*}^{LR} = \sigma_\epsilon^2 \frac{n_0}{m-n_0-1}\frac{2(1-w)w\alpha^{2\theta-1}(1-\alpha^2)\log(\alpha)}{(1+w(\alpha^{2\theta}-1))^2} \leq 0$, which implies $E_{\Sigma_1^*}^{LR} \leq E_{\Sigma_2^*}^{LR}$ whenever $\theta_1 \geq \theta_2$, in accordance with the discussion in Sec. 2.3. It follows from Eq. (10) that the condition $\theta_1 \geq \theta_2$ not only implies $\bar{\mathrm{tr}}(\Sigma_1^*\Sigma^{-1}) \leq \bar{\mathrm{tr}}(\Sigma_2^*\Sigma^{-1})$, but also that the ratios of overlap coefficients $r_{i,1}/r_{i,2}$ form a nondecreasing sequence. It is this condition involving all the eigendirections that will generalize to the nonlinear random feature setting in Sec. 4.*

The following proposition captures the essence of these considerations in the context of linear regression. See Sec. A4 for the proof.

**Proposition 3.2.** *Let $r_{i,1}$ and $r_{i,2}$ denote the overlap coefficients[3] of $\Sigma_1^*$ and $\Sigma_2^*$ relative to $\Sigma$. If $\mathrm{tr}(\Sigma_2^*) \geq \mathrm{tr}(\Sigma_1^*)$ and the ratios $r_{i,1}/r_{i,2}$ form a nondecreasing sequence, then in the setting of Prop. 3.1, $E_{\Sigma_2^*}^{LR} \geq E_{\Sigma_1^*}^{LR}$.*

Whereas the $(\alpha, \theta)$-diatomic LJSDs explicitly enforce the trace normalizations $\mathrm{tr}(\Sigma) = \mathrm{tr}(\Sigma_1^*) = \mathrm{tr}(\Sigma_2^*) = 1$, Prop. 3.2 provides sufficient conditions for the ordering of test errors for non-unit traces. The fact that $E_{\Sigma^*}^{\mathsf{LR}}$ scales linearly with the overall scale of $\Sigma^*$ is a unique feature of linear regression and does not generalize to the nonlinear random feature setting. We return to this issue in Sec. 4.

## 4 Definition of shift strength

Deriving conditions on whether a shift will hurt or help a model's performance is crucial to building an understanding of covariate shift. Motivated in part by the above results for linear regression, and in part by the results for random feature regression that we present in Sec. 5.3, we introduce the following definition of shift strength, which is a direct generalization of the conditions of Prop. 3.2:

**Definition 4.1.** *Let $\mu_1$ and $\mu_2$ be LJSDs with the same marginal distribution of $\lambda$, denoted $\mu_{train}$. If the asymptotic overlap coefficients are such that $\mathbb{E}_{\mu_1}[r|\lambda]/\mathbb{E}_{\mu_2}[r|\lambda]$ is nondecreasing as a function of $\lambda$ on the support of $\mu_{train}$ and $\mathbb{E}_{\mu_1}[r] \leq \mathbb{E}_{\mu_2}[r]$, we say $\mu_1$ is easier than $\mu_2$ (or $\mu_2$ is harder than $\mu_1$), and write $\mu_1 \leq \mu_2$. Comparing against the case of no shift $\mu_\emptyset$, we say $\mu_1$ is easy when $\mu_1 \leq \mu_\emptyset$ and hard when $\mu_1 \geq \mu_\emptyset$.*

A priori, there is little reason to hope that such a *model-independent* definition of shift strength would adequately characterize a shift's impact on the total error, bias, or variance of a given model. Even for the relatively simple case of random feature kernel regression, the nonlinear feature maps of Eq. (2) would seem to inextricably couple the covariance distribution to the model.

Nevertheless, as we show in Sec. 5.1, the coupling between model and shift simplifies considerably in the high-dimensional proportional asymptotics. It is characterized by a handful of constants that depend solely on the overall covariance scale $\mathbb{E}_\mu[r]$ and a collection of functionals of $\mu$, whose magnitudes can be bounded in terms of the ratio $\mathbb{E}_\mu[r|\lambda]/\mathbb{E}_{\mu_\emptyset}[r|\lambda]$. The conditions that Def. 4.1 places on $\mathbb{E}_\mu[r]$ and $\mathbb{E}_\mu[r|\lambda]/\mathbb{E}_{\mu_\emptyset}[r|\lambda]$ can be augmented by various constraints on the model to derive bounds on how the total error and bias will respond to a shift of a given strength. This perspective introduces considerable complexity and we present the details of this analysis elsewhere.

In this work, we focus on a simpler, surprising result: by merely normalizing the scales of the covariate distributions (i.e. enforcing $s = s_*$), Def. 4.1 provides a *model-independent* definition of shift strength that determines how random feature models respond to shifts of different strength. This observation motivates the following assumption.

**Assumption 4.** *The training and test covariance scales are equal, $\mathbb{E}_\mu[\lambda] = \mathbb{E}_\mu[r]$, i.e. $s = s_*$.*

We emphasize that Assump. 4 reflects common practice for many models and data modalities, as preprocessing techniques such as standardization are ubiquitous and many architectural components such as layer- or batch-normalization achieve a similar effect [21, 28, 46].

## 5 Covariate shift in random feature kernel regression

### 5.1 Main results

Our main results characterize the high-dimensional limits of the test error, bias, and variance of the nonlinear random feature model of Sec. 2. Before stating them, we first introduce some additional constants that capture the effect of the nonlinearity $\sigma$. For $z \sim \mathcal{N}(0, s)$, define

$$\eta := \mathbb{V}[\sigma(z)], \quad \rho := (\tfrac{1}{s}\mathbb{E}[z\sigma(z)])^2, \quad \zeta := s\rho, \quad \text{and} \quad \omega := s(\eta/\zeta - 1). \tag{13}$$

Our results also depend on the covariance spectra through two sets of functionals of $\mu$,

$$\mathcal{I}_{a,b}(x) := \phi \, \mathbb{E}_\mu \left( \lambda^a \left( \phi + x\lambda \right)^{-b} \right) \quad \text{and} \quad \mathcal{I}_{a,b}^*(x) := \phi \, \mathbb{E}_\mu \left( r\lambda^{a-1} \left( \phi + x\lambda \right)^{-b} \right). \tag{14}$$

---

[3]Recall the definition of the overlap coefficients in Eq. (6).

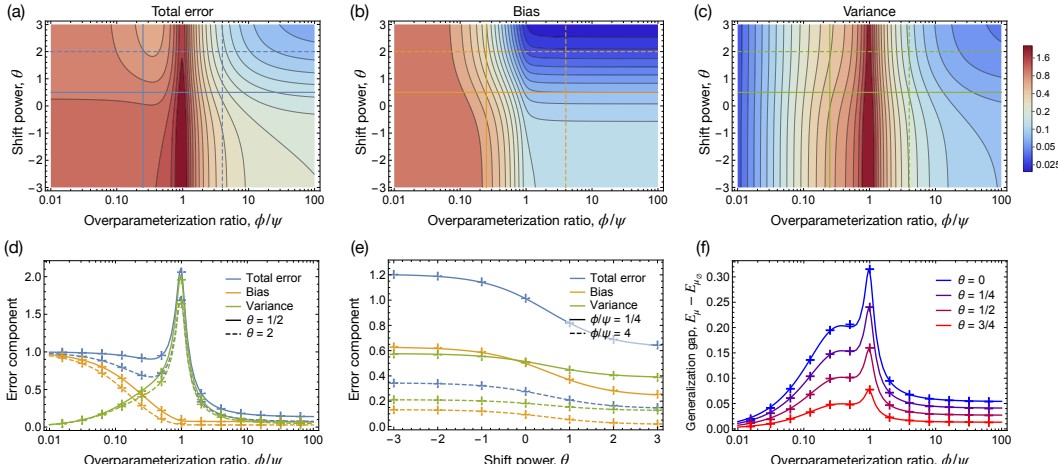

Figure 1: The asymptotic predictions of Thm. 5.1 as a function of the overparameterization ratio $(\phi/\psi = n_1/m)$ and the shift power $(\theta)$ for the $(2,\theta)$-diatomic LJSD (Eq. (9)) with $\phi = n_0/m = 0.5$, $\sigma = \text{ReLU}$, $\gamma = 0.001$, and $\sigma_\varepsilon^2 = 0.1$. (a) The test error exhibits the characteristic double descent behavior for all shift powers. (b) The bias is a nonincreasing function of $\phi/\psi$ for all shift powers, as in Prop. 5.2. (c) The variance is the source of the double-descent peak, and is a nonincreasing function of $\phi/\psi$ for all shift powers in the overparameterized regime, as in Prop. 5.3. In (a,b), the total error and bias are nonincreasing functions of $\theta$, as in Prop. 5.1. (d) 1D horizontal slices of (a,b,c) demonstrate the monotonicity in $\phi/\psi$ predicted by Props. 5.2 and 5.3. (e) 1D vertical slices of (a,b,c) demonstrate the monotonicity in $\theta$ predicted by Prop. 5.1 (the variance also appears monotonic, but it need not be in general). (f) The generalization gap between the error on shifted and unshifted distributions is a nonincreasing function of $\phi/\psi$ in the overparameterized regime, as in Prop. 5.4. Markers in (d,e,f) show simulations for $n_0 = 512$ and agree well with the asymptotic predictions.

**Theorem 5.1.** *Under Assumps. 1, 2, 3 and 4, as $n_0, n_1, m \to \infty$ the test error $E_{\Sigma^*}$ converges to $E_\mu = B_\mu + V_\mu$, with the bias $B_\mu$ and variance $V_\mu$ given by*

$$B_\mu = \phi \mathcal{I}_{1,2}^* \tag{15}$$

$$V_\mu = -\rho \frac{\psi}{\phi} \frac{\partial x}{\partial \gamma} \Big( \mathcal{I}_{1,1}(\omega + \phi \mathcal{I}_{1,2})(\omega + \mathcal{I}_{1,1}^*) + \frac{\phi^2}{\psi} \gamma \bar\tau \mathcal{I}_{1,2} \mathcal{I}_{2,2}^*$$

$$+ \gamma \tau \mathcal{I}_{2,2}(\omega + \phi \mathcal{I}_{1,2}^*) + \sigma_\varepsilon^2 \Big( (\omega + \phi \mathcal{I}_{1,2})(\omega + \mathcal{I}_{1,1}^*) + \frac{\phi}{\psi} \gamma \bar\tau \mathcal{I}_{2,2}^* \Big) \Big), \tag{16}$$

*where $x$ is the unique nonnegative real root of $x = \frac{1-\gamma\tau}{\omega + \mathcal{I}_{1,1}}$, $\frac{\partial x}{\partial \gamma} = -\frac{x}{\gamma + \rho\gamma(\tau\psi/\phi + \bar\tau)(\omega + \phi \mathcal{I}_{1,2})}$, and*

$$\tau = \frac{\sqrt{(\psi-\phi)^2 + 4x\psi\phi\gamma/\rho} + \psi - \phi}{2\psi\gamma} \quad and \quad \bar\tau = \frac{1}{\gamma} + \frac{\psi}{\phi}\Big(\tau - \frac{1}{\gamma}\Big). \tag{17}$$

Numerical predictions from Thm. 5.1 can be obtained by first solving the self-consistent equation for $x$ by fixed-point iteration, $x \mapsto \frac{1-\gamma\tau}{\omega + \mathcal{I}_{1,1}}$, and then plugging the result into the remaining terms. Fig. 1 shows excellent agreement between these asymptotic predictions and finite-size simulations.

At times we will find it convenient to consider the ridgeless limit of Thm. 5.1. By carefully expanding $x$ and $\tau$ for small $\gamma$, it is straightforward to obtain the following corollary.

**Corollary 5.1.** *In the setting of Thm. 5.1, as the ridge regularization constant $\gamma \to 0$, $E_\mu = B_\mu + V_\mu$ with $B_\mu$ given in Eq. (15) and $V_\mu$ given by*

$$V_\mu = \frac{\psi}{|\phi - \psi|} x(\sigma_\varepsilon^2 + \mathcal{I}_{1,1})(\omega + \mathcal{I}_{1,1}^*) + \begin{cases} x\Big(1 - \frac{x(\omega - \sigma_\varepsilon^2)}{1 - x^2 \mathcal{I}_{2,2}}\Big)\mathcal{I}_{2,2}^* & \phi \geq \psi \\ \frac{x^2 \psi \mathcal{I}_{2,2}}{\phi - x^2 \psi \mathcal{I}_{2,2}}\big(\omega + \phi \mathcal{I}_{1,2}^*\big) & \phi < \psi \end{cases}, \tag{18}$$

*where $x$ is the unique positive real root of $x = \frac{\min(1, \phi/\psi)}{\omega + \mathcal{I}_{1,1}}$.*

Taking $\sigma(x) = x$ and $\psi \to 0$ in Cor. 5.1 yields an expression for the test error of ridgeless linear regression that agrees with [24] and with the asymptotic form of Prop. 3.1 (see Sec. A4.2).

## 5.2 Harder shifts increase the bias and test error

The bias and variance in Thm. 5.1 depend on the covariate shift exclusively through $\mathcal{I}_{a,b}^*$ — all other terms such as $x$, $\tau$, $\bar{\tau}$, and $\mathcal{I}_{a,b}$ only depend on the marginal of $\lambda$ under $\mu$. The functionals $\mathcal{I}_{a,b}^*$ generalize the simple ratio of overlap coefficients to eigenvalues, $\mathbb{E}_\mu[r/\lambda]$, that characterizes the error for linear regression (indeed, $\mathcal{I}_{0,0}^* = \phi\mathbb{E}_\mu[r/\lambda]$). In contrast, the error in the random feature setting is a combination of multiple such terms. Nevertheless, Def. 4.1 enables comparisons of the individual $\mathcal{I}_{a,b}^*$ functionals, which provide sufficient conditions to order the error and bias.

**Proposition 5.1.** *Consider two LJSDs such that $\mu_1 \leq \mu_2$ (see Def. 4.1). Then, in the setting of Thm. 5.1, $B_{\mu_1} \leq B_{\mu_2}$ and, if $\sigma_\varepsilon^2 \leq \omega$, $E_{\mu_1} \leq E_{\mu_2}$.*

Prop. 5.1 shows that Def. 4.1 provides an essentially model-independent condition to determine the impact of covariate shift on the test error[4]. Interestingly, both the bias (which arises from both regularization and model misspecification) and the total error (which has additional variance contributions from the randomness induced by $W$, $X$, and $\epsilon$) respond to shifts in tandem in the regime of small label noise. (For large label noise, the variance can dominate the error and cause violations of monotonicity; see Sec. 5.5.) Prop. 5.1 is illustrated for the $(\alpha, \theta)$-diatomic LJSD in Fig. 1: following the vertical lines upward in (a), (b), or the x-axis rightward in (e) yields easier shifts and a corresponding decrease in the bias and total error.

## 5.3 The benefit of overparameterization

While Prop. 5.1 shows that harder shifts increase the error, it is natural to wonder whether this increase can be mitigated by judicious model selection. In practice, empirical investigations have shown that the performance of large, overparameterized models tends to deteriorate less under distribution shift than their smaller counterparts [26]. We obtain a number of theoretical results that formally prove the benefit of overparameterization in our random feature setting.

First, we show that the bias decreases (or stays constant) when additional random features are added, which increases the model capacity and accords with the intuition of the bias as a measure of the model's ability to fit the data.

**Proposition 5.2.** *In the setting of Thm. 5.1, the bias $B_\mu$ is a nonincreasing function of the overparameterization ratio $\phi/\psi$.*

In contrast to the bias, which is monotonic for all overparameterization ratios, the variance can exhibit nonmonotonic behavior in the underparameterized regime. On the other hand, the following proposition shows that in the overparameterized regime, the variance is also nonincreasing. Note that our proof requires the setting of ridgeless regression ($\gamma = 0$), but numerical investigation suggests this condition may not be necessary (see Fig. 1).

**Proposition 5.3.** *In the setting of Cor. 5.1 and in the overparameterized regime (i.e. $\psi < \phi$), the variance $V_\mu$ is a nonincreasing function of the overparameterization ratio $\phi/\psi$.*

The explosion of variance at the interpolation threshold and then its subsequent decay have been demonstrated in previous exact asymptotic studies of random feature regression in the absence of covariate shift, in stark contrast to what classical theory would suggest [2, 41]. Prop. 5.3 confirms the existence of analogous behavior under covariate shift.

Taken together, Props. 5.2 and 5.3 imply that some of the benefits of overparameterization extend to models evaluated out-of-distribution. An additional benefit is that overparameterized models are more robust: the difference in error between unshifted and shifted test distributions is smaller for larger models. A formal statement of this enhanced robustness is given in the following result.

**Proposition 5.4.** *Consider two LJSDs such that $\mu_1 \leq \mu_2$ (see Def. 4.1). Then, in the setting of Cor. 5.1 and in the overparameterized regime (i.e. $\psi < \phi$), the generalization gap $E_{\mu_2} - E_{\mu_1}$ is a nonincreasing function of the overparameterization ratio $\phi/\psi$.*

---

[4]The weak model-dependence arising from the condition $\sigma_\varepsilon^2 < \omega$ can instead be regarded as a small label-noise condition that can always be satisfied for $\omega > 0$, which is true for any nonlinear $\sigma$ (see Sec. A1).

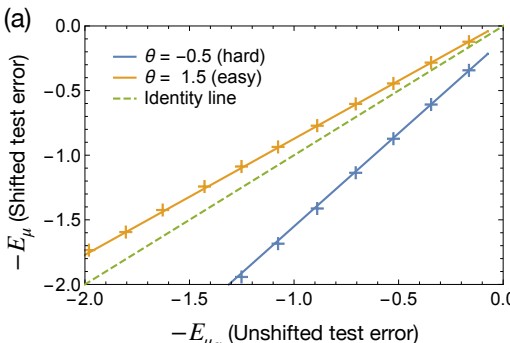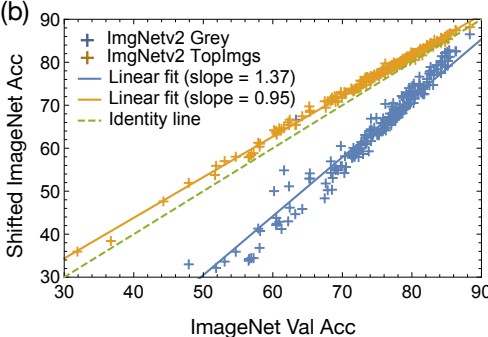

Figure 2: Linear relationship between in-distribution and out-of-distribution generalization error. **(a)** Asymptotic predictions for shifted versus unshifted error for models with varying degrees of overparameterization $\phi/\psi > 1$, obtained via Cor. 5.1 for the $(3, \theta)$-diatomic LJSD (Eq. (9)) with $\phi = n_0/m = 0.5$, $\sigma = $ ReLU, $\sigma_\varepsilon^2 = 0.01$ and two different values of the shift-power $\theta$. Markers represent simulations for $n_0 = 512$. The negated errors are plotted so that performance improves left to right and bottom to top, in order to match the behavior of the accuracy metric. **(b)** Reproduction of the empirical results of [56, 63], showing the relationship between the classification accuracy of various models on the original ImageNet test set and two shifted ImageNet datasets: a "hard" dataset with greyscale corruptions, Grey, and an "easy" dataset with high inter-annotator agreement, TopImgs. In both **(a)** and **(b)**, the slope is greater than one for the hard shift and less than one for the easy shift, in accordance with Prop. 5.5.

In Fig. 1, Props. 5.2, 5.3 and 5.4 are illustrated. Following the horizontal lines rightward in (a), (b), and (c) or the x-axis rightward in (d) and (f) leads to models with more parameters. The monotonicity of the bias across the whole range of parameterization is evident, as is the necessity of considering the monotonicity of the variance and generalization gap only when $\phi > \psi$.

## 5.4 Linear trends between in-distribution and out-of-distribution generalization

We have discussed how overparameterization yields improvements on both unshifted and shifted test distributions, which hints that these two quantities are positively correlated. Indeed, recent work has suggested increasing model size as a path to increased robustness [26]. Additional empirical studies have further refined this observation by discovering a linear relationship between the performance of models of varying complexity on unshifted and shifted data [56, 63]. In the context of ridgeless random feature regression, we provide a formal proof of this linear relationship.

**Proposition 5.5.** *In the setting of Cor. 5.1 and in the overparameterized regime (i.e. $\psi < \phi$),*

$$E_\mu = E_0 + \underbrace{\left(\frac{\omega + \mathcal{I}_{1,1}^*}{\omega + \mathcal{I}_{1,1}}\right)}_{\text{SLOPE}} E_{\mu_\emptyset} , \tag{19}$$

*parametrically in the overparameterization ratio $\phi/\psi$, where $E_0$ and SLOPE are constants independent of $\phi/\psi$, and $E_{\mu_\emptyset}$ is the error on the unshifted distribution. Moreover, SLOPE $\geq 1$ when $\mu$ is hard and SLOPE $\leq 1$ when $\mu$ is easy.*

Eq. (19) implies a parametrically linear relationship between $E_\mu$ and $E_{\mu_\emptyset}$ by varying $\phi/\psi$. Prop. 5.5 also makes the nontrivial prediction that an improvement on the unshifted distribution leads to a relatively greater improvement on the shifted distribution when the shift is hard, and to a relatively smaller improvement when the shift is easy. This prediction is corroborated qualitatively in the data from [55, 56, 43]. We plot this linear behavior in Fig. 2, where (a) shows the random feature model and (b) shows an example of data from [56, 63]. The striking similarity in these plots is evident.

## 5.5 Importance of assumptions

Props. 5.1, 5.2, 5.3, 5.4 and 5.5 rely on a number of assumptions and conditions; here we show the necessity of some of these prerequisites for our results.

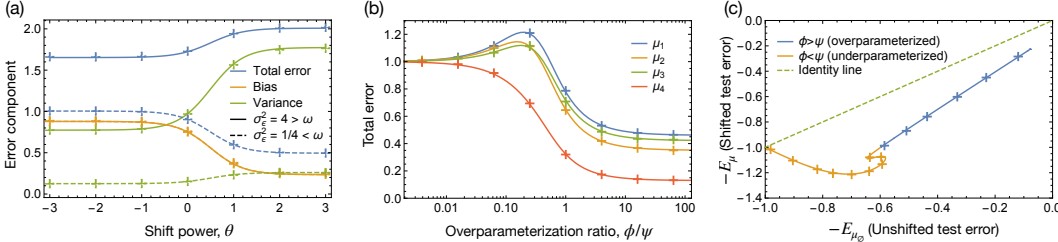

Figure 3: Relaxing the assumptions and conditions can lead to counterexamples to the propositions. (**a**) Asymptotic predictions (solid lines) and simulations with $n_0 = 4096$ (markers) for the total error, bias, and variance for the $(4, \theta)$-diatomic LJSD with $\phi = 4$, $\psi = 0.25$, $\gamma = 10^{-4}$, and $\sigma = \text{ReLU}$ (implying $\omega = 1 - \frac{2}{\pi} \approx 0.36$) as a function of increasing shift power $\theta$. When $\sigma_\varepsilon^2 < \omega$ (dashed curves), the bias and total error are nonincreasing, as predicted by Prop. 5.1, though the variance is not. When $\sigma_\varepsilon^2 > \omega$ (solid curves), the total error is no longer nonincreasing. (**b**) Asymptotic predictions (solid lines) and simulations with $n_0 = 256$ (markers) for the total error with $\phi = 0.5$, $\gamma = 0.1$, $\sigma = \text{ReLU}$ and $\sigma_\varepsilon^2 = 0.01$ as a function of the overparameterization ratio $\phi/\psi$ for four different LJSDs $\mu_1, \ldots, \mu_4$, chosen such that the only comparable pairs of LJSDs under the partial order in Def. 4.1 are $\mu_1 \geq \mu_4$ and $\mu_2 \geq \mu_4$, and the strict ordering of the error for those pairs is seen for all values of $\phi/\psi$. The orange ($\mu_2$) and green ($\mu_3$) curves cross one another, illustrating how nonmonotonicity of overlap ratios in Def. 4.1 can induce model-dependence in the ordering of the error. (**c**) Asymptotic predictions (solid lines) and simulations for $n_0 = 512$ (markers) for shifted versus unshifted error for models with varying values of the overparameterization ratio $\phi/\psi$, obtained via Thm. 5.1 for the $(3, -1/2)$-diatomic LJSD with $\phi = 0.5$, $\sigma = \text{ReLU}$, $\sigma_\varepsilon^2 = 0.01$, and $\gamma = 0.005$. While the relationship is nearly linear in the overparameterized regime, it is markedly nonlinear in the underparameterized regime, highlighting the importance of overparameterization in Prop. 5.5.

Prop. 5.1 relies on a small label-noise condition ($\sigma_\varepsilon^2 \leq \omega$) to ensure that the total error is ordered with respect to shift strength. The reason this condition is necessary is that the variance can actually increase as shifts become easier. While Fig. 1 presented a configuration for which the bias, variance, and error all decrease for easier shifts, Fig. 3(a) shows that a decrease is not guaranteed for the variance, and that it can increase even under the small label-noise condition. Moreover, while the bias continues to decrease for large label noise ($\sigma_\epsilon^2 > \omega$), the variance can become so large that bias can no longer offset it, causing the error itself to increase, as seen in Fig. 3(a).

The strict ordering of overlap coefficients in Def. 4.1 are also necessary to guarantee a complete decoupling of the model and the shift strength. In the absence of these conditions, Fig. 3(b) shows how even a single out-of-order overlap coefficient induces a violation of the monotonicity with respect to shift strength suggested by Prop. 5.1 (this example is detailed further in Sec. A8).

Finally, we note that the exact linear relationship between in-distribution and out-of-distribution generalization characterized by Prop. 5.5 in Eq. (19) relies crucially on the overparameterization condition, $\psi < \phi$, as evidenced in Fig. 3(c), which shows marked nonlinearity in the underparameterized regime. This observation is perhaps unsurprising, as severely underparameterized models tend towards chance predictions, which produce comparable errors on shifted and unshifted data. Indeed, similar nonlinear behavior is seen in the low-accuracy regime for realistic models [56, Figure 17].

## 6   Conclusion

We have presented an exact, asymptotic calculation of the test error, bias, and variance for random feature kernel regression in the presence of covariate shift. After defining a partial order over covariate shifts (motivated by the setting of linear regression), we have proved that harder shifts imply increased error. Our results capture many empirical phenomena such as the fact that overparameterization is beneficial even under covariate shift and that a linear relationship exists between the generalization error on shifted and unshifted data. Future directions include extending our results to the non-asymptotic regime, accommodating feature learning and more general neural network models, and investigating the impact of covariate shift for other loss functions.

## Acknowledgments and Disclosure of Funding

The authors would like to thank Rodolphe Jenatton, Horia Mania, Ludwig Schmidt, D. Sculley, Vaishaal Shankar, Lechao Xiao, and Steve Yadlowsky for valuable discussions.

This work was performed at and funded by Google. No third party funding was used.

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
