## Table of Contents: Appendix

## A1 Useful inequalities

Here we include the statements and proofs of several auxiliary inequalities that we use throughout the Appendix.

### A1.1 Basic properties of the self-consistent equation for $x$

We begin by establishing several basic inequalities. The definitions of the following quantities can be found in Thm. 5.1.

**Lemma A1.1.** *We have the following bounds:* $\omega, \tau, \bar{\tau}, x, \mathcal{I}_{a,b} \geq 0$ *and* $\frac{\partial x}{\partial \gamma} \leq 0$.

*Proof.* As shown in [53] for the unit-variance case, a simple Hermite expansion argument establishes the relation $\eta \geq \zeta$, which implies $\omega = s(\eta/\zeta - 1) \geq 0$. From Eqs. (A343) and (A344), $\tau$ and $\bar{\tau}$ are traces of positive semi-definite matrices and are therefore nonnegative. From the same equations, it follows that $x = \gamma \rho \tau \bar{\tau} \geq 0$. Nonnegativity of $x$ implies $\mathcal{I}_{a,b} \geq 0$ and $\mathcal{I}_{a,b}^* \geq 0$ from their definitions in Eq. (14). Finally, using the nonnegativity of $\omega$, $\tau$, $\bar{\tau}$, $x$, and $\mathcal{I}_{a,b}$, the expression for $\frac{\partial x}{\partial \gamma}$ in Thm. 5.1 immediately gives,

$$\frac{\partial x}{\partial \gamma} = -\frac{x}{\gamma + \rho\gamma(\frac{\psi}{\phi}\tau + \bar{\tau})(\omega + \phi\mathcal{I}_{1,2})} \leq 0. \tag{A1}$$

$\square$

Next we show that the self-consistent equation $x = \frac{1-\gamma\tau}{\omega + \mathcal{I}_{1,1}}$ appearing in Thm. 5.1 and defined in Eq. (A345) admits a unique positive real solution for $x$.

**Lemma A1.2.** *There is a unique real* $x \geq 0$ *satisfying* $x = \frac{1-\gamma\tau}{\omega + \mathcal{I}_{1,1}}$.

*Proof.* Let $t = 1/x \geq 0$ and define

$$h(t) = t\left(\frac{\rho(\psi - \phi) + \sqrt{\rho^2(\psi-\phi)^2 + 4\gamma\rho\phi\psi/t}}{2\rho\psi} - 1\right) + \omega + \mathcal{I}_{1,1}(1/t), \tag{A2}$$

which is a rewriting of Eq. (A345). It suffices to show that $h$ admits a unique real positive root. To that end, first observe that $\lim_{t \to 0} \mathcal{I}_{1,1}(1/t) = 0$ and $\lim_{t \to \infty} \mathcal{I}_{1,1}(1/t) = s$, so that

$$h(0) = \omega > 0 \quad \text{and} \quad \lim_{t \to \infty} h(t)/t = -\min\{1, \phi/\psi\} < 0, \tag{A3}$$

which together imply that $h$ has an odd number of positive real roots. Next, we show that $h$ is concave for $t \geq 0$:

$$h''(t) = -\frac{2\phi}{t^3}\left(\frac{\gamma^2\rho\phi\psi}{(\rho^2(\psi-\phi)^2 + 4\gamma\rho\phi\psi/t)^{3/2}} + \mathcal{I}_{2,3}(1/t)\right) \leq 0, \tag{A4}$$

which implies that $h$ has at most two positive real roots. Therefore, we conclude that $h$ has exactly one positive real root.

$\square$

### A1.2 $\mathcal{I}$ and $\mathcal{I}^*$ inequalities

We now establish some useful properties of the $\mathcal{I}$ and $\mathcal{I}^*$ functionals defined in Eq. (14). To begin, we note that simple algebraic manipulations establish the following raising and lowering identities:

$$\mathcal{I}_{a-1,b-1} = \phi\mathcal{I}_{a-1,b} + x\mathcal{I}_{a,b} \quad \text{and} \quad \mathcal{I}_{a-1,b-1}^* = \phi\mathcal{I}_{a-1,b}^* + x\mathcal{I}_{a,b}^*. \tag{A5}$$

Next, we consider how the partial order of LJSDs given in Def. 4.1 leads to inequalities on the $\mathcal{I}^*$ functionals. We let $(\mathcal{I}_{a,b}^*)_1$ and $(\mathcal{I}_{a,b}^*)_2$ to denote the corresponding functionals with the LJSDs $\mu_1$ and $\mu_2$ respectively. Similarly, when comparing two different LJSDs we define $s_i$ as the test covariance scale for $\mu_i$ for $i \in \{1, 2\}$. We can then establish the following useful lemma used in the sequel.

**Lemma A1.3.** *Let $\mu_1 \leq \mu_2$, so $\mu_2$ is harder than $\mu_1$ (recall Def. 4.1). Suppose the functions $f, g, h : \mathbb{R} \to \mathbb{R}$ are such that $f(\lambda) = g(\lambda)h(\lambda)$, where $g(\lambda)$ is nonnegative and $h(\lambda)$ is nonincreasing for all $\lambda > 0$, then*

$$\frac{\mathbb{E}_{\mu_1}[rf(\lambda)]}{\mathbb{E}_{\mu_2}[rf(\lambda)]} \leq \frac{\mathbb{E}_{\mu_1}[rg(\lambda)]}{\mathbb{E}_{\mu_2}[rg(\lambda)]}. \tag{A6}$$

*If instead $h(\lambda)$ is nondecreasing for all $\lambda > 0$, then*

$$\frac{\mathbb{E}_{\mu_1}[rf(\lambda)]}{\mathbb{E}_{\mu_2}[rf(\lambda)]} \geq \frac{\mathbb{E}_{\mu_1}[rg(\lambda)]}{\mathbb{E}_{\mu_2}[rg(\lambda)]}. \tag{A7}$$

*Proof.* By the law of iterated expectation, we have

$$\mathbb{E}_{\mu_1}[rf(\lambda)] = \mathbb{E}_{\mu_2}[rg(\lambda)]\mathbb{E}_\lambda\left[\frac{\mathbb{E}_{\mu_2}[rg(\lambda)|\lambda]}{\mathbb{E}_{\mu_2}[rg(\lambda)]}\frac{\mathbb{E}_{\mu_1}[r|\lambda]}{\mathbb{E}_{\mu_2}[r|\lambda]}h(\lambda)\right]. \tag{A8}$$

Note that the expectation $\mathbb{E}_\lambda$ in Eq. (A8) over $\lambda$ is the same under $\mu_1$ and $\mu_2$ by assumption. Moreover, the function $h(\lambda)$ is nonincreasing in $\lambda > 0$ by assumption. Finally, observe that the factor $\mathbb{E}_{\mu_2}[rg(\lambda)|\lambda]/\mathbb{E}_{\mu_2}[rg(\lambda)]$ defines a change in distribution for the random variable $\lambda$, since taking its expectation over $\lambda$ yields 1. Denote a new random variable with this distribution by $\tilde{\lambda}$. Then, we may apply the Harris inequality[5] to Eq. (A8) to see

$$\mathbb{E}_{\mu_1}[rf(\lambda)] = \mathbb{E}_{\mu_2}[rg(\lambda)]\mathbb{E}_{\tilde{\lambda}}\left[\frac{\mathbb{E}_{\mu_1}[r|\tilde{\lambda}]}{\mathbb{E}_{\mu_2}[r|\tilde{\lambda}]}h(\tilde{\lambda})\right] \tag{A9}$$

$$\leq \mathbb{E}_{\mu_2}[rg(\lambda)]\mathbb{E}_{\tilde{\lambda}}\left[\frac{\mathbb{E}_{\mu_1}[r|\tilde{\lambda}]}{\mathbb{E}_{\mu_2}[r|\tilde{\lambda}]}\right]\mathbb{E}_{\tilde{\lambda}}\left[h(\tilde{\lambda})\right] \tag{A10}$$

$$= \mathbb{E}_{\mu_2}[rg(\lambda)]\mathbb{E}_\lambda\left[\frac{\mathbb{E}_{\mu_2}[rg(\lambda)|\lambda]}{\mathbb{E}_{\mu_2}[rg(\lambda)]}\frac{\mathbb{E}_{\mu_1}[r|\lambda]}{\mathbb{E}_{\mu_2}[r|\lambda]}\right]\mathbb{E}_\lambda\left[\frac{\mathbb{E}_{\mu_2}[rg(\lambda)|\lambda]}{\mathbb{E}_{\mu_2}[rg(\lambda)]}h(\lambda)\right] \tag{A11}$$

$$= \frac{\mathbb{E}_{\mu_1}[rg(\lambda)]}{\mathbb{E}_{\mu_2}[rg(\lambda)]}\mathbb{E}_{\mu_2}\left[rf(\lambda)\right]. \tag{A12}$$

To prove Eq. (A7), apply the same argument to $-h$, which is nonincreasing when $h$ is nondecreasing. $\square$

The following corollary is an immediate consequence of Lem. A1.3.

**Corollary A1.1.** *Let $\mu_1 \leq \mu_2$ (recall Def. 4.1). Then, for $a \geq 0$,*

$$\frac{(\mathcal{I}^*_{1,a})_2}{s_2} - \frac{(\mathcal{I}^*_{1,a})_1}{s_1} \geq 0. \tag{A13}$$

*If Assump. 4 also holds,*

$$(\mathcal{I}^*_{1,a})_2 \geq (\mathcal{I}^*_{1,a})_1. \tag{A14}$$

*Proof.* This result follows from Lem. A1.3 by choosing $g : \lambda \mapsto 1$ and $h : \lambda \mapsto \phi(\phi + x\lambda)^{-a}$ and recalling by definitions $s_1 = \mathbb{E}_{\mu_1}[r]$ and $s_2 = \mathbb{E}_{\mu_2}[r]$. For the second conclusion, note when Assump. 4 holds $s_1 = s = s_2$. $\square$

## A2   Hardness is a partial order

We restate Def. 4.1 for clarity.

**Definition A2.1** (Restatement of Def. 4.1). *Let $\mu_1$ and $\mu_2$ be LJSDs with the same marginal distribution of $\lambda$. If the asymptotic overlap coefficients are such that $\mathbb{E}_{\mu_1}[r|\lambda]/\mathbb{E}_{\mu_2}[r|\lambda]$ is nondecreasing as a function of $\lambda$ and $\mathbb{E}_{\mu_1}[r] \leq \mathbb{E}_{\mu_2}[r]$, we say $\mu_1$ is* easier *than $\mu_2$ (or $\mu_2$ is* harder *than $\mu_1$), and write $\mu_1 \leq \mu_2$. Comparing against the case of no shift $\mu_\emptyset$, we say $\mu_1$ is* easy *when $\mu_1 \leq \mu_\emptyset$ and* hard *when $\mu_1 \geq \mu_\emptyset$.*

---
[5]See for example [23, Section 2.2].

**Proposition A2.1.** *Def. 4.1 (and Def. A2.1) form a partial order over covariate shifts $\mu$.*

*Proof.* Reflexivity is clearly satisfied as $\mathbb{E}_\mu[r] \leq \mathbb{E}_\mu[r]$ and $\mathbb{E}_\mu[r|\lambda]/\mathbb{E}_\mu[r|\lambda] = 1$ is nondecreasing for all $\mu$.

For antisymmetry, we see $\mu_1 \leq \mu_2$ and $\mu_2 \leq \mu_1$ imply $\mathbb{E}_{\mu_1}[r] = \mathbb{E}_{\mu_2}[r]$ and that $\mathbb{E}_{\mu_1}[r|\lambda]/\mathbb{E}_{\mu_2}[r|\lambda]$ is constant in $\lambda$ as it is nonincreasing and nondecreasing. However, setting $\mathbb{E}_{\mu_1}[r|\lambda] = c\mathbb{E}_{\mu_2}[r|\lambda]$ and taking expectation over $\lambda$ and rearranging yields $1 = \mathbb{E}_{\mu_1}[r]/\mathbb{E}_{\mu_2}[r] = c$, so in fact $\mathbb{E}_{\mu_1}[r|\lambda] = \mathbb{E}_{\mu_2}[r|\lambda]$. Assuming that $\mu_1$ and $\mu_2$ are absolutely continuous (the case where they are a sum of point masses is similar), we can write their densities as $p_i(\lambda, r) = p_i(\lambda)p_i(r|\lambda)$. By assumption $p_1(\lambda) = p_2(\lambda)$, so it suffices to show $p_1(r|\lambda) = p_2(r|\lambda)$ almost everywhere. Next note

$$0 = \mathbb{E}_{\mu_1}[r|\lambda] - \mathbb{E}_{\mu_2}[r|\lambda] = \int_{\mathbb{R}^+} r\,(p_1(r|\lambda) - p_2(r|\lambda))\,dr, \tag{A15}$$

but since $r > 0$ over the domain of the integral, we have $p_1(r|\lambda) - p_2(r|\lambda) = 0$ almost everywhere.

Finally, for transitivity assume $\mu_1 \leq \mu_2$ and $\mu_2 \leq \mu_3$, then clearly $\mathbb{E}_{\mu_1}[r] \leq \mathbb{E}_{\mu_2}[r] \leq \mathbb{E}_{\mu_3}[r]$. Next note

$$\frac{\mathbb{E}_{\mu_1}[r|\lambda]}{\mathbb{E}_{\mu_3}[r|\lambda]} = \frac{\mathbb{E}_{\mu_1}[r|\lambda]}{\mathbb{E}_{\mu_2}[r|\lambda]} \cdot \frac{\mathbb{E}_{\mu_2}[r|\lambda]}{\mathbb{E}_{\mu_3}[r|\lambda]}, \tag{A16}$$

so $\mathbb{E}_{\mu_1}[r|\lambda]/\mathbb{E}_{\mu_3}[r|\lambda]$ is the product of two nondecreasing, positive functions and is thus also nondecreasing. $\square$

## A3 Repeated eigenvalues of $\Sigma$

When $\Sigma$ has repeated eigenvalues (denote one such by $\lambda$), its eigendecomposition is not unique, since the eigenvectors associated to $\lambda$ need only span the eigenspace $\{\mathbf{v} : \Sigma\mathbf{v} = \lambda\mathbf{v}\}$. Specifically, if the eigenspace of $\lambda$ has dimension $n_\lambda$ then the eigenvectors $\mathbf{v}_1^\lambda, \ldots, \mathbf{v}_{n_\lambda}^\lambda$ are orthonormal but not unique. However, for any other choice of orthonormal vectors $\mathbf{w}_1^\lambda, \ldots, \mathbf{w}_{n_\lambda}^\lambda$ that span the eigenspace of $\lambda$, there exists some orthogonal matrix $O$ such that $W = VO$, where $V$ and $W$ contain the two bases as their columns.

Let $\mathbf{v}_1^*, \ldots, \mathbf{v}_{n_0}^*$ and $\lambda_1^*, \ldots, \lambda_{n_0}^*$ denote a choice for the eigenvectors and eigenvalues of $\Sigma^*$. The nonuniqueness of the eigenvectors implies that the corresponding overlaps to $\Sigma^*$, defined as

$$r_i^\lambda = \sum_{j=1}^{n_0} (\mathbf{v}_j^* \cdot \mathbf{v}_i^{*\lambda})^2 \lambda_j^* = (\mathbf{v}_i^\lambda)^\top \Sigma^* \mathbf{v}_i^\lambda \tag{A17}$$

are also not unique (but do not depend on the choice of eigendecomposition for $\Sigma^*$). However, we note that the conditional expectation $\mathbb{E}[r|\lambda]$ for the EJSD is invariant to the choice of eigendecomposition for $\Sigma$. Indeed, for $V$, we have

$$\mathbb{E}[r|\lambda] = \frac{1}{n_\lambda} \sum_{i=1}^{n_\lambda} r_i^\lambda = \frac{1}{n_\lambda} \sum_{i=1}^{n_\lambda} (\mathbf{v}_i^\lambda)^\top \Sigma^* \mathbf{v}_i^\lambda = \bar{\mathrm{tr}}(V^\top \Sigma^* V), \tag{A18}$$

but this is the same as $\bar{\mathrm{tr}}(W^\top \Sigma^* W)$ using $W = VO$ and the cyclic property of the trace.

Since $\mathbb{E}[r|\lambda]$ under the EJSD is invariant to the choice of eigendecomposition for $\Sigma$, this will also be true of the LJSD. Said differently, while the choice of eigendecomposition affects both the EJSD and its corresponding LJSD, all possible choices of eigendecomposition lead to JSD in the same equivalence class, where $\mu_1$ and $\mu_2$ are equivalent when $\mathbb{E}_{\mu_1}[r|\lambda] = \mathbb{E}_{\mu_2}[r|\lambda]$.

Finally, since all potential EJSDs associated to $\Sigma$ and $\Sigma^*$ are in the same equivalence class, the particular choice has no affect on their downstream use. Specifically, Def. 4.1 is not changed as it depends only on $\mathbb{E}[r|\lambda]$, and none of the functionals of $\mu$ (e.g. $\mathcal{I}_{a,b}$ and $\mathcal{I}_{a,b}^*$) are changed due to the law of iterated expectation.

## A4 Test error for linear regression

### A4.1 Asymptotic and nonasymptotic results for $m > n_0 + 1$

We present a short proof of the nonasymptotic test error for linear regression.

Recall data is generated via the model for $y_i = \beta^\top \mathbf{x}_i / \sqrt{n_0} + \epsilon_i$, which is fit with the ridgeless linear regression estimator $\hat{\beta} = (XX^\top)^{-1} XY$. Although in the main text we have assumed the same isotropic prior on $\beta$ that we utilize for the random feature model, no generative assumption is needed on $\beta$ for this result.

*Proof of Prop. 3.1.* Under the condition $m > n_0 + 1$, the sample covariance is almost surely invertible so the test error can be written as

$$E_{\Sigma^*} = \mathbb{E}\left[\left(\beta^\top \mathbf{x}/\sqrt{n_0} - \hat{\beta}^\top \mathbf{x}\right)^2\right] \tag{A19}$$

$$= \sigma_\epsilon^2 \mathrm{tr}\left(\Sigma^* \mathbb{E}[(XX^\top)^{-1}]\right) \tag{A20}$$

$$= \sigma_\epsilon^2 \frac{n_0}{m - n_0 - 1} \mathrm{tr}(\Sigma^* \Sigma^{-1}), \tag{A21}$$

where we used the cyclicity and linearity of the trace, as well as a formula for the expectation of the inverse sample covariance of a Gaussian matrix (which applies when $m > n_0 + 1$) from [58, Theorem 3.1]. The asymptotic form of the result follows from the limit in the proportional asymptotics:

$$E_\mu = \lim_{n_0, m \to \infty} E_{\Sigma^*} = \frac{\sigma_\epsilon^2 \phi}{1 - \phi} \mathbb{E}_\mu[r/\lambda]. \tag{A22}$$

$\square$

Next we prove Prop. 3.2 using the Harris inequality before proving a slightly more general result that properly handles the case where $\Sigma$ may have repeated eigenvalues.

*Proof of Prop. 3.2.* Using the Harris inequality,

$$E_{\Sigma_1^*}^{\mathrm{LR}} = \frac{\sigma_\epsilon^2 n_0}{m - n_0 - 1} \frac{\bar{\mathrm{tr}}(\Sigma_2^*)}{n_0} \sum_{i=1}^{n_0} \frac{r_{i,2}}{\bar{\mathrm{tr}}(\Sigma_2^*)} \frac{r_{i,1}}{r_{i,2}} \frac{1}{\lambda_i} \leq \frac{\sigma_\epsilon^2 n_0}{m - n_0 - 1} \frac{\bar{\mathrm{tr}}(\Sigma_1^*)}{\bar{\mathrm{tr}}(\Sigma_2^*)} \frac{1}{n_0} \sum_{i=1}^{n_0} \frac{r_{i,2}}{\lambda_i} \leq E_{\Sigma_2^*}^{\mathrm{LR}}, \tag{A23}$$

since $1/\lambda_i$ and $r_{i,1}/r_{i,2}$ are nonincreasing and nondecreasing in $i$ respectively. $\square$

Prop. 3.2 can be strengthened, and doing so motivates the occurrence of the conditional expectation in Def. 4.1. Note that we assume that the eigenvalues of $\Sigma$ are in nondecreasing order, and each eigenvalue has associated to it two overlap coefficients, $r_{i,1}$ and $r_{i,2}$. Prop. 3.2 assumes that $r_{i,1}/r_{i,2}$ form a nondecreasing sequence. However, what happens when $\Sigma$ has repeated eigenvalues? In this case, the ordering of the $\lambda_i$ can be changed, which in turn changes the associated $r_{i,1}$ and $r_{i,2}$. Therefore, the assumption on $r_{i,1}/r_{i,2}$ is too strong—reordering the repeated eigenvalues might be sufficient to satisfy the condition even if it is violated for the original ordering. Instead, we can introduce the conditional expectation to handle this more gracefully.

In the following, we use $\tilde{\lambda}_1, \ldots, \tilde{\lambda}_k$ to denote the $k$ non-repeated eigenvalues of the training covariance $\Sigma$ and the sets $S_j$ for $j \in \{1, \ldots, k\}$ to denote the indices of eigenvalues in $\{1, \ldots, n_0\}$ associated to the $j$th repeated eigenvalue of $\Sigma$. Analogously, we define the corresponding non-repeated overlap coefficients as,

$$\tilde{r}_j = \frac{1}{|S_j|} \sum_{i \in S_j} r_i, \quad j \in \{1, \ldots, k\} \tag{A24}$$

This is equivalent to the conditional expectation discussed in Eq. (A18). In this case, the measure-theoretic definition of *hardness* in Def. 4.1 becomes equivalent to stating that the sequence $\frac{\tilde{r}_{j,2}}{\tilde{r}_{j,1}}$ is nondecreasing as $j$ ranges from $\{1, \ldots, k\}$ when $\Sigma_2^*$ is *harder* then $\Sigma_1^*$ (here $\tilde{r}_{j,1}$ and $\tilde{r}_{j,2}$ denote the non-repeated overlap coefficients of $\Sigma_1^*$ and $\Sigma_2^*$ respectively).

**Proposition A4.1.** *Let $\tilde{r}_{j,1}$ and $\tilde{r}_{j,2}$ denote the non-repeated overlap coefficients[6] of $\Sigma_1^*$ and $\Sigma_2^*$ relative to $\Sigma$. If $\operatorname{tr}(\Sigma_2^*) \geq \operatorname{tr}(\Sigma_1^*)$ and the ratios $\tilde{r}_{j,1}/\tilde{r}_{j,2}$ form a nondecreasing sequence, then $E_{\Sigma_2^*}^{LR} \geq E_{\Sigma_1^*}^{LR}$*

*Proof.* From Prop. 3.1 it is enough to show $\operatorname{tr}(\Sigma_2^* \Sigma^{-1}) \geq \operatorname{tr}(\Sigma_1^* \Sigma^{-1})$. First, note that $\sum_{i=1}^{n_0} r_{i,1} = \sum_{j=1}^{k} |S_j| \tilde{r}_{j,1} = \operatorname{tr}(\Sigma_1^*)$ and $\sum_{i=1}^{n_0} r_{i,2} = \sum_{j=1}^{k} |S_j| \tilde{r}_{j,2} = \operatorname{tr}(\Sigma_2^*)$. Then,

$$\operatorname{tr}(\Sigma_1^* \Sigma^{-1}) = \sum_{i=1}^{n_0} \frac{r_{i,1}}{\lambda_i} = \sum_{j=1}^{k} |S_j| \frac{\tilde{r}_{j,1}}{\tilde{\lambda}_j} = \operatorname{tr}(\Sigma_2^*) \sum_{j=1}^{k} |S_j| \frac{\tilde{r}_{j,2}}{\operatorname{tr}(\Sigma_2^*)} \frac{\tilde{r}_{j,1}}{\tilde{r}_{j,2}} \frac{1}{\tilde{\lambda}_j} \tag{A25}$$

$$\leq \operatorname{tr}(\Sigma_2^*) \left( \sum_{j=1}^{k} |S_j| \frac{\tilde{r}_{j,2}}{\operatorname{tr}(\Sigma_2^*)} \frac{\tilde{r}_{j,1}}{\tilde{r}_{j,2}} \right) \left( \sum_{j=1}^{k} |S_j| \frac{\tilde{r}_{j,2}}{\operatorname{tr}(\Sigma_2^*)} \frac{1}{\tilde{\lambda}_j} \right) \tag{A26}$$

$$= \frac{\operatorname{tr}(\Sigma_1^*)}{\operatorname{tr}(\Sigma_2^*)} \operatorname{tr}(\Sigma_2^* \Sigma^{-1}) \tag{A27}$$

$$\leq \operatorname{tr}(\Sigma_2^* \Sigma^{-1}), \tag{A28}$$

where the inequality is a consequence of the Harris inequality (see e.g. [23, Section 2.2]): since $|S_j| \tilde{r}_{j,1}/\tilde{\lambda}_j$ is a normalized measured with respect to $j$, the sequence $1/\tilde{\lambda}_j$ is nonincreasing in $j$, while the sequence $\tilde{r}_{j,1}/\tilde{r}_{j,2}$ is nondecreasing in $j$. □

### A4.2 Linear regression limit of random feature regression

In this section, we show that taking an appropriate limit of Cor. 5.1 recovers existing results for ridgeless linear regression in high-dimensions. These results fall into two cases based on whether $\phi < 1$ or $\phi > 1$. In Sec. A4.2.1, we consider $\phi < 1$ and Prop. 3.1 from the main text. In Sec. A4.2.2, we consider $\phi > 1$ and Cor. 2 of [24].

To recover these results, we let $\sigma$ approach the identity activation function and $\psi \to 0$[7], since as more random features are added the kernel concentrate around its asymptotic limit. Moreover, since we are taking the limit $\psi \to 0$ and we assume $\phi > 0$, we may assume $\phi > \psi$ in all calculations for simplicity. We also note that as $\sigma$ approaches the identity, the constants associated to $\sigma$ approach the following limits:

$$\eta, \zeta \to 1 \quad \text{and} \quad \omega \to 0. \tag{A29}$$

From Cor. 5.1, the expression for the total error is

$$E_\mu = \phi \mathcal{I}_{1,2}^* + \frac{\psi}{\phi - \psi} x(\sigma_\varepsilon^2 + \mathcal{I}_{1,1})(\omega + \mathcal{I}_{1,1}^*) + x\left(1 - \frac{x(\omega - \sigma_\varepsilon^2)}{1 - x^2 \mathcal{I}_{2,2}}\right) \mathcal{I}_{2,2}^*, \tag{A30}$$

where $x = 1/(\omega + \mathcal{I}_{1,1})$ since we are assuming $\psi < \phi$. Taking the limit $\psi \to 0$, the total error $E_\mu$ converges to

$$\phi \mathcal{I}_{1,2}^* + x\left(1 - \frac{x(\omega - \sigma_\varepsilon^2)}{1 - x^2 \mathcal{I}_{2,2}}\right) \mathcal{I}_{2,2}^*. \tag{A31}$$

#### A4.2.1 Recovering asymptotic form of Prop. 3.1 for $\phi < 1$

Recall that Prop. 3.1 assumes that $\phi < 1$. We begin by analyzing the solution to the self-consistent equation for $x$ in the ridgeless limit when the activation function $\sigma$ becomes linear.

**Lemma A4.1.** *Suppose $0 < \phi < 1$ and $\psi < \phi$. In the ridgeless limit, the solution $x$ to the self-consistent equation in Cor. 5.1,*

$$x = \frac{1}{\omega + \mathcal{I}_{1,1}}, \tag{A32}$$

*satisfies $\lim_{\omega \to 0} x = \infty$.*

---

[6]Recall the definition in Eq. (A24).

[7]The order of these limits does not change the result, but for concreteness we take take the limit as $\psi \to 0$ first.

*Proof.* From the definition of $\mathcal{I}_{1,1}$, we see

$$\mathcal{I}_{1,1} = \phi\mathbb{E}_\mu\left[\frac{\lambda}{\phi + \lambda x}\right] = \frac{\phi}{x}\mathbb{E}_\mu\left[\frac{\lambda}{\phi/x + \lambda}\right] \leq \frac{\phi}{x}. \tag{A33}$$

Using Eq. (A32), we find $x \geq (1 - \phi)/\omega$. Taking $\omega \to 0$ completes the proof. $\qquad\square$

From Lem. A4.1, the solution to the self-consistent equation for $x$ diverges, i.e. $x \to \infty$ as $\sigma$ becomes linear. In this case, by the dominated convergence theorem, we have that

$$\lim_{\omega\to 0} x\mathcal{I}_{1,1}^* = \lim_{x\to\infty} \mathbb{E}_\mu\left[\frac{xr}{\phi + x\lambda}\right] = \phi\mathbb{E}_\mu\left[r/\lambda\right], \tag{A34}$$

$$\lim_{\omega\to 0} x^2\mathcal{I}_{2,2} = \lim_{x\to\infty} \phi\mathbb{E}_\mu\left[\frac{x^2\lambda^2}{(\phi + x\lambda)^2}\right] = \phi, \tag{A35}$$

$$\lim_{\omega\to 0} x^2\mathcal{I}_{2,2}^* = \lim_{x\to\infty} \phi\mathbb{E}_\mu\left[\frac{x^2 r\lambda}{(\phi + x\lambda)^2}\right] = \phi\mathbb{E}_\mu\left[r/\lambda\right], \tag{A36}$$

$$\lim_{\omega\to 0} \mathcal{I}_{1,1}^* = \lim_{x\to\infty} \phi\mathbb{E}_\mu\left[\frac{r}{(\phi + x\lambda)^2}\right] = 0, \tag{A37}$$

$$\text{and} \quad \lim_{\omega\to 0} \mathcal{I}_{1,2}^* = \lim_{x\to\infty} \phi\mathbb{E}_\mu\left[\frac{r}{(\phi + x\lambda)^2}\right] = 0. \tag{A38}$$

As such, the total error in Eq. (A31) converges to

$$\sigma_\epsilon^2\frac{\phi}{1 - \phi}\mathbb{E}_\mu[r/\lambda] \tag{A39}$$

as $\omega \to 0$ and $\psi \to 0$ as desired.

### A4.2.2 Recovering results from [24] when $\Sigma = \Sigma^*$ and $\phi > 1$

The minimum-norm solution for under-determined linear regression is the same as the limiting ridge-regularized solution as the ridge constant converges to 0. As such, studying the ridgeless limit as in Cor. 5.1 allows for a comparison to prior results on minimum-norm interpolants [24]. To do so, we again take the limit as $\sigma$ becomes linear; however, in contrast to the previous section, we now assume $\phi > 1$ in order to compare with [24, Cor. 2].

As [24] examines the setting in which the training and test covariate distributions are equal, $\Sigma = \Sigma^*$, we have that $\mathcal{I}_{a,b}^* = \mathcal{I}_{a,b}$. Using this relation, the total error Eq. (A31) becomes

$$E_\mu = \phi\mathcal{I}_{1,2} + x\left(1 - \frac{x(\omega - \sigma_\varepsilon^2)}{1 - x^2\mathcal{I}_{2,2}}\right)\mathcal{I}_{2,2}. \tag{A40}$$

Next, letting $\sigma$ become linear as in Eq. (A29), we obtain $x = 1/\mathcal{I}_{1,1}$ and

$$E_\mu \xrightarrow{\omega\to 0} \mathcal{I}_{1,1} + \sigma_\epsilon^2\frac{x^2\mathcal{I}_{2,2}}{1 - x^2\mathcal{I}_{2,2}}, \tag{A41}$$

where we used the identity Eq. (A5).

We now simplify and relate the result for the asymptotic error from [24, Cor. 2] in the case of isotropic $\beta$ satisfying $\|\beta\|_2 = 1$, where the total error is written as[8]

$$\frac{1}{\phi^2 c_0(H, \phi)} + \sigma_\epsilon^2 c_0\phi\frac{\int \frac{s^2}{(1 + c_0\phi s)^2}dH(s)}{\int \frac{s}{(1 + c_0\phi s)^2}dH(s)}. \tag{A42}$$

The measure $H$ is the limiting empirical spectral density of the covariance, which is equivalent to the marginal distribution of $\lambda$, and $c_0$ satisfies the equation

$$1 - \frac{1}{\phi} = \int \frac{1}{1 + c_0\phi s}dH(s). \tag{A43}$$

---

[8][24, Cor. 2] provides expressions for a bias and variance term separately, but the decomposition is defined slightly differently than the one we utilize, so we compare directly to the total error. Note that Eq. (A42) corrects a typo in [24].

Substituting $c_0 = x/\phi^2$ and rearranging terms, we see

$$0 = 1 - \frac{1}{\phi} - \int \frac{1}{1 + c_0 \phi s} dH(s) \tag{A44}$$

$$= -\frac{1}{\phi} + 1 - \int \frac{1}{1 + (x/\phi)s} dH(s) \tag{A45}$$

$$= -\frac{1}{\phi} + x \int \frac{s}{\phi + xs} dH(s) \tag{A46}$$

$$= -\frac{x}{\phi}\left(\frac{1}{x} - \mathcal{I}_{1,1}\right), \tag{A47}$$

which is satisfied by $x = 1/\mathcal{I}_{1,1}$, implying the two self-consistent equations are equivalent and validating the identification $c_0 = x/\phi^2$. Using the same substitution $c_0 = x/\phi^2$ in Eq. (A42) gives

$$\mathcal{I}_{1,1} + \sigma_\epsilon^2 \frac{1}{x\phi} \frac{x^2 \mathcal{I}_{2,2}}{\mathcal{I}_{1,2}} = \mathcal{I}_{1,1} + \sigma_\epsilon^2 \frac{x^2 \mathcal{I}_{2,2}}{1 - x^2 \mathcal{I}_{2,2}}, \tag{A48}$$

which matches our expression in Eq. (A41).

## A5  Harder shifts increase the bias and the total error

To begin, we provide the proof of Prop. 5.1.

*Proof of Prop. 5.1.* Consider two LJSDs $\mu_1$ and $\mu_2$ such that $\mu_1 \leq \mu_2$ where Assump. 4 holds. Then,

$$B_{\mu_2} - B_{\mu_1} = \phi\left((\mathcal{I}_{1,2}^*)_2 - (\mathcal{I}_{1,2}^*)_1\right) \geq 0, \tag{A49}$$

where we have used the inequalities in Eq. (A60) which follow from Cor. A1.1 when Assump. 4 holds.

Now, additionally assume that $\sigma_\varepsilon^2 \leq \omega$. Reorganizing the terms specifying the asymptotic variance, we can rewrite the total error as

$$E_\mu = B_\mu + V_\mu \tag{A50}$$

$$= \phi(\mathcal{I}_{1,2}^*) - \rho \frac{\psi}{\phi} \frac{\partial x}{\partial \gamma}\Bigg((\sigma_\varepsilon^2 + \mathcal{I}_{1,1})(\omega + \phi \mathcal{I}_{1,2})(\omega + \mathcal{I}_{1,1}^*) + \gamma\tau\mathcal{I}_{2,2}(\omega + \phi\mathcal{I}_{1,2}^*)$$

$$+ \frac{\phi}{\psi x}\gamma\bar{\tau}(\sigma_\varepsilon^2 + \phi\mathcal{I}_{1,2})(\mathcal{I}_{1,1}^* - \phi\mathcal{I}_{1,2}^*)\Bigg) \tag{A51}$$

$$\equiv C_1\omega + C_2\mathcal{I}_{1,1}^* + C_3\mathcal{I}_{1,2}^*, \tag{A52}$$

where the $C_i \geq 0$ and depend on $\mu$ only through the marginal $\lambda$ (i.e. they only depend on the training distribution):

$$C_1 = -\rho\frac{\psi}{\phi}\frac{\partial x}{\partial \gamma}\left((\sigma_\varepsilon^2 + \mathcal{I}_{1,1})(\omega + \phi\mathcal{I}_{1,2}) + \gamma\tau\mathcal{I}_{2,2}\right) \geq 0 \tag{A53}$$

$$C_2 = -\rho\frac{\partial x}{\partial \gamma}\left(\frac{\psi}{\phi}(\sigma_\varepsilon^2 + \mathcal{I}_{1,1})(\omega + \phi\mathcal{I}_{1,2}) + \frac{\gamma\bar{\tau}}{x}(\sigma_\varepsilon^2 + \phi\mathcal{I}_{1,2})\right) \geq 0 \tag{A54}$$

$$C_3 = -\rho\frac{\partial x}{\partial \gamma}\left(\psi\gamma\tau\mathcal{I}_{2,2} - \frac{\phi\gamma\bar{\tau}}{x}(\sigma_\varepsilon^2 + \phi\mathcal{I}_{1,2}) - \frac{\phi}{\rho\frac{\partial x}{\partial \gamma}}\right) \tag{A55}$$

$$= -\rho\frac{\partial x}{\partial \gamma}\left(\psi\gamma\tau\mathcal{I}_{2,2} - \frac{\phi\gamma\bar{\tau}}{x}(\sigma_\varepsilon^2 + \phi\mathcal{I}_{1,2}) + \frac{\phi}{\rho x}(\gamma + \rho\gamma(\tau\psi/\phi + \bar{\tau})(\omega + \phi\mathcal{I}_{1,2}))\right) \tag{A56}$$

$$= -\rho\gamma\frac{\partial x}{\partial \gamma}\left(\psi\tau\mathcal{I}_{2,2} + \frac{\phi\bar{\tau}}{x}(\omega - \sigma_\varepsilon^2) + \frac{\phi}{\rho x}\left(1 + \rho\tau\frac{\psi}{\phi}(\omega + \phi\mathcal{I}_{1,2})\right)\right) \tag{A57}$$

$$\geq 0, \tag{A58}$$

where the inequalities follow from Lem. A1.1 and from the assumption $\sigma_\varepsilon^2 \leq \omega$. It is now straightforward to see that

$$
\begin{aligned}
E_{\mu_2} - E_{\mu_1} &= \left(C_1\omega + C_2(\mathcal{I}_{1,1}^*)_2 + C_3(\mathcal{I}_{1,2}^*)_2\right) - \left(C_1\omega + C_2(\mathcal{I}_{1,1}^*)_1 + C_3(\mathcal{I}_{1,2}^*)_1\right) \\
&= C_2\left((\mathcal{I}_{1,1}^*)_2 - (\mathcal{I}_{1,1}^*)_1\right) + C_3\left((\mathcal{I}_{11,2}^*)_2 - (\mathcal{I}_{1,2}^*)_1\right) \geq 0\,,
\end{aligned}
\tag{A59}
$$

where we have used the inequalities

$$
(\mathcal{I}_{1,1}^*)_2 - (\mathcal{I}_{1,1}^*)_1 \geq 0\,, \quad (\mathcal{I}_{1,2}^*)_2 - (\mathcal{I}_{1,2}^*)_1 \geq 0\,.
\tag{A60}
$$

The former two inequalities again follow from Cor. A1.1 when Assump. 4 holds. □

## A6 The benefit of overparameterization

### A6.1 The bias is nonincreasing

We begin by examining the behavior of the bias as a function of the overparameterization ratio $\phi/\psi$ by showing Prop. 5.2.

*Proof of Prop. 5.2.* Recall from Thm. 5.1 that the bias is given by

$$
B_\mu = \phi\mathcal{I}_{1,2}^*\,,
\tag{A61}
$$

where $x$ is the unique positive real root of the self-consistent equation

$$
x = \frac{1 - \gamma\tau}{\omega + \mathcal{I}_{1,1}}\,.
\tag{A62}
$$

Differentiating Eq. (A61) with respect to $\phi/\psi$ gives,

$$
\frac{\partial B_\mu}{\partial(\phi/\psi)} = -\frac{\psi^2}{\phi}\frac{\partial B_\mu}{\partial\psi} = 2\frac{\psi^2}{\phi}\frac{\partial x}{\partial\psi} \cdot \left(\phi\mathcal{I}_{2,3}^*\right)\,.
\tag{A63}
$$

Since Lem. A1.1 gives $\mathcal{I}_{a,b}^* \geq 0$, it is sufficient to show $\frac{\partial x}{\partial\psi} \leq 0$, which immediately follows by implicitly differentiating Eq. (A62) and simplifying the expression:

$$
\frac{\partial x}{\partial\psi} = -\frac{\rho x\tau(\omega + \mathcal{I}_{1,1})}{\phi\left(1 + \rho(\bar{\tau} + \frac{\psi}{\phi}\tau)(\omega + \phi\mathcal{I}_{1,2})\right)} \leq 0\,.
\tag{A64}
$$

Therefore we conclude that $\frac{\partial B_\mu}{\partial(\phi/\psi)} \leq 0$.

□

### A6.2 The variance is nonincreasing

Next, we turn our attention to the variance. We note the proposition only focuses on the ridgeless limit, whereas Fig. 1(d) show that result may in fact hold for nonzero ridge constant. As the proof is considerably simpler in the ridgeless limit, we defer the analysis of the nonzero ridge setting to future work.

*Proof of Prop. 5.3.* Using the chain rule we have that

$$
\frac{\partial V_\mu}{\partial(\phi/\psi)} = \frac{\partial V_\mu}{\partial\psi}\left[\frac{\partial(\phi/\psi)}{\partial\psi}\right]^{-1} = -\frac{\partial V_\mu}{\partial\psi}\frac{\psi^2}{\phi}\,,
\tag{A65}
$$

so it is sufficient to show that $\frac{\partial V_\mu}{\partial\psi} \geq 0$.

From Cor. 5.1 in the overparameterized regime, the self-consistent equation for $x$ reads $x = \frac{1}{\omega + \mathcal{I}_{1,1}}$ and is independent of $\psi$. Therefore, $\partial x/\partial\psi = 0$ and the expression for $\partial V_\mu/\partial\psi$ follows directly from Eq. (18),

$$
\frac{\partial V_\mu}{\partial\psi} = \frac{\phi}{(\phi - \psi)^2}x(\sigma_\varepsilon^2 + \mathcal{I}_{1,1})(\omega + \mathcal{I}_{1,1}^*) \geq 0\,,
\tag{A66}
$$

where the inequality follows from Lem. A1.1. □

### A6.3 The generalization gap is nonincreasing

Finally, we prove that overparameterization also confers enhanced robustness, specifically that the generalization gap between shifted and unshifted test error is a nonincreasing function of the overparameterization ratio in the overparameterized regime.

*Proof of Prop. 5.4.* The chain rule gives $\frac{\partial(E_{\mu_2}-E_{\mu_1})}{\partial(\phi/\psi)} = -\frac{\psi^2}{\phi}\frac{\partial(E_{\mu_2}-E_{\mu_1})}{\partial\psi}$ so it is sufficient to show $\frac{\partial(E_{\mu_2}-E_{\mu_1})}{\partial\psi} \geq 0$. To that end, recall from Cor. 5.1 in the overparameterized regime that the self-consistent equation for $x$ reads $x = \frac{1}{\omega + \mathcal{I}_{1,1}}$ and is independent of $\psi$. Therefore, $\partial x/\partial\psi = 0$, which implies $\partial B_{\mu_1}/\partial\psi = 0$ and $\partial B_{\mu_2}/\partial\psi = 0$, so that $\frac{\partial(E_{\mu_2}-E_{\mu_1})}{\partial\psi} = \frac{\partial(V_{\mu_2}-V_{\mu_1})}{\partial\psi}$, the expression for which follows directly from Eq. (18). Finally,

$$\frac{\partial(E_{\mu_2}-E_{\mu_1})}{\partial\psi} = \left(C_E(\omega+(\mathcal{I}^*_{1,1})_2)\right) - \left(C_E(\omega+(\mathcal{I}^*_{1,1})_1)\right) \tag{A67}$$

$$\geq C_E\left((\mathcal{I}^*_{1,1})_2 - (\mathcal{I}^*_{1,1})_1\right) \tag{A68}$$

$$= 0\,, \tag{A69}$$

where we have introduced the shorthand $C_E = \frac{\phi}{(\phi-\psi)^2}x(\sigma^2_\varepsilon + \mathcal{I}_{1,1}) \geq 0$ and in the last inequality we have used Cor. A1.1 to argue $(\mathcal{I}^*_{1,1})_2 \geq (\mathcal{I}^*_{1,1})_1$ which holds whenever Assump. 4 holds. $\square$

## A7 Linear trends between in-distribution and out-of-distribution generalization

In Sec. 5.4, we investigated the linear relationship between the shifted and unshifted test error in the ridgeless, overparameterized regime. Here, we generalize the result to show that the linear relationship holds between any two LJSDs in the ridgeless, overparameterized regime.

**Proposition A7.1** (Strengthened form of Prop. 5.5). *Consider two LJSDs $\mu_1$ and $\mu_2$. In the setting of Cor. 5.1 and in the overparameterized regime (i.e. $\psi < \phi$),*

$$E_{\mu_2} = E_0 + \underbrace{\left(\frac{\omega+(\mathcal{I}^*_{1,1})_2}{\omega+(\mathcal{I}^*_{1,1})_1}\right)}_{\text{SLOPE}} E_{\mu_1}\,, \tag{A70}$$

*parametrically in $\phi/\psi$, where $E_0$ and SLOPE are constants independent of the overparameterization ratio $\phi/\psi$. Moreover, SLOPE $\geq 1$ when $\mu_1 \leq \mu_2$.*

*Proof of Prop. A7.1.* In the overparameterized regime (i.e. $\phi > \psi$) and in the ridgeless limit, the asymptotic test error for LJSD $\mu_1$ is given by Cor. 5.1 as

$$E_{\mu_1} = B_{\mu_1} + \frac{1}{\phi/\psi - 1}x(\sigma^2_\varepsilon + \mathcal{I}_{1,1})(\omega+(\mathcal{I}^*_{1,1})_1) + x\left(1 - \frac{x(\omega-\sigma^2_\varepsilon)}{1 - x^2\mathcal{I}_{2,2}}\right)(\mathcal{I}^*_{2,2})_1\,. \tag{A71}$$

Similarly, the asymptotic test error for LJSD $\mu_2$ is given by Cor. 5.1 as

$$E_{\mu_2} = B_{\mu_2} + \frac{1}{\phi/\psi - 1}x(\sigma^2_\varepsilon + \mathcal{I}_{1,1})(\omega+(\mathcal{I}^*_{1,1})_2) + x\left(1 - \frac{x(\omega-\sigma^2_\varepsilon)}{1 - x^2\mathcal{I}_{2,2}}\right)(\mathcal{I}^*_{2,2})_2\,. \tag{A72}$$

Note that in both expressions, the self-consistent equation for $x$ reads $x = \frac{1}{\omega + \mathcal{I}_{1,1}}$, which has no dependence on $\psi$, and, as such, $x$, $\mathcal{I}_{a,b}$, $(\mathcal{I}^*_{a,b})_1$, $(\mathcal{I}^*_{a,b})_2$, $B_{\mu_1}$, and $B_{\mu_2}$ do not depend on the overparameterization ratio $\phi/\psi$. Hence we can simply eliminate the quantity $\frac{1}{\phi/\psi-1}$ from Eqs. (A71) and (A72) to obtain

$$E_{\mu_2} = E_0 + \left(\frac{\omega+(\mathcal{I}^*_{1,1})_2}{\omega+(\mathcal{I}^*_{1,1})_1}\right)E_{\mu_1}\,, \tag{A73}$$

where $E_0$ does not depend on the overparameterization ratio $\phi/\psi$ and is given by

$$E_0 := B_{\mu_2} - \frac{\omega+(\mathcal{I}^*_{1,1})_2}{\omega+(\mathcal{I}^*_{1,1})_1}B_{\mu_1} + x\left(1 - \frac{x(\omega-\sigma^2_\varepsilon)}{1 - x^2\mathcal{I}_{2,2}}\right)\left((\mathcal{I}^*_{2,2})_2 - \frac{\omega+(\mathcal{I}^*_{1,1})_2}{\omega+(\mathcal{I}^*_{1,1})_1}(\mathcal{I}^*_{2,2})_1\right)\,. \tag{A74}$$

To establish the second conclusion, recall from Sec. A5 that the conditions $\mu_1 \leq \mu_2$ in conjunction with Assump. 4 show $(\mathcal{I}_{1,1}^*)_1 \leq (\mathcal{I}_{1,1}^*)_2$ which establishes SLOPE $\geq 1$. $\qquad\square$

The first half of Prop. 5.5, namely Eq. (19), can be obtained from Prop. A7.1 by specializing to the case $\mu_1 = \mu_\emptyset$. The second half of Prop. 5.5, namely the conditions on the slope, can be obtained from Prop. A7.1 by separately specializing to the cases $\mu_1 = \mu_\emptyset$ and $\mu_2 = \mu_\emptyset$.

## A8  Necessity of monotonicity of overlap coefficients in Def. 4.1

Here we provide an explicit description of the construction used in Sec. 5.5 and Fig. 3(b) that shows the monotonicity of overlap coefficients in Def. 4.1 is necessary in order to guarantee the ordering of errors in Prop. 5.1. The example is given by the following four LJSDs:

$$\mu_1 = \frac{1}{4}(\delta_{\lambda_1,\lambda_4} + \delta_{\lambda_2,\lambda_3} + \delta_{\lambda_3,\lambda_2} + \delta_{\lambda_4,\lambda_1}) \tag{A75}$$

$$\mu_2 = \frac{1}{4}(\delta_{\lambda_1,\lambda_4} + \delta_{\lambda_2,\lambda_3} + \delta_{\lambda_3,\lambda_1} + \delta_{\lambda_4,\lambda_2}) \tag{A76}$$

$$\mu_3 = \frac{1}{4}(\delta_{\lambda_1,\lambda_2} + \delta_{\lambda_2,\lambda_4} + \delta_{\lambda_3,\lambda_3} + \delta_{\lambda_4,\lambda_1}) \tag{A77}$$

$$\mu_4 = \frac{1}{4}(\delta_{\lambda_1,\lambda_1} + \delta_{\lambda_2,\lambda_2} + \delta_{\lambda_3,\lambda_3} + \delta_{\lambda_4,\lambda_4}), \tag{A78}$$

where $\lambda_1 = 0.6$, $\lambda_2 = 0.24$, $\lambda_3 = 0.12$ and $\lambda_4 = 0.04$. Note that $\mu_4 = \mu_\emptyset$, and these LJSDs have equal training and test covariance scales, i.e. $s = s_1 = s_2 = s_3 = s_4 = 1$. Moreover, the partial order in Def. 4.1 gives $\mu_1 \geq \mu_4$ and $\mu_2 \geq \mu_4$, and all other pairs of LJSDs are incomparable. In particular, focusing on $\mu_2$ and $\mu_3$, the ratios of overlap coefficients are

$$\frac{r_{2,1}}{r_{3,1}} = \frac{1}{6} \quad \frac{r_{2,2}}{r_{3,2}} = 3\,, \quad \frac{r_{2,3}}{r_{3,3}} = 5\,, \quad \text{and} \quad \frac{r_{2,4}}{r_{3,4}} = \frac{2}{5}\,, \tag{A79}$$

so the sequence $r_{2,i}/r_{3,i}$ is nonmonotonic in $i$. The violation of monotonicity is minimal, in the sense that only a single ratio ($r_{2,4}/r_{3,4}$) is out of order; nevertheless, the strict model-independent ordering of test errors is broken because of this nonmonotonicity, as can be seen in Fig. 3(b). For the comparable pairs of LJSDs, $\mu_1 \geq \mu_4$ and $\mu_2 \geq \mu_4$, and the strict ordering of the error is seen for all values of $\phi/\psi$. The ordering is not guaranteed for the other pairs, and indeed the ordering is not satisfied for $\mu_2$ and $\mu_3$, which is evidenced by the crossing of the corresponding curves in the figure. In this way, we see that the error can exhibit model dependence induced by nonmonotonicity of overlap coefficients in Def. 4.1, thereby showing that the monotonicity condition is necessary for Prop. 5.1. Finally, we note that even though Prop. 5.1 may no longer hold, it is nevertheless possible to develop nontrivial bounds when the strict monotonicity is violated, but we pursue this investigation elsewhere.

## A9  Proof of Thm. 5.1

As discussed in Sec. 2, we consider predictive functions $\hat{y}$ defined by random feature kernel ridge regression,

$$\hat{y}(\mathbf{x}) := YK^{-1}K_{\mathbf{x}} \tag{A80}$$

for $K := K(X,X) + \gamma I_m$, $K_{\mathbf{x}} := K(X,\mathbf{x})$, with $\gamma$ is a ridge regularization constant. Here

$$K = \frac{1}{n_1}F^\top F + \gamma I_m\,, \tag{A81}$$

and, as in the main text, we have introduced the abbreviations $F := \sigma(WX/\sqrt{n_0})$. The labels are generated by a linear function parameterized by $\beta \in \mathbb{R}^{n_0}$, whose entries are drawn independently from $\mathcal{N}(0,1)$, i.e. $Y = \beta^\top X/\sqrt{n_0} + \varepsilon$. In this section, we develop the techniques and detailed calculations needed to determine the high-dimensional asymptotic limit of the test error,

$$E_{\Sigma^*} = \mathbb{E}_{\mathbf{x},\beta}\mathbb{E}[(y(\mathbf{x}) - \hat{y}(\mathbf{x}))^2] - \sigma_\epsilon^2 = \mathbb{E}_{\mathbf{x},\beta}\mathbb{E}[(\beta^\top \mathbf{x}/\sqrt{n_0} - YK^{-1}K_{\mathbf{x}})^2]\,, \tag{A82}$$

as well as its decomposition into bias and variance terms,

$$E_{\Sigma^*} = \underbrace{\mathbb{E}_{\mathbf{x},\beta}\mathbb{E}[(\mathbb{E}[\hat{y}(\mathbf{x})] - y(\mathbf{x}))^2]}_{B_{\Sigma^*}} + \underbrace{\mathbb{E}_{\mathbf{x},\beta}[\mathbb{V}[\hat{y}(\mathbf{x})]]}_{V_{\Sigma^*}}. \tag{A83}$$

We may also write the test loss as

$$E_{\Sigma^*} = \mathbb{E}_{(\mathbf{x},y)}(y - \hat{y}(\mathbf{x}))^2 = E_1 + E_2 + E_3 \tag{A84}$$

with

$$E_1 = \mathbb{E}_{(\mathbf{x},\beta,\varepsilon)}y(\mathbf{x})^2 = \mathbb{E}_{(\mathbf{x},\beta,\varepsilon)}\mathrm{tr}(y(\mathbf{x})y(\mathbf{x})^\top) \tag{A85}$$

$$E_2 = -2\mathbb{E}_{(\mathbf{x},\beta,\varepsilon)}(K_{\mathbf{x}}^\top K^{-1}Y)y(\mathbf{x}) = -2\mathbb{E}_{(\mathbf{x},\varepsilon)}\mathrm{tr}(K_{\mathbf{x}}^\top K^{-1}Y^\top y(\mathbf{x})) \tag{A86}$$

$$E_3 = \mathbb{E}_{(\mathbf{x},\beta,\varepsilon)}(K_{\mathbf{x}}^\top K^{-1}Y^\top)^2 = \mathbb{E}_{(\mathbf{x},\varepsilon)}\mathrm{tr}(K_{\mathbf{x}}^\top K^{-1}Y^\top Y K^{-1}K_{\mathbf{x}}). \tag{A87}$$

## A9.1 Reducing to the mean-zero case

In this section, we aim to show that we may assume without loss of generality that the activation function is centered. More precisely, we define

$$\bar{F}_{ij} := F_{ij} - \mathbb{E}_Z\sigma\left(\sqrt{\bar{\mathrm{tr}}(\Sigma)}Z\right) \quad \text{and} \quad \bar{f}_i := f_i - \mathbb{E}_Z\sigma\left(\sqrt{\bar{\mathrm{tr}}(\Sigma^*)}Z\right) \tag{A88}$$

for all $i$ and $j$ and $Z \sim \mathcal{N}(0,1)$. Note that this centering operation is $n_0$-dependent, but in the limit under Assump. 4, $\bar{\mathrm{tr}}(\Sigma), \bar{\mathrm{tr}}(\Sigma^*) \to s$, which appear in the limiting self-consistent equation. Note that although we invoke Assump. 4 throughout the rest of the paper, this proof holds non-asymptotically and allows for $\bar{\mathrm{tr}}(\Sigma) \neq \bar{\mathrm{tr}}(\Sigma^*)$.

To show that replacing $F$ and $f$ by $\bar{F}$ and $\bar{f}$ does not alter the test error in the limit, we have to show that the $E_1$, $E_2$, and $E_3$ terms of Eqs. (A85)-(A87) are not changed in the limit. Clearly this is true for $E_1$ as it contains neither $F$ nor $f$. For $E_2$ we must show

$$\mathbb{E}\left[(K_{\mathbf{x}}^\top K^{-1}Y^\top - \bar{K}_{\mathbf{x}}^\top \bar{K}^{-1}Y^\top)y(\mathbf{x})\right] \to 0 \tag{A89}$$

and for $E_3$ we must show

$$\mathbb{E}\left[(K_{\mathbf{x}}^\top K^{-1}Y^\top)^2 - (\bar{K}_{\mathbf{x}}^\top \bar{K}^{-1}Y^\top)^2\right] \to 0, \tag{A90}$$

where

$$\bar{K} := \frac{1}{n_1}\bar{F}^\top \bar{F} + \gamma I \quad \text{and} \quad \bar{K}_{\mathbf{x}} := \frac{1}{n_1}\bar{F}^\top \bar{f}. \tag{A91}$$

Define the random variable

$$\Delta := (K_{\mathbf{x}}^\top K^{-1} - \bar{K}_{\mathbf{x}}\bar{K}^{-1})Y^\top. \tag{A92}$$

To control the typical behavior of $\Delta$, we will define an event $\mathcal{E}$ (see Def. A9.1), and let $\mathbf{1}_\mathcal{E}$ be an indicator for $\mathcal{E}$. In Lem. A9.7 we show $\mathbb{P}[\mathcal{E}^c] \to 0$.

We can argue

$$|\mathbb{E}[\Delta y(\mathbf{x})]| \leq |\mathbb{E}[\mathbf{1}_\mathcal{E}\Delta y(\mathbf{x})]| + |\mathbb{E}[(1 - \mathbf{1}_\mathcal{E})\Delta y(\mathbf{x})]|, \tag{A93}$$

for Eq. (A89), and

$$|\mathbb{E}[\Delta y(\mathbf{x})]| \leq |\mathbb{E}[\mathbf{1}_\mathcal{E}\Delta(K_{\mathbf{x}}^\top K^{-1}Y^\top + \bar{K}_{\mathbf{x}}^\top \bar{K}^{-1}Y^\top)]| \tag{A94}$$

$$+ |\mathbb{E}[(1 - \mathbf{1}_\mathcal{E})((K_{\mathbf{x}}^\top K^{-1}Y^\top)^2 - (\bar{K}_{\mathbf{x}}^\top \bar{K}^{-1}Y^\top)^2)]|, \tag{A95}$$

for Eq. (A90). We demonstrate Eqs. (A93) and (A94) are $o(1)$ in two steps. The first terms on the right-hand sides of Eqs. (A93) and (A94) represent the typical behavior of the random variables. To bound them, we show that, given $\mathcal{E}$, $\Delta \to 0$ in Lem. A9.8. The second terms on the right-hand sides of Eqs. (A93) and (A94) represent the atypical behavior of the random variables. To bound them, we use the Cauchy-Schwarz inequality and the fact that $\mathbb{P}[\mathcal{E}^c] \to 0$ in Lem. A9.6.

To briefly outline the structure of this section: Sec. A9.1.1 proves concentration of some quadratic forms of the underlying random matrices and bounds their operator norms with high probability; Sec. A9.1.2 controls the atypical behavior mentioned above; Sec. A9.1.3 applies the Schur complement formula to derive an expression for $\Delta$ that we can bound; Sec. A9.1.4 defines an event where the expression for $\Delta$ from the Schur complement formula can be easily bounded; and Sec. A9.1.5 completes the argument by bounding $\Delta$ on this typical event.

### A9.1.1 Prerequisites for concentration and linearization

Just as in [1], the concentration of the linear data kernel about its expectation is a key ingredient to the analysis. Define the random variables

$$\Upsilon := \frac{1}{n_0} X^\top X \quad \text{and} \quad \upsilon^* := \|\mathbf{x}\|_2^2 / n_0. \tag{A96}$$

**Remark A9.1.** *Throughout this section and the next we will use $C$ to denote an arbitrarily large, $n_0$-independent, positive constant, which can increase from line to line. For example, if $X \leq C$ and $Y \leq C$, we will simply write $XY \leq C$, since $C^2$ is still some $n_0$-independent constant. This approach is valid as such replacements only occur a finite number of times. Similarly, $c$ denotes some arbitrarily small, $n_0$-independent, positive constant that can decrease from line to line.*

Then we note

$$\mathbb{E}\Upsilon = \bar{\mathrm{tr}}(\Sigma) I_{n_0} \quad \text{and} \quad \mathbb{E}\upsilon^* = \bar{\mathrm{tr}}(\Sigma^*). \tag{A97}$$

For the variance, we see by Assump. 1 that

$$\mathbb{V}\Upsilon_{ij} = \frac{1 + \mathbb{I}(i = j)}{n_0} \bar{\mathrm{tr}}(\Sigma) \leq C/n_0 \quad \text{and} \quad \mathbb{V}\upsilon^* = \frac{2}{n_0} \bar{\mathrm{tr}}((\Sigma^*)^2) \leq C/n_0. \tag{A98}$$

This motivates the following lemma.

**Lemma A9.1.** *The event*

$$\mathcal{E}_{data} := \{\|\Upsilon\|_\infty \leq C\} \cap \left\{ |\upsilon^* - \bar{\mathrm{tr}}(\Sigma^*)| \leq C n_0^{c-1/2} \right\} \cap \bigcap_{i,j} \left\{ |\Upsilon_{ij} - \delta_{ij}\bar{\mathrm{tr}}(\Sigma)| \leq C n_0^{c-1/2} \right\} \tag{A99}$$

*occurs with high-probability, that is, for some positive, $n_0$-independent constant $C$*

$$\mathbb{P}[\mathcal{E}_{data}^{\mathsf{c}}] \leq C n_0^{-c} \tag{A100}$$

*for any positive, $n_0$-independent constant $c$.*

*Proof.* Tighter concentration than is obtained directly from the variance can be shown by observing that $\Upsilon$ and $\upsilon^*$ are equal in distribution to

$$\frac{1}{n_0} Z^T \Sigma Z \quad \text{and} \quad \frac{1}{n_0} z^\top \Sigma^* z \tag{A101}$$

for $Z$ an $(n_0, m)$-dimensional matrix and $z$ an $n_0$-dimensional vector both containing i.i.d. standard Gaussian random variables. Then using the assumption $\|\Sigma\|_\infty, \|\Sigma^*\|_\infty \leq C$ and applying the results of Sec. B of [18] (or similar concentration results), we find that for some positive constant $C$ that

$$\mathbb{P}[|\Upsilon_{ij} - \delta_{ij}\bar{\mathrm{tr}}(\Sigma)| \geq C n_0^{c-1/2}] \leq C \exp(-n_0^c), \tag{A102}$$

where $c$ is any positive constant. An identical concentration result holds for $\upsilon^* - \bar{\mathrm{tr}}(\Sigma^*)$. Then, by the union bound, we see

$$\mathbb{P}\left[ \left\{ |\upsilon^* - \bar{\mathrm{tr}}(\Sigma^*)| \geq C n_0^{-c} \right\} \cup \bigcup_{i,j} \left\{ |\Upsilon_{ij} - \delta_{ij}\bar{\mathrm{tr}}(\Sigma)| \geq C n_0^{c-1/2} \right\} \right] \tag{A103}$$

$$\leq C n_0^2 C \exp\left(-n_0^c\right) \tag{A104}$$

$$\leq C \exp(-n_0^c). \tag{A105}$$

The operator norm of $\Upsilon$ can also be bounded probabilistically. For some $n_0$-independent constant $C$,

$$\|\Upsilon\|_\infty \leq \|\Sigma\|_\infty \|Z/\sqrt{n_0}\|_\infty^2 \leq C\|\Sigma\|_\infty \left(1 + \sqrt{\frac{m}{n_0}} + \sqrt{\frac{\log(2/\delta)}{n_0}}\right) \leq C \tag{A106}$$

with probability at least $1 - \delta$, where $Z$ is i.i.d. Gaussian (see for example [65, Theorem 4.4.5]). Thus, setting $\delta = n_0^{-c}$, we see $\mathbb{P}[\|\Upsilon\|_\infty \geq C] \leq n_0^{-c}$. Applying the union bound again completes the proof. $\qquad\square$

**Lemma A9.2.** *The (squared) expected operator norm of $\Upsilon$ is also bounded:*

$$\mathbb{E}\|\Upsilon\|_\infty^2 \leq C \tag{A107}$$

*for some positive, $n_0$-independent constant $C$.*

*Proof.* This result follows from a tail integration argument. That is from [65, Theorem 4.4.5]) we have that

$$\|\Upsilon\|_\infty \leq \|\Sigma\|_\infty \|Z/\sqrt{n_0}\|_\infty^2 \leq C\|\Sigma\|_\infty \left(1 + \sqrt{\frac{m}{n_0}} + u\right) \tag{A108}$$

with probability at least $1 - 2\exp(-n_0 u^2)$, where $Z$ is i.i.d. Gaussian. Hence we have that $\mathbb{P}[\|\Upsilon\|_\infty \geq C\|\Sigma\|_\infty \left(1 + \sqrt{\frac{m}{n_0}} + u\right)] \leq 2\exp(-n_0 u^2)$. Now by the tail integration identity,

$$\mathbb{E}[\|\Upsilon\|_\infty^2] = 2\int_0^\infty u\mathbb{P}[\|\Upsilon\|_\infty \geq u]du \tag{A109}$$

$$\leq C^2\|\Sigma\|_\infty^2 \left(1 + \sqrt{\frac{m}{n_0}}\right)^2 + C^2\|\Sigma\|_\infty^2 \int_0^\infty u\mathbb{P}\left[\|\Upsilon\|_\infty \geq C\|\Sigma\|_\infty \left(1 + \sqrt{\frac{m}{n_0}} + u\right)\right]du \tag{A110}$$

$$\leq C^2\|\Sigma\|_\infty^2 \left(1 + \sqrt{\frac{m}{n_0}}\right)^2 + C^2\|\Sigma\|_\infty^2 \cdot \frac{1}{n_0} \tag{A111}$$

$$\leq C. \tag{A112}$$

by integrating the Gaussian tail. $\qquad\square$

Next, we state similar results for the matrix $\tilde{\Upsilon} := W\Sigma^* W^\top/n_0$ as were obtained for $\Upsilon$ above.

**Lemma A9.3.** *The event*

$$\mathcal{E}_W := \left\{\left\|\tilde{\Upsilon}\right\|_\infty \leq C\right\} \cap \bigcap_{i,j}\left\{\left|\tilde{\Upsilon}_{ij} - \delta_{ij}\bar{\mathrm{tr}}(\Sigma^*)\right| \leq Cn_0^{c-1/2}\right\} \tag{A113}$$

*occurs with high-probability, that is, for some positive, $n_0$-independent constant $C$*

$$\mathbb{P}[\mathcal{E}_W^c] \leq Cn_0^{-c} \tag{A114}$$

*for any positive, $n_0$-independent constant $c$.*

*Proof.* The claims follow identically to Lem. A9.1. $\qquad\square$

**Lemma A9.4.** *Moreover on the event $\mathcal{E}_W$, we have*

$$\mathbb{E}_\mathbf{x}\bar{\sigma}(W\mathbf{x}/\sqrt{n_0})_i \leq Cn_0^{c-1/2} \tag{A115}$$

*and*

$$\mathbb{E}_\mathbf{x}\bar{\sigma}(W\mathbf{x}/\sqrt{n_0})\bar{\sigma}(W\mathbf{x}/\sqrt{n_0})^\top = \zeta\tilde{\Upsilon} + (\eta - \zeta)I_m + \Delta, \tag{A116}$$

*where $\|\Delta\|_\infty \leq Cn_0^c$.*

*Proof.* Note that conditional on $W$, $W\mathbf{x}/\sqrt{n_0} \sim \mathcal{N}(0, \tilde{\Upsilon})$. Using a Taylor expansion in $\varepsilon := \tilde{\Upsilon}_{ii} - \bar{\mathrm{tr}}(\Sigma^*)$ below with Assump. 3 shows that elementwise

$$|\mathbb{E}_\mathbf{x}\bar{\sigma}(W\mathbf{x}/\sqrt{n_0})_i| = \left|\mathbb{E}_Z\bar{\sigma}\left(\sqrt{\tilde{\Upsilon}_{ii}}Z\right)\right| \tag{A117}$$

$$= \left|\mathbb{E}_Z\sigma\left(\sqrt{\varepsilon + \bar{\mathrm{tr}}(\Sigma^*)}Z\right) - \mathbb{E}_Z\sigma\left(\sqrt{\bar{\mathrm{tr}}(\Sigma^*)}Z\right)\right| \tag{A118}$$

$$\leq C\epsilon \tag{A119}$$

$$\leq Cn_0^{c-1/2}, \tag{A120}$$

where $Z$ is a standard Gaussian. See [1] for additional details, but note that we are not Taylor expanding the function $\sigma$, we are expanding the p.d.f. we integrate against. This allows us to assume

that $\sigma(W\mathbf{x})$ is centered as a conditional expectation over $\mathbf{x}$ at the expense of an elementwise $Cn_0^{c-1/2}$ error. Denoting $\bar{f} := \bar{\sigma}(W\mathbf{x}/\sqrt{n_0})$, we note this can be extended to the matrix $\mathbb{E}\bar{f}\bar{f}^\top$. We see

$$\mathbb{E}_\mathbf{x}\bar{f}\bar{f}^\top = \mathbb{E}_\mathbf{x}[(\bar{f} - \mathbb{E}_\mathbf{x}\bar{f})(\bar{f} - \mathbb{E}_\mathbf{x}\bar{f})^\top] + (\mathbb{E}_\mathbf{x}\bar{f})(\mathbb{E}_\mathbf{x}\bar{f})^\top. \tag{A121}$$

However, the second term on the right-hand side of Eq. (A121) is small is operator norm, since it is rank-1 and

$$\left\|(\mathbb{E}_\mathbf{x}\bar{f})(\mathbb{E}_\mathbf{x}\bar{f})^\top\right\|_\infty = \left\|\mathbb{E}_\mathbf{x}\bar{f}\right\|_2^2 \leq Cn_0^c \tag{A122}$$

by Eq. (A120). Next, we see for $i \neq j$ that

$$\mathbb{E}_\mathbf{x}\bar{f}_i\bar{f}_j = \mathbb{E}_{Z_1,Z_2}\bar{\sigma}(c_1 Z_1)\bar{\sigma}(c_{12}Z_1 + c_2 Z_2) \tag{A123}$$

for constants $c_1$, $c_2$, and $c_{12}$ depending on $W$ and $\Sigma^*$, where $Z_1$ and $Z_2$ are independent, standard Gaussians. See [1] for details. The $i = j$ terms can be handled similarly. Moreover, $c_{12} = \tilde{\Upsilon}_{ij}/\sqrt{\tilde{\Upsilon}_{ii}} \leq Cn_0^{c-1/2}$, so Taylor expanding in $c_{12}Z_1$, we find

$$\mathbb{E}_\mathbf{x}\bar{f}\bar{f}^\top = \zeta\Upsilon + (\eta - \zeta)I + \tilde{\Delta} + (\mathbb{E}_\mathbf{x}\bar{f})(\mathbb{E}_\mathbf{x}\bar{f})^\top. \tag{A124}$$

As a consequence of the Taylor expansion and Assump. 3, $\tilde{\Delta}$ has entries that are bounded by $Cn_0^{c-1}$ in absolute value. Again, see [1]. The final conclusion for $\Delta = \tilde{\Delta} + (\mathbb{E}_\mathbf{x}\bar{f})(\mathbb{E}_\mathbf{x}\bar{f})^\top$ follows by upper bounding $\|\tilde{\Delta}\|_\infty \leq \|\tilde{\Delta}\|_F \leq Cn_0^c$ and using the previous bound in Eq. (A122). $\square$

Finally we include a bound on the operator norm of the random feature matrix. The argument follows [41, Lemma C.3] closely.

**Lemma A9.5.** *Under Assump. 3, the event*

$$\mathcal{E}_F := \left\{\left\|\bar{F}/\sqrt{n_0}\right\|_\infty \leq Cn_0^c\right\} \tag{A125}$$

*occurs with high-probability, that is, for some positive, $n_0$-independent constant $C$*

$$\mathbb{P}[\mathcal{E}_F^\mathsf{c}] \leq Cn_0^{-c} \tag{A126}$$

*for any positive, $n_0$-independent constant $c < 10$.*

*Proof.* Consider the matrix

$$\bar{R}_{ij} = 1_{i \neq j}\bar{\sigma}(z_i^\top\Sigma^{1/2}z_j/\sqrt{n_1})/\sqrt{n_1} \tag{A127}$$

for $z_i = \Sigma^{-1/2}X_i$ (i.e. columns of $\Sigma^{-1/2}X$) for $1 \leq i \leq m$ and $z_{m+i} = W_i$ (i.e. rows of $W$) for $1 \leq i \leq n_0$. By construction, we note that $\bar{F}/\sqrt{n_1}$ is a minor of $\bar{R}$. Thus, bounding the operator norm of $\bar{R}$ suffices to bound the operator norm of $\bar{F}$. Moreover, $z_i$ are independent Gaussians distributed as $\mathcal{N}(0, I_{n_0})$.

Next we show that the entries to the activation function cannot be too large with high probability. For convenience throughout we define $M = m + n_0$ and let $\Omega$ be a symmetric matrix with $\|\Omega\|_\infty \leq C$. Note that, for $i \neq j$, the random variable $z_i^\top\Omega z_j/\sqrt{n_0} = \sum_{k=1}^{n_0}\lambda_k(\Omega)a_k b_k/\sqrt{n_0}$ where $a_k$ and $b_k$ are i.i.d. from $\sim \mathcal{N}(0, 1)$. So the moment generating function can be bounded as,

$$\mathbb{E}\left[\exp\left(\sum_{i=1}^{n_0} t\lambda_i(\Omega)a_i b_i/\sqrt{n_0}\right)\right] = \prod_{i=1}^{n_0}\mathbb{E}[\exp(a_i \cdot t\lambda_i(\Omega)b_i/\sqrt{n_0})] \tag{A128}$$

$$= \prod_{i=1}^{n_0}\mathbb{E}[\exp(t^2 b_i^2\lambda_i(\Omega)^2/(2n_0))] \tag{A129}$$

$$\leq \prod_{i=1}^{n_0}\mathbb{E}[\exp(t^2 b_i^2 C^2/(2n_0))] \tag{A130}$$

$$\leq \mathbb{E}[\exp(t^2 b_i^2 C^2/2)] \tag{A131}$$

$$= \frac{1}{\sqrt{1 - C^2 t^2}} \tag{A132}$$

for $|t| \leq \frac{1}{C}$. Further $\frac{1}{\sqrt{1-C^2 t^2}} \leq \exp(C^2 t^2)$ for $|t| \leq \frac{1}{2C}$. This establishes the original random variable is subexponential. Thus this random variables satisfies

$$\mathbb{P}[|z_i^\top \Omega z_j / \sqrt{n_0}| \geq u] \leq 2 \max\{\exp(-u^2/2C^2), \exp(-u/2C)\} \tag{A133}$$

(see for example [66, Proposition 2.9]). Define the event

$$\mathcal{G} := \bigcap_{1 \leq i < j \leq M} \left\{ \left| \frac{z_i^\top \Sigma^{1/2} z_j}{\sqrt{n_0}} \right| \geq C\sqrt{\log M} \right\} \cap \bigcap_{1 \leq i < j \leq M} \left\{ \left| \frac{z_i^\top \Sigma z_j}{\sqrt{n_0}} \right| \geq C^2 \sqrt{\log M} \right\}. \tag{A134}$$

Then by a union bound and Markov's inequality, we have that

$$\mathbb{P}[\mathcal{G}^c] \leq 4/M^{20} \tag{A135}$$

for large enough $C$.

We now define a modified version of $\bar{\sigma}$, that is the same up to a constant factor on $\mathcal{G}$ but is truncated outside of $\mathcal{G}$. Let $\bar{u} = C\sqrt{\log M}$ and taking the constants from Assump. 3, we see

$$\tilde{\sigma}(u) := \begin{cases} \bar{\sigma}(u) \exp(-c_1|\bar{u}|)/c_0 & \text{for } |u| \leq \bar{u} \\ \bar{\sigma}(\bar{u}) \exp(-c_1|\bar{u}|)/c_0 & \text{for } u > \bar{u} \\ \bar{\sigma}(-\bar{u}) \exp(-c_1|\bar{u}|)/c_0 & \text{for } u < -\bar{u} \end{cases}. \tag{A136}$$

Just as we did with $\sigma$ in Eq. (A88), we center $\tilde{\sigma}$ with its mean $\tilde{a}$. Note $\tilde{\sigma}$ is a 1-bounded and 1-Lipschitz function so $|\tilde{a}| \leq 1$.

Now consider the matrix

$$\tilde{R}_{ij} := 1_{i \neq j}(\tilde{\sigma}(z_i^\top \Sigma^{1/2} z_j / \sqrt{n_0}) - \tilde{a})/\sqrt{n_1}. \tag{A137}$$

By controlling the operator norm of $\tilde{R}$, we can control the operator norm of $\bar{R}$ at the end of the proof. By a covering argument it suffices to control

$$\|\tilde{R}\|_\infty \leq \max_{v \in S} 10 \underbrace{|v^\top \tilde{R} v|}_{F_v(Z)}, \tag{A138}$$

where $S$ is a $1/4$-covering of the $M$-dimensional sphere with cardinality $\exp(cM)$ and the matrix $Z$ is of all the variables $z_i$.

We now seek to apply [13, Lemma 9, Lemma 20]. To this end we wish to show that $F_v(Z)$ is Lipschitz in $Z$, that is, if we define $\mathcal{Z} := \sqrt{t}Z + \sqrt{1-t}Z'$ for $(Z, Z') \in \mathcal{G} \times \mathcal{G}$, we need to show that

$$\max_{v \in S} \max_{t \in [0,1]} \left\| \nabla F_v(\sqrt{t}Z + \sqrt{1-t}Z') \right\|_F \leq L \tag{A139}$$

for suitable $L$. Consider the gradient with respect to a column of $\mathcal{Z}$ (denoted by $\zeta_l$):

$$\nabla_{\zeta_l} F_v(\mathcal{Z}) = 2\frac{v_l}{\sqrt{n_0 n_1}} \sum_{i \neq l} \Sigma^{1/2} \zeta_i \underbrace{v_i \tilde{\sigma}'(\zeta_i^\top \Sigma^{1/2} \zeta_l^\top / \sqrt{n_0})}_{\xi_i} \tag{A140}$$

$$= 2\frac{v_l}{\sqrt{n_0 n_1}} \Sigma^{1/2} \mathcal{Z} \xi, \tag{A141}$$

where $\xi$ is the vector with coordinates $\xi_i$ except at $l$ where it is zero. Continuing, on the set $\mathcal{G}$, we find

$$\|\nabla_{w_l} F_v(W)\|_2^2 \leq C^2 v_l^2/n_0^2 \sum_{i \neq l, j \neq l} \left| \xi_i \xi_j w_i^\top \Sigma w_j \right| \tag{A142}$$

$$\leq C v_l^2 \sqrt{\log M}/n_0^2 \sum_i \sum_{j \neq i} |\xi_i \xi_j| \tag{A143}$$

$$\leq C v_l^2 \sqrt{\log M}/n_0^2 \sum_i \sum_{j \neq i} (\xi_i^2 + \xi_j^2) \tag{A144}$$

$$\leq C v_l^2 \sqrt{\log M}/n_0 \tag{A145}$$

where the last line follows since $|\tilde{\sigma}'(\cdot)|_2 \leq 1$ implies that $\|\xi\| \leq 1$. So finally, we obtain

$$\|\nabla_W F_v(W)\|_F^2 \leq C\sqrt{\log M}/n_0 =: L^2. \tag{A146}$$

Now [13, Lemma 9] shows for constant a $D$ that

$$\mathbb{P}[|v^\top \tilde{R}v| > D] \leq C\exp\left(Cn_0 - \frac{D^2}{L^2}\right) + \frac{C}{D^2}\mathbb{E}[\max_{v \in S}(F_v(Z) - F_v(Z'))^2 \cdot \mathbf{1}_{\mathcal{G}^c}]. \tag{A147}$$

We use a crude bound on the complement. First note that

$$\max_{v \in S}(F_v(Z) - F_v(Z'))^2 \leq C\|\tilde{\sigma}(Z^\top \Sigma^{1/2}Z/\sqrt{n_0})\|_F^2 \leq C\|Z^\top Z/\sqrt{n_0}\|_F^2 + Cn_0^2, \tag{A148}$$

since removing the 1-bounded-Lipschitz $\tilde{\sigma}(\cdot)$ can be done by centering each activation with $\tilde{\sigma}(0)$. Then,

$$\mathbb{E}\left[\max_{v \in S}(F_v(Z) - F_v(Z'))^2 \cdot \mathbf{1}_{\mathcal{G}^c}\right] \leq \sqrt{\frac{C}{n_1}(\mathbb{E}[\|Z^\top \Sigma^{1/2}Z/\sqrt{n_0}\|_F^4] + n_0^4)\mathbb{P}[\mathcal{G}^c]}. \tag{A149}$$

A short computation shows that $\mathbb{E}[\|Z^\top \Sigma^{1/2}Z/\sqrt{n_0}\|_F^4] \leq Cn_0^8$. Combining all terms then shows that

$$\mathbb{P}[|v^\top \tilde{R}v| > D] \leq C\exp(Cn_0) \cdot \exp\left(-\frac{n_0 D^2}{\log M}\right) + \frac{C}{D^2} \cdot \frac{C}{M^5}. \tag{A150}$$

Choosing $D = c_3\sqrt{\log M}$ for sufficiently large $c_3$ shows that

$$\mathbb{P}[|v^\top \tilde{R}v| > D] \leq C\exp(-cn_0) + \frac{C}{n_0^{10}} \leq \frac{C}{n_0^{10}}, \tag{A151}$$

where $c > 0$.

Recalling the original $\epsilon$-net covering, this implies that

$$\|\tilde{R}\|_\infty \leq C\sqrt{\log n_0} \tag{A152}$$

with probability at least $C/n^{10}$. We can now finish the argument by relating the operator norm of $\bar{R}$ and $\tilde{R}$. Recalling the rescaling between the modified and unmodified activations, we see

$$\mathbb{P}[\|\bar{R}\|_\infty \geq C\sqrt{\log n_0} \cdot c_0\exp(c_1|\bar{u}|)] \leq \mathbb{P}[\|\tilde{R}\|_\infty \geq C\sqrt{\log(n_0)}, \mathcal{G}] + \mathbb{P}[\mathcal{G}^c] \leq C/n_0^{10}. \tag{A153}$$

$\square$

### A9.1.2 Controlling the atypical behavior

Since the argument for the atypical event is the most straightforward, we provide that first.

**Lemma A9.6.** *Suppose* $\mathbb{P}[\mathcal{E}^c] = Cn_0^{-c}$ *for some* $n_0$-*independent constants* $C > 0$ *and* $c > 0$. *Then,*

$$|\mathbb{E}\left[(1 - \mathbf{1}_{\mathcal{E}})\Delta y(\mathbf{x})\right]| \to 0 \tag{A154}$$

*and*

$$\left|\mathbb{E}\left[(1 - \mathbf{1}_{\mathcal{E}})\left((K_{\mathbf{x}}^\top K^{-1}Y^\top)^2 - (\bar{K}_{\mathbf{x}}^\top \bar{K}^{-1}Y^\top)^2\right)\right]\right| \to 0 \tag{A155}$$

*as* $n_0 \to \infty$.

*Proof.* Applying the Cauchy-Schwarz inequality to the left-hand side of Eq. (A154), we may bound it by

$$\mathbb{E}\left|[(1 - \mathbf{1}_{\mathcal{E}})\Delta y(\mathbf{x})]\right| \leq (\mathbb{E}\left[\mathbf{1}_{\mathcal{E}^c}\right])^{1/2}\left(\mathbb{E}\left|\Delta y\right|^2\right)^{1/2}. \tag{A156}$$

Since the first term $\mathbb{P}[\mathcal{E}_c] \to 0$ it also follows that $(\mathbb{P}[\mathcal{E}_c])^{1/2} \to 0$. So it suffices to show $\mathbb{E}[|\Delta y|^2] = O(1)$.

We now apply the Cauchy-Schwarz inequality again to bound the second term by

$$\mathbb{E}\left|(\Delta y(\mathbf{x}))^2\right| \leq \sqrt{\mathbb{E}[\Delta^4]\mathbb{E}[y(\mathbf{x})^4]}. \tag{A157}$$

A simple argument shows that $\mathbb{E}[y(\mathbf{x})^4] \leq C(\mathbb{E}[(\beta^\top \mathbf{x})^4/n_0^2] + \mathbb{E}[\epsilon^4]) \leq C\mathbb{E}[\|\Sigma^{1/2}\mathbf{z}\|^4/n_0^2] + C\sigma_\epsilon^4 \leq C$ for $\mathbf{z} \sim \mathcal{N}(0, I_d)$ since $\mathbb{E}[\|\mathbf{z}\|_2^4]/n_0^2 \leq C$. Hence it suffices to show $\mathbb{E}[\Delta^4] \leq C$. To this end note that

$$\mathbb{E}[\Delta^4] \leq C(\mathbb{E}[(K_\mathbf{x}^\top K^{-1} Y^\top)^4] + \mathbb{E}[(\bar{K}_\mathbf{x}^\top \bar{K}^{-1} Y^\top)^4]) \tag{A158}$$

An identical argument can be applied to bound both terms. We outline the argument for the first term. We see

$$|\mathbb{E}[(K_\mathbf{x}^\top K^{-1} Y^\top)^4]| \leq \mathbb{E}|\mathbb{E}_{\beta,\varepsilon}[(K_\mathbf{x}^\top K^{-1} Y^\top)^4 | W, X, x]|. \tag{A159}$$

Now note that by the definition of $Y = \beta^\top X/\sqrt{n_0} + \varepsilon$ we have that,

$$(K_\mathbf{x}^\top K^{-1} Y^\top)^4 \leq C\left((K_\mathbf{x}^\top K^{-1} \frac{X}{\sqrt{n_0}}\beta)^4 + (K_\mathbf{x}^\top K^{-1}\varepsilon)^4\right) \tag{A160}$$

Recalling that $\beta \sim \mathcal{N}(0, I_{n_0})$ and $\varepsilon \sim \mathcal{N}(0, I_m)$, we can compute the Gaussian moments (marginally in $\beta, \varepsilon$) as,

$$\mathbb{E}_{\beta,\varepsilon}[(K_\mathbf{x}^\top K^{-1} \frac{X}{\sqrt{n_0}}\beta)^4 + (K_\mathbf{x}^\top K^{-1}\varepsilon)^4 | W, X, x] \tag{A161}$$

$$\leq C\left((K_\mathbf{x}^\top K^{-1} \left(\frac{X^\top X}{n_0}\right) K^{-1} K_\mathbf{x}^\top)^2 + \sigma_\epsilon^2(K_\mathbf{x}^\top K^{-2} K_\mathbf{x}^\top)^2\right) \tag{A162}$$

Continuing, using the definition of $K_\mathbf{x}$,

$$\mathbb{E}\left[(K_\mathbf{x}^\top K^{-1} \left(\frac{X^\top X}{n_0}\right) K^{-1} K_\mathbf{x}^\top)^2\right] \leq \mathbb{E}\left[\left\|\frac{f^\top}{\sqrt{n_0}}\frac{FK^{-1}}{\sqrt{n_0}}\frac{X^\top X}{n_0}\frac{K^{-1}F^\top}{\sqrt{n_0}}\frac{f}{\sqrt{n_0}}\right\|_\infty^2\right] \tag{A163}$$

$$\leq \mathbb{E}\left[\left\|\frac{f}{\sqrt{n_0}}\right\|_2^4 \cdot \left\|\frac{FK^{-1}}{\sqrt{n_0}}\right\|_\infty^4 \cdot \left\|\frac{X^\top X}{n_0}\right\|_\infty^2\right]. \tag{A164}$$

Noting that $K = FF^\top/n_0 + \gamma I_m$ so an application of the SVD to $\frac{F}{\sqrt{n_0}}$ (denoting the corresponding singular values as $s_i$) shows that $\|\frac{FK^{-1}}{\sqrt{n_0}}\|_\infty = \max_i \frac{s_i}{s_i^2+\gamma} \leq \max_{s\geq 0} \frac{s}{s^2+\gamma} = 1/(2\sqrt{\gamma})$. Hence, we can continue bounding Eq. (A164) by

$$C\mathbb{E}[\|f/\sqrt{n_0}\|_\infty^4] \cdot \mathbb{E}[\|X^\top X/n_0\|_\infty^2], \tag{A165}$$

where we exploited the independence of $f$ and $X$. Lem. A9.2 ensures that $\mathbb{E}[\|X^\top X/n_0\|_\infty^2] = \mathbb{E}[\|\Upsilon\|_\infty^2] \leq C$. The former term becomes $\mathbb{E}[\|f/\sqrt{n_0}\|_2^4] = \mathbb{E}[\sigma(\sqrt{\mathrm{tr}(\Sigma^*)}z)^4] \leq C$ for $z \sim \mathcal{N}(0,1)$ by Assump. 3. Thus, the previous displays are bounded by some constant $C$. An entirely analogous argument shows that $\mathbb{E}[(K_\mathbf{x}^\top K^{-2} K_\mathbf{x}^\top)^2] \leq C$. Together these two results along with the previous computations establish that,

$$|\mathbb{E}[(K_\mathbf{x}^\top K^{-1} Y^\top)^4]| \leq C \tag{A166}$$

Note that our argument did not exploit any explicit properties of the centered vs uncentered activation function $\sigma$ vs. $\bar{\sigma}$. Hence an identical argument shows that, $|\mathbb{E}[(\bar{K}_\mathbf{x}^\top \bar{K}^{-1} Y^\top)^4]| \leq C$. These two results imply $\mathbb{E}[\Delta^4] \leq C$ as desired.

Now we turn to the second term. Using an identical application of the Cauchy-Schwarz inequality as used for the first term, we can bound Eq. (A155) by

$$(\mathbb{E}[\mathbf{1}_{\mathcal{E}^c}])^{1/2}\left(\mathbb{E}\left|(K_\mathbf{x}^\top K^{-1} Y^\top)^2 - (\bar{K}_\mathbf{x}^\top \bar{K}^{-1} Y^\top)^2\right|^2\right)^{1/2}. \tag{A167}$$

Hence we can show Eq. (A155) is $o(1)$ if $\mathbb{E}\left|(K_\mathbf{x}^\top K^{-1} Y^\top)^2 - (\bar{K}_\mathbf{x}^\top \bar{K}^{-1} Y^\top)^2\right|^2 = O(1)$. This result follows immediately from the computations shown for the previous term since we have already established that $\mathbb{E}[(K_\mathbf{x}^\top K^{-1} Y^\top)^4] \leq C$ and $\mathbb{E}[(\bar{K}_\mathbf{x}^\top \bar{K}^{-1} Y^\top)^4] \leq C$. These two previously shown results imply

$$\mathbb{E}\left|(K_\mathbf{x}^\top K^{-1} Y^\top)^2 - (\bar{K}_\mathbf{x}^\top \bar{K}^{-1} Y^\top)^2\right|^2 \leq C\left(\mathbb{E}[(K_\mathbf{x}^\top K^{-1} Y^\top)^4] + \mathbb{E}[(\bar{K}_\mathbf{x}^\top \bar{K}^{-1} Y^\top)^4]\right) \leq C. \tag{A168}$$

$\square$

### A9.1.3 Schur complement formula for bounding $\Delta$

The centering procedure is a rank-1 change to $F$ and $f$, so applying the Schur complement formula to understand its effect is natural. Write $a := \mathbb{E}_Z \sigma\left(\sqrt{\mathrm{tr}(\Sigma)}Z\right)$, $a^* := \mathbb{E}_Z \sigma\left(\sqrt{\mathrm{tr}(\Sigma^*)}Z\right)$, $\mathbf{v} := 1/n_1 \bar{F}^\top 1_{n_1}$, $\mathbf{u} := a 1_m$, $U := [\mathbf{u}, \mathbf{v}]^\top$, and $C := \left(\begin{smallmatrix} 1 & 1 \\ 1 & 0 \end{smallmatrix}\right)$. Then,

$$K^{-1} = (\bar{K} + \mathbf{u}\mathbf{v}^\top + \mathbf{v}\mathbf{u}^\top + \mathbf{u}\mathbf{u}^\top)^{-1} \tag{A169}$$

$$= (\bar{K} + U^\top C U)^{-1} \tag{A170}$$

$$= \bar{K}^{-1} - \bar{K}^{-1} U^\top (C^{-1} + U \bar{K}^{-1} U^\top)^{-1} U \bar{K}^{-1}, \tag{A171}$$

and, for $\delta := 1/n_1 \bar{f}^\top 1_{n_1}$ and $P := (\delta + a^*, a^*)^\top$,

$$K_{\mathbf{x}} = \bar{K}_{\mathbf{x}} + \frac{1}{n_1}(F - \bar{F})^\top \bar{f} + \frac{1}{n_1} \bar{F}^\top (f - \bar{f}) + \frac{1}{n_1}(F - \bar{F})^\top (f - \bar{f}) \tag{A172}$$

$$= \bar{K}_{\mathbf{x}} + (\delta + a^*)\mathbf{u} + a^* \mathbf{v} \tag{A173}$$

$$= \bar{K}_{\mathbf{x}} + U^\top P. \tag{A174}$$

Combining these expressions,

$$K^{-1} K_{\mathbf{x}} = K^{-1}(\bar{K}_{\mathbf{x}} + U^\top P) \tag{A175}$$

$$= \bar{K}^{-1}(\bar{K}_{\mathbf{x}} + U^\top P) - \bar{K}^{-1} U^\top (C^{-1} + U \bar{K}^{-1} U^\top)^{-1} U \bar{K}^{-1}(\bar{K}_{\mathbf{x}} + U^\top P) \tag{A176}$$

$$= \bar{K}^{-1} \bar{K}_{\mathbf{x}} + T_1 + T_2, \tag{A177}$$

with

$$T_1 = \bar{K}^{-1} U^\top \left(I_2 - (C^{-1} + U \bar{K}^{-1} U^\top)^{-1} U \bar{K}^{-1} U^\top\right) P \tag{A178}$$

$$= \bar{K}^{-1} U^\top \left(I_2 + C U \bar{K}^{-1} U^\top\right)^{-1} P \tag{A179}$$

$$T_2 = -\bar{K}^{-1} U^\top (C^{-1} + U \bar{K}^{-1} U^\top)^{-1} U \bar{K}^{-1} \bar{K}_{\mathbf{x}} \tag{A180}$$

$$= -\bar{K}^{-1} U^\top \left(I_2 + C U \bar{K}^{-1} U^\top\right)^{-1} C U \bar{K}^{-1} \bar{K}_{\mathbf{x}}. \tag{A181}$$

Furthermore, writing $c_0 := \frac{1}{m}\mathbf{u}^\top \bar{K}^{-1} \mathbf{u}$, $c_1 := 1 + \mathbf{u}^\top \bar{K}^{-1} \mathbf{v}$, $c_2 := 1 - \mathbf{v}^\top \bar{K}^{-1} \mathbf{v}$,

$$\left(I_2 + C U \bar{K}^{-1} U^\top\right)^{-1} = \frac{1}{c_1^2 + m c_0 c_2} \begin{pmatrix} c_1 & c_2 - c_1 \\ -m c_0 & m c_0 + c_1 \end{pmatrix}, \tag{A182}$$

so that,

$$\left(I_2 + C U \bar{K}^{-1} U^\top\right)^{-1} P = \frac{1}{c_1^2 + m c_0 c_2} \begin{pmatrix} c_1 \delta + c_2 a^* \\ -m c_0 \delta + c_1 a^* \end{pmatrix}, \tag{A183}$$

and,

$$\left(I_2 + C U \bar{K}^{-1} U^\top\right)^{-1} C = \frac{1}{c_1^2 + m c_0 c_2} \begin{pmatrix} c_2 & c_1 \\ c_1 & -m c_0 \end{pmatrix}. \tag{A184}$$

### A9.1.4 Concentration with high-probability

**Definition A9.1.** *Recall the definitions from Sec. A9.1.3. To characterize the typical behavior of the random variables, we define the event*

$$\mathcal{E} := \mathcal{E}_{data} \cap \mathcal{E}_W \cap \mathcal{E}_F \cap \left\{|\delta| \le C n_0^{c-1/2}\right\} \cap \left\{c n_0^{-c} \le c_0 \le C n_0^c\right\} \cap \left\{c_1 \le C n_0^{c+1/2}\right\} \tag{A185}$$

$$\cap \left\{c n_0^{-c} \le c_2 \le C n_0^c\right\} \cap \left\{Y \bar{K}^{-1} \mathbf{v}^\top \le C n_0^c\right\} \cap \left\{Y \bar{K}^{-1} \mathbf{u}^\top \le C n_0^{c+1/2}\right\}. \tag{A186}$$

$$\cap \left\{\bar{K}_{\mathbf{x}} \bar{K}^{-1} \mathbf{u}^\top \le C n_0^c\right\} \cap \left\{\bar{K}_{\mathbf{x}} \bar{K}^{-1} \mathbf{v}^\top \le C n_0^{c-1/2}\right\} \cap \{y(\mathbf{x}) \le C n_0^c\} \tag{A187}$$

**Lemma A9.7.** *The event $\mathcal{E}$ is high-probability, that is, $\mathbb{P}[\mathcal{E}^c] \le C n_0^{-c}$ for some constant $C > 0$ and any constant $0 < c < 10$.*

*Proof.* First, we already know $\mathcal{E}_{\text{data}}$, $\mathcal{E}_W$, and $\mathcal{E}_F$ all occur with high-probability (see Lems. A9.1, A9.3 and A9.5). When bounding the other events in Eqs. (A185)-(A187), it will be useful to introduce conditioning on one of $\mathcal{E}_{\text{data}}$, $\mathcal{E}_W$, or $\mathcal{E}_F$. This will suffice since for any event $\mathcal{A}$, we have

$$\mathbb{P}[\mathcal{A}^{\text{c}}] \leq \mathbb{P}[\mathcal{A}^{\text{c}}|\mathcal{E}_{\text{data}}] + \mathbb{P}[\mathcal{E}_{\text{data}}^{\text{c}}] \leq \mathbb{P}[\mathcal{A}^{\text{c}}|\mathcal{E}_{\text{data}}] + Cn_0^{-c} \tag{A188}$$

for example. We will explicitly denote this conditioning, but when taking expectation over some random variables (e.g. $x$ or $W$) we will not explicitly denote that we are conditioning on the remaining independent random variables (e.g. $X$). Once we have bounded the probability of the complement of each event in Eqs. (A185)-(A187), we can apply the union bound to complete the proof.

For future reference recall $\mathbf{v} := 1/n_1 \bar{F}^\top 1_{n_1}$ and $\mathbf{u} := a1_m$. We deal with each event in Eqs. (A185)-(A187) in turn.

**Controlling $\delta$:** Here, we introduce conditioning on the event $\mathcal{E}_{\text{data}}$. To control $\delta$, we want to calculate its mean and variance to apply Chebyshev's inequality. Note that $\mathcal{E}_{\text{data}}$ is independent of $W$, so the expectations over $W$ below are unchanged by this conditioning. Recall

$$\delta = \frac{1}{n_1} \sum_{k=1}^{n_1} \sigma \left( \sum_{j=1}^{n_0} W_{kj} x_j / \sqrt{n_0} \right) - a^* \tag{A189}$$

and note that conditional on $x$ the sum $\sum_j W_{kj} x_j / \sqrt{n_0}$ is distributed as $\mathcal{N}(0, \upsilon^*)$ for all $k$.

Expanding in $\upsilon^* - \bar{\text{tr}}(\Sigma^*)$, we see

$$\mathbb{E}_W[\delta|\mathcal{E}_{\text{data}}] = \mathbb{E}[\mathbb{E}_{Z \sim \mathcal{N}(0,1)}[\sigma(\sqrt{\bar{\text{tr}}(\Sigma^*) + (\upsilon^* - \bar{\text{tr}}(\Sigma^*))}Z)] - a^*|\mathcal{E}_{\text{data}}] \tag{A190}$$

$$= \mathbb{E}[\mathbb{E}_Z[\sigma(\sqrt{\bar{\text{tr}}(\Sigma^*)}Z)] - a^*|\mathcal{E}_{\text{data}}] \tag{A191}$$

$$+ \mathbb{E}\left[ \frac{1}{2} \frac{(\upsilon^* - \bar{\text{tr}}(\Sigma^*))}{\bar{\text{tr}}(\Sigma^*)} \mathbb{E}_Z[\sqrt{\bar{\text{tr}}(\Sigma^*)}Z\sigma(\sqrt{\bar{\text{tr}}(\Sigma^*)}Z)] + \Delta \Big| \mathcal{E}_{\text{data}} \right] \tag{A192}$$

$$= \mathbb{E}\left[ \frac{1}{2} \frac{(\upsilon^* - \bar{\text{tr}}(\Sigma^*))}{\bar{\text{tr}}(\Sigma^*)} \mathbb{E}_Z[\sqrt{\bar{\text{tr}}(\Sigma^*)}Z\sigma(\sqrt{\bar{\text{tr}}(\Sigma^*)}Z)] + \Delta|\mathcal{E}_{\text{data}} \right], \tag{A193}$$

where $Z$ is a standard Gaussian. The first term of Eq. (A193) is less than $Cn_0^{c-1/2}$ and the remainder term $\Delta$ is less than $Cn_0^{c-1}$ on $\mathcal{E}_{\text{data}}$. More detail on this argument can be found in [1]. Taking expectation over the remaining randomness, we see $\mathbb{E}[\delta|\mathcal{E}_{\text{data}}] \leq Cn_0^{c-1/2}$.

Similarly,

$$\mathbb{V}_W[\delta|\mathcal{E}_{\text{data}}] = \frac{1}{n_1^2} \sum_{k=1}^{n_1} \mathbb{V}_W \left[ \sigma \left( \sum_{j=1}^{n_0} W_{kj} x_j / \sqrt{n_0} \right) \Big| \mathcal{E}_{\text{data}} \right] = \frac{1}{n_1} \mathbb{V}_{Z \sim \mathcal{N}(0,\upsilon^*)} \sigma(Z), \tag{A194}$$

where we can again Taylor expand in $\upsilon^* - \bar{\text{tr}}(\Sigma^*)$ to get the bound $\mathbb{V}_W[\delta|\mathcal{E}_{\text{data}}] \leq C/n_0$ on $\mathcal{E}_{\text{data}}$. Using the law of total variance, we can then bound $\mathbb{V}[\delta|\mathcal{E}_{\text{data}}] \leq C/n_0$.

Finally applying Chebyshev's inequality implies for all $c > 0$ that

$$\mathbb{P}[|\delta - \mathbb{E}[\delta|\mathcal{E}_{\text{data}}]| > Cn_0^{c-1/2}|\mathcal{E}_{\text{data}}] \leq Cn_0^{-c}, \tag{A195}$$

and so

$$\mathbb{P}[|\delta| > Cn_0^{c-1/2}|\mathcal{E}_{\text{data}}] \leq Cn_0^{-c}. \tag{A196}$$

**Controlling $c_0$, $c_1$, and $c_2$:** Here, we introduce conditioning on the event $\mathcal{E}_F$. The results for $c_0$, $c_1$, and $c_2$ can be found using a similar argument to terms in [41, Lemma 9.6/Step 3]. The identifications $c_0 \leftrightarrow K_{11}$, $c_1 \leftrightarrow K_{12}$, $c_2 \leftrightarrow 1 - K_{22}$ hold.

For $c_0 = \frac{1}{m} \mathbf{u}^\top \bar{K}^{-1} \mathbf{u}$ we have that

$$c_0 \leq \frac{a^2 \|1_m\|^2}{m} \|\bar{K}^{-1}\|_\infty \leq \frac{C}{\gamma} \tag{A197}$$

and

$$c_0 \geq \frac{a^2 \|1_m\|^2}{m} \lambda_{\min}(\bar{K}^{-1}) \geq \frac{C}{(\gamma + \|\bar{F}\|_\infty^2/n_0)} \geq c n_0^{-c} \tag{A198}$$

using the operator norm bound on the event $\mathcal{E}_F$.

For $c_1 = 1 + \mathbf{u}^\top \bar{K}^{-1} \mathbf{v}$ we have

$$|c_1| = \left| 1 + a 1_m^\top \bar{K}^{-1} \frac{\bar{F}^\top}{\sqrt{n_1}} \frac{1_{n_1}}{\sqrt{n_1}} \right| \leq 1 + a \|1_m\|_2 \frac{\|1_{n_1}\|_2}{\sqrt{n_1}} \|\bar{K}^{-1} \frac{\bar{F}}{\sqrt{n_1}}\| \leq C n_0^{1/2}, \tag{A199}$$

where the bound $\|\bar{K}^{-1} \frac{\bar{F}}{\sqrt{n_1}}\| \leq C$ follows by considering the SVD of $\bar{F}$.

For $c_2 = 1 - \mathbf{v}^\top \bar{K}^{-1} \mathbf{v}$ we obviously have that $c_2 \leq 1$ since $\mathbf{v}^\top \bar{K}^{-1} \mathbf{v} \geq 0$. Additionally, we have that

$$c_2 = 1 - \frac{1}{n_1} 1_{n_1}^\top \frac{\bar{F}}{\sqrt{n_1}} \bar{K}^{-1} \frac{\bar{F}^\top}{\sqrt{n_1}} 1_{n_1} \tag{A200}$$

$$= \frac{1_{n_1}^\top}{\sqrt{n_1}} \left( I_{n_1} - \frac{\bar{F}}{\sqrt{n_1}} \bar{K}^{-1} \frac{\bar{F}^\top}{\sqrt{n_1}} \right) \frac{1_{n_1}}{\sqrt{n_1}} \tag{A201}$$

$$\geq 1 - \| \frac{\bar{F}}{\sqrt{n_1}} \bar{K}^{-1} \frac{\bar{F}^\top}{\sqrt{n_1}} \|_\infty \tag{A202}$$

$$= 1 - \frac{\|\bar{F}/\sqrt{n_1}\|_\infty^2}{\gamma + \|\bar{F}/\sqrt{n_1}\|_\infty^2} \geq 1 - \frac{c n_0^c}{\gamma + c n_0^c} \tag{A203}$$

$$\geq \frac{C\gamma}{c n_0^c} \tag{A204}$$

$$\geq c n_0^{-c} \tag{A205}$$

as desired once again using the conditioning on the event $\mathcal{E}_F$.

Finally,

$$\mathbf{v}^\top \bar{K}^{-1} \mathbf{v} = \frac{1}{n_1} 1_{n_1}^\top \frac{\bar{F}}{\sqrt{n_1}} \bar{K}^{-1} \frac{\bar{F}^\top}{\sqrt{n_1}} 1_{n_1} \leq \frac{\|1_{n_1}\|_2^2}{n_1} \left\| \frac{\bar{F}}{\sqrt{n_1}} \bar{K}^{-1} \frac{\bar{F}^\top}{\sqrt{n_1}} \right\|_\infty \leq C \tag{A206}$$

since $\frac{\bar{F}}{\sqrt{n_1}} \bar{K}^{-1} \frac{\bar{F}^\top}{\sqrt{n_1}} \leq C$ follows once again by considering the SVD of $\bar{F}$.

**Controlling $Y \bar{K}^{-1} \mathbf{v}^\top$:** Here, we introduce conditioning on the event $\mathcal{E}_{\text{data}}$ and note its independence from $\beta$ and $\varepsilon$. Recalling that $Y = \beta^\top X/\sqrt{n_0} + \varepsilon$, we note that $Y$ has expectation zero since $\mathbb{E}_{\beta,\varepsilon}[Y \bar{K}^{-1} \mathbf{v}^\top | \mathcal{E}_{\text{data}}] = 0$. Similarly the conditional variance can be bounded as

$$\mathbb{V}_{\beta,\varepsilon}[Y \bar{K}^{-1} \mathbf{v}^\top | \mathcal{E}_{\text{data}}] = \mathbf{v}^\top \bar{K}^{-1} (\Upsilon + \sigma_\varepsilon^2 I_m) \bar{K}^{-1} \mathbf{v} \tag{A207}$$

$$\leq \left\| \Upsilon + \sigma_\varepsilon^2 I_m \right\|_\infty \left\| \bar{K}^{-1} \frac{\bar{F}^\top}{\sqrt{n_1}} \right\|_\infty^2 \left\| \frac{1_{n_1}}{\sqrt{n_1}} \right\|_2^2 \tag{A208}$$

$$\leq C. \tag{A209}$$

The bound $\|\bar{K}^{-1} \frac{\bar{F}^\top}{\sqrt{n_1}}\|_\infty \leq C$ follows by considering the SVD of $F$ as in the proof of Lem. A9.6, while the bound $\|\Upsilon\| \leq C$ holds on $\mathcal{E}_{\text{data}}$. The law of total variance then shows $\mathbb{V}[Y \bar{K}^{-1} \mathbf{v}^\top | \mathcal{E}_{\text{data}}] \leq C$ also. Hence an application of Chebyshev's inequality shows that

$$\mathbb{P}[|Y \bar{K}^{-1} \mathbf{v}^\top| \geq C n_0^c | \mathcal{E}_{\text{data}}] \leq C n_0^{-c} \tag{A210}$$

for some $C > 0$ and any $c > 0$.

**Controlling $Y \bar{K}^{-1} \mathbf{u}^\top$:** Again, we introduce conditioning on the event $\mathcal{E}_{\text{data}}$ and note its independence from $\beta$ and $\varepsilon$. A nearly identical argument to the previous one suffices, so we only outline the

argument. Again $\mathbb{E}[Y\bar{K}^{-1}\mathbf{v}^\top|\mathcal{E}_{\text{data}}] = 0$ and the conditional variance can be bounded as

$$\mathbb{V}_{\beta,\varepsilon}[Y\bar{K}^{-1}\mathbf{u}^\top|\mathcal{E}_{\text{data}}] = \mathbf{u}^\top\bar{K}^{-1}(\Upsilon + \sigma_\varepsilon^2 I_m)\bar{K}^{-1}\mathbf{u} \tag{A211}$$

$$\leq \|\bar{K}^{-1}\|_\infty^2\|\Upsilon + \sigma_\varepsilon^2 I_m\|_\infty\|\mathbf{u}\|_2^2 \tag{A212}$$

$$\leq \frac{1}{\gamma^2}\cdot a^2 m \cdot \|\Upsilon + \sigma_\varepsilon^2 I_m\|_\infty \tag{A213}$$

$$\leq Cn_0. \tag{A214}$$

Finally, Chebyshev's inequality suffices to show

$$\mathbb{P}[|Y\bar{K}^{-1}\mathbf{u}^\top| \geq Cn_0^{c+1/2}|\mathcal{E}_{data}] \leq Cn_0^{-c} \tag{A215}$$

for some $C > 0$ and any $c > 0$.

**Controlling $\bar{K}_{\mathbf{x}}^\top\bar{K}^{-1}\mathbf{u}^\top$:** Here, we introduce conditioning on the event $\mathcal{E}_W$ and note its independence from $\mathbf{x}$. The overall argument then uses Chebyshev's inequality exploiting the randomness in $\mathbf{x}$. First,

$$\left|\mathbb{E}_{\mathbf{x}}[\bar{K}_{\mathbf{x}}^\top\bar{K}^{-1}\mathbf{u}^\top|\mathcal{E}_W]\right| = \left|\mathbb{E}_{\mathbf{x}}[\bar{f}^\top|\mathcal{E}_W]\frac{\bar{F}}{\sqrt{n_1}}\bar{K}^{-1}\mathbf{u}^\top/\sqrt{n_1}\right| \tag{A216}$$

$$\leq \left\|\mathbb{E}_{\mathbf{x}}[\bar{f}|\mathcal{E}_W]\right\|_2\left\|\frac{\bar{F}}{\sqrt{n_1}}\bar{K}^{-1}\right\|_\infty\|\mathbf{u}/\sqrt{n_1}\|_2 \tag{A217}$$

$$\leq Cn_0^c. \tag{A218}$$

The bound $\|\frac{\bar{F}}{\sqrt{n_1}}\bar{K}^{-1}\|_\infty \leq C$ follows by an SVD as before, while the first bound $\|\mathbb{E}_{\mathbf{x}}[\bar{f}^\top|\mathcal{E}_W]\|_2 \leq Cn_0^c$ follows from the fact $\mathbb{E}_{\mathbf{x}}\bar{\sigma}(W\mathbf{x})_i \leq Cn_0^{c-1/2}$ on $\mathcal{E}_W$ (see Lem. A9.4). Next we turn to the conditional variance. We see

$$\mathbb{V}_{\mathbf{x}}[\bar{K}_{\mathbf{x}}\bar{K}^{-1}\mathbf{u}|\mathcal{E}_W] = \mathbf{u}^\top\bar{K}^{-1}\mathbb{E}_{\mathbf{x}}[\bar{K}_{\mathbf{x}}\bar{K}_{\mathbf{x}}^\top|\mathcal{E}_W]\bar{K}^{-1}\mathbf{u} \tag{A219}$$

$$= \frac{1}{n_1^2}\mathbf{u}^\top\bar{K}^{-1}\bar{F}^\top\left(\frac{\rho}{n_0}W\Sigma^*W^\top + (\eta - \zeta)I_{n_1} + \Delta\right)\bar{F}\bar{K}^{-1}\mathbf{u} \tag{A220}$$

$$\leq \left\|\frac{\mathbf{u}}{\sqrt{n_1}}\right\|_2^2\left\|\bar{K}^{-1}\frac{\bar{F}^\top}{\sqrt{n_1}}\right\|_2^2\cdot\left(\left\|\rho\tilde{\Upsilon}\right\|_\infty + \|(\eta - \zeta)I_{n_1}\|_\infty + \|\Delta\|_\infty\right) \tag{A221}$$

$$\leq C\cdot(C + C + n_0^c) \tag{A222}$$

$$\leq Cn_0^c. \tag{A223}$$

Once again the bound $\|\bar{K}^{-1}\frac{\bar{F}^\top}{\sqrt{n_1}}\|_2^2$ follows by considering the SVD, while the nontrivial operator norm bounds follow from Lem. A9.4. Using the law of total variance, we see

$$\mathbb{V}[\bar{K}_{\mathbf{x}}\bar{K}^{-1}\mathbf{u}|\mathcal{E}_W] = \mathbb{E}[\mathbb{V}_{\mathbf{x}}[\bar{K}_{\mathbf{x}}\bar{K}^{-1}\mathbf{u}|\mathcal{E}_W]|\mathcal{E}_W] + \mathbb{V}[\mathbb{E}_{\mathbf{x}}[\bar{K}_{\mathbf{x}}\bar{K}^{-1}\mathbf{u}|\mathcal{E}_W]|\mathcal{E}_W] \leq Cn_0^c \tag{A224}$$

by Eqs. (A218) and (A223). Applying Chebyshev's inequality yields

$$\mathbb{P}[|\bar{K}_{\mathbf{x}}^\top\bar{K}^{-1}\mathbf{u} - \mathbb{E}_{\mathbf{x}}[\bar{K}_{\mathbf{x}}^\top\bar{K}^{-1}\mathbf{u}]| \geq Cn_0^c|\mathcal{E}_W] \leq Cn_0^{-c}, \tag{A225}$$

and so

$$\mathbb{P}[|\bar{K}_{\mathbf{x}}^\top\bar{K}^{-1}\mathbf{u}| \geq Cn_0^c|\mathcal{E}_W] \leq Cn_0^{-c} \tag{A226}$$

by the triangle inequality since $|\mathbb{E}_{\mathbf{x}}[\bar{K}_{\mathbf{x}}^\top\bar{K}^{-1}\mathbf{u}]| \leq Cn_0^c$.

**Controlling $\bar{K}_{\mathbf{x}}^\top\bar{K}^{-1}\mathbf{v}^\top$:** Again, we introduce conditioning on the event $\mathcal{E}_W$ and note its independence from $\mathbf{x}$. A nearly identical argument to the previous one shows the result for this term. Hence we only outline the argument. First, we see

$$\left|\mathbb{E}_{\mathbf{x}}[\bar{K}_{\mathbf{x}}^\top\bar{K}^{-1}\mathbf{v}^\top|\mathcal{E}_W]\right| = \left|\mathbb{E}_{\mathbf{x}}[\bar{f}^\top|\mathcal{E}_W]\frac{\bar{F}}{\sqrt{n_1}}\bar{K}^{-1}\frac{\bar{F}^\top}{\sqrt{n_1}}\frac{1_{n_1}}{n_1}\right| \tag{A227}$$

$$\leq \left\|\mathbb{E}_{\mathbf{x}}[\bar{f}^\top|\mathcal{E}_W]\right\|_2\left\|\frac{\bar{F}}{\sqrt{n_1}}\bar{K}^{-1}\frac{\bar{F}^\top}{\sqrt{n_1}}\right\|_\infty\left\|\frac{1_{n_1}}{n_1}\right\|_2 \tag{A228}$$

$$\leq Cn_0^{c-1/2}. \tag{A229}$$

The bound $\|\frac{\bar{F}}{\sqrt{n_1}}\bar{K}^{-1}\frac{\bar{F}^\top}{\sqrt{n_1}}\|_\infty \leq 1$ follows by an SVD as before, while the first bound again follows from Lem. A9.4. Next we compute the conditional variance and bound as in Eq. (A223) to find

$$\mathbb{V}_\mathbf{x}[\bar{K}_\mathbf{x}\bar{K}^{-1}\mathbf{v}^\top|\mathcal{E}_W] = \mathbf{v}^\top \bar{K}^{-1}\mathbb{E}_\mathbf{x}[\bar{K}_\mathbf{x}\bar{K}_\mathbf{x}^\top|\mathcal{E}_W]\bar{K}^{-1}\mathbf{v} \tag{A230}$$

$$= \frac{1}{n_1^4}1_{n_1}^\top \bar{F}\bar{K}^{-1}\bar{F}^\top \left(\frac{\rho}{n_0}W\Sigma^*W^\top + (\eta-\zeta)I_{n_1} + \Delta\right)\bar{F}\bar{K}^{-1}\bar{F}^\top 1_{n_1} \tag{A231}$$

$$\leq \frac{1}{n_1}\left\|\frac{1_{n_1}}{\sqrt{n_1}}\right\|_2^2 \cdot \left\|\frac{\bar{F}}{\sqrt{n_1}}\bar{K}^{-1}\frac{\bar{F}^\top}{\sqrt{n_1}}\right\|_2^2 \cdot \left(\left\|\rho\tilde{\Upsilon}\right\|_\infty + \|(\eta-\zeta)I_{n_1}\|_\infty + \|\Delta\|_\infty\right) \tag{A232}$$

$$\leq \frac{C\cdot(C+C+n_0^c)}{n_1} \tag{A233}$$

$$\leq Cn_0^{c-1}. \tag{A234}$$

Once again the bound $\|\frac{\bar{F}}{\sqrt{n_1}}\bar{K}^{-1}\frac{\bar{F}^\top}{\sqrt{n_1}}\|_2 \leq 1$ follows by considering the SVD while the nontrivial operator norm bounds follow on $\mathcal{E}_W$. Now applying Chebyshev's inequality as before shows

$$\mathbb{P}[|\bar{K}_\mathbf{x}^\top \bar{K}^{-1}\mathbf{v}^\top| \geq Cn_0^{c-1/2}|\mathcal{E}_W] \leq Cn_0^{-c}. \tag{A235}$$

**Controlling $y(\mathbf{x})$:** Finally we show $y(\mathbf{x}) \leq Cn_0^c$ with probability at least $Cn_0^{-c}$. Note that $\mathbb{E}[y(\mathbf{x})] = 0$ and a short computation shows that $\mathbb{V}[y(\mathbf{x})] = \bar{\text{tr}}(\Sigma^*) + \sigma_\epsilon^2 \leq C$. So a direct application of Chebyshev's inequality shows that

$$\mathbb{P}[|y(\mathbf{x})| \geq Cn_0^c] \leq Cn_0^{-c}. \tag{A236}$$

$\square$

### A9.1.5  Completing the argument for the typical behavior

**Lemma A9.8.** *For some $n_0$-independent constants $c > 0$ and $C > 0$, we have*

$$|\mathbb{E}[\mathbf{1}_\mathcal{E}\Delta y(\mathbf{x})]| \leq Cn_0^{-c} \tag{A237}$$

*and*

$$\left|\mathbb{E}[\mathbf{1}_\mathcal{E}\Delta(K_\mathbf{x}^\top K^{-1}Y^\top + \bar{K}_\mathbf{x}^\top \bar{K}^{-1}Y^\top)]\right| \leq Cn_0^{-c} \tag{A238}$$

*Proof.* Obviously, $|\mathbb{E}[\mathbf{1}_\mathcal{E}\Delta y(\mathbf{x})]|$ is bounded by $|\mathbb{E}[\Delta y(\mathbf{x})]|\mathcal{E}|$, so we can remove the indicator $\mathbf{1}_\mathcal{E}$ from the expectations in Eqs. (A237) and (A238) by conditioning on $\mathcal{E}$.

We want to show that conditional on $\mathcal{E}$ and for some $n_0$-independent constants $c > 0$ and $C > 0$, the bounds $|\Delta| \leq Cn_0^{-2c}$, $|y(\mathbf{x})| \leq Cn_0^c$, and $\left|(K_\mathbf{x}^\top K^{-1}Y^\top + \bar{K}_\mathbf{x}^\top \bar{K}^{-1}Y^\top)\right| \leq Cn_0^c$ hold.

Recall, $\quad a := \mathbb{E}_Z\sigma\left(\sqrt{\bar{\text{tr}}(\Sigma)}Z\right)$, $\quad a^* := \mathbb{E}_Z\sigma\left(\sqrt{\bar{\text{tr}}(\Sigma^*)}Z\right)$, $\quad \mathbf{v} := 1/n_1\bar{F}^\top 1_{n_1}$, $\quad \mathbf{u} := a1_m$, $U := [\mathbf{u}, \mathbf{v}]^\top$, and $C := (\begin{smallmatrix}1 & 1\\ 1 & 0\end{smallmatrix})$.

First, consider the term $\Delta = YT_1 + YT_2$. To show that $|\Delta| \leq Cn_0^{-2c}$ on $\mathcal{E}$, we write out $YT_1$ and $YT_2$ more explicitly:

$$YT_1 = Y\bar{K}^{-1}U^\top\left(I_2 + CU\bar{K}^{-1}U^\top\right)^{-1}P \tag{A239}$$

$$= \frac{1}{c_1^2 + mc_0c_2}\begin{pmatrix}Y\bar{K}^{-1}\mathbf{u}\\ Y\bar{K}^{-1}\mathbf{v}\end{pmatrix}^\top \begin{pmatrix}c_1\delta + c_2a^*\\ -mc_0\delta + c_1a^*\end{pmatrix} \tag{A240}$$

$$= \frac{1}{c_1^2 + mc_0c_2}\left(Y\bar{K}^{-1}\mathbf{u}\cdot(c_1\delta + c_2a^*) + Y\bar{K}^{-1}\mathbf{v}\cdot(-mc_0\delta + c_1a^*)\right) \tag{A241}$$

$$\tag{A242}$$

and

$$YT_2 = -Y\bar{K}^{-1}U^\top \left(I_2 + CU\bar{K}^{-1}U^\top\right)^{-1} CU\bar{K}^{-1}\bar{K}_\mathbf{x} \tag{A243}$$

$$= \frac{1}{c_1^2 + mc_0c_2} \begin{pmatrix} Y\bar{K}^{-1}\mathbf{u} \\ Y\bar{K}^{-1}\mathbf{v} \end{pmatrix}^\top \begin{pmatrix} c_2 & c_1 \\ c_1 & -mc_0 \end{pmatrix} \begin{pmatrix} \mathbf{u}^\top \bar{K}^{-1}\bar{K}_\mathbf{x} \\ \mathbf{v}^\top \bar{K}^{-1}\bar{K}_\mathbf{x} \end{pmatrix} \tag{A244}$$

$$\leq \frac{1}{c_1^2 + mc_0c_2} \left(Y\bar{K}^{-1}\mathbf{u} \cdot (c_2\mathbf{u}^\top \bar{K}^{-1}\bar{K}_\mathbf{x} + c_1\mathbf{v}^\top \bar{K}^{-1}\bar{K}_\mathbf{x}) \right. \tag{A245}$$

$$\left. + Y\bar{K}^{-1}\mathbf{v} \cdot (c_1\mathbf{u}^\top \bar{K}^{-1}\bar{K}_\mathbf{x} - mc_0\mathbf{v}^\top \bar{K}^{-1}\bar{K}_\mathbf{x})\right). \tag{A246}$$

Now using Eq. (A239) and the definition of $\mathcal{E}$, we see that on $\mathcal{E}$

$$|YT_1| \leq \left|\frac{1}{c_1^2 + mc_0c_2}\right| \left(\left|Y\bar{K}^{-1}\mathbf{u}\right| (|c_1|\,|\delta| + |c_2|\,|a^*|) + \left|Y\bar{K}^{-1}\mathbf{v}\right| (m\,|c_0|\,|\delta| + |c_1|\,|a^*|)\right) \tag{A247}$$

$$\leq Cn_0^{2c-1}\left(Cn_0^{c+1/2}(Cn_0^c Cn_0^{c-1/2} + Cn_0^c) + Cn_0^c(Cn_0 Cn_0^c Cn_0^{c-1/2} + Cn_0^c)\right) \tag{A248}$$

$$\leq Cn_0^{5c-1/2}. \tag{A249}$$

Since the $c$s used in the bounds above can be arbitrarily small, we may replace write $|YT_1| \leq Cn_0^{c-1/2}$ for arbitrarily small $c > 0$. Finally, a similar argument for Eq. (A243) yields the bound

$$|YT_2| \leq \left|\frac{1}{c_1^2 + mc_0c_2}\right| \left(\left|Y\bar{K}^{-1}\mathbf{u}\right| \cdot (|c_2||\mathbf{u}^\top \bar{K}^{-1}\bar{K}_\mathbf{x}| + |c_1||\mathbf{v}^\top \bar{K}^{-1}\bar{K}_\mathbf{x}|) \right. \tag{A250}$$

$$\left. + |Y\bar{K}^{-1}\mathbf{v}| \cdot (|c_1||\mathbf{u}^\top \bar{K}^{-1}\bar{K}_\mathbf{x}| - m|c_0||\mathbf{v}^\top \bar{K}^{-1}\bar{K}_\mathbf{x}|)\right) \tag{A251}$$

$$\leq Cn_0^{2c-1} \cdot \left(Cn_0^{c+1/2} \cdot (Cn_0^c Cn_0^c + Cn_0^{c+1/2} Cn_0^{c-1/2}) \right. \tag{A252}$$

$$\left. + Cn_0^c \cdot (Cn_0^{c+1/2} Cn_0^c + Cn_0 Cn_0^c Cn_0^{c-1/2})\right) \tag{A253}$$

$$\leq Cn_0^{2c-1} \cdot (Cn_0^{3c+1/2} + Cn_0^{3c+1/2}) \tag{A254}$$

$$\leq Cn_0^{5c-1/2}. \tag{A255}$$

and as before since $c$ can be chosen arbitrarily small we can write $|YT_2| \leq Cn_0^{c-1/2}$. $\square$

### A9.2 Gaussian equivalents

Before describing how Gaussian equivalents can be used to compute the test error in the high-dimensional limit, we make the following observation: the asymptotic test error is invariant to the centering of the activation function across the train and test distributions. More concretely, from the results in Sec. A9.1, the asymptotic test error is unchanged under the replacements $F_{ij} \to \bar{F}_{ij}$ and $f_i \to \bar{f}_i$, for all $i$ and $j$, where,

$$\bar{F}_{ij} := F_{ij} - \mathbb{E}_{z \sim \mathcal{N}(0, \bar{\mathrm{tr}}(\Sigma))} \sigma(z) \quad \text{and} \quad \bar{f}_i := f_i - \mathbb{E}_{z \sim \mathcal{N}(0, \bar{\mathrm{tr}}(\Sigma^*))} \sigma(z) . \tag{A256}$$

With this simplification in mind, the proof of Thm. 5.1 relies on the concept of Gaussian equivalents and the linearization analysis developed in [2, 3, 1], which we briefly review here, though we refer the reader to these works for a more detailed description. The centered activation-constants are defined as $\bar{\zeta} = s\bar{\rho}$ with

$$\bar{\eta} := \mathbb{E}_{z \sim \mathcal{N}(0,s)}[\bar{\sigma}(z)^2] \quad \text{and} \quad \bar{\rho} := \left(\frac{1}{s}\mathbb{E}_{z \sim \mathcal{N}(0,s)}[z\bar{\sigma}(z)]\right)^2 . \tag{A257}$$

Note that by the continuity of $\sigma$, we know that as $n_0 \to \infty$

$$\mathbb{E}_{z \sim \mathcal{N}(0, \bar{\mathrm{tr}}(\Sigma))}[\bar{\sigma}(z)^2] \to \bar{\eta}. \tag{A258}$$

Similarly for $\bar{\zeta}$ and $\bar{\rho}$ or when $\bar{\mathrm{tr}}(\Sigma)$ is replaced by $\bar{\mathrm{tr}}(\Sigma^*)$. To proceed, we define the moment-matched Gaussian linearizations,

$$\bar{F} \to \sqrt{\frac{\bar{\rho}}{n_0}}WX + \sqrt{\bar{\eta} - \bar{\zeta}}\Theta \quad \text{and} \quad \bar{f} \to \sqrt{\frac{\bar{\rho}}{n_0}}W\mathbf{x} + \sqrt{\bar{\eta} - \bar{\zeta}}\theta \tag{A259}$$

where $\bar{f} := \bar{\sigma}(W\mathbf{x}/\sqrt{n_0})$ is the random feature representation of the test point $\mathbf{x}$ and $y := \beta^\top \mathbf{x}/\sqrt{n_0}$ is its corresponding label, which we assume has no additive noise[9].

The new objects $\Theta$ and $\theta$ are matrices of the appropriate shapes with i.i.d. standard Gaussian entries independent of the other random variables under consideration. The constants $\bar{\eta}$ and $\bar{\zeta}$ are chosen so that the mixed moments up to second order are the same for the original and linearized matrices. The Gaussian equivalents defined are essentially constructed via a Taylor expansion of the nonlinearity $\bar{\sigma}$. The explicit calculations defining these expressions can be found via the Gaussian moment matching technique in [1].

Having appropriately linearized the random feature matrices, we can map back to the definition of the original activation functions by recalling that the variable $\zeta$, defined in Sec. 5.1 as $\zeta = s\rho$, with

$$\eta := \bar{\eta} = \mathbb{E}_{z\sim\mathcal{N}(0,s)}[(\sigma(z) - \mathbb{E}_{z\sim\mathcal{N}(0,s)}[\sigma(z)])^2] = \mathbb{V}_{z\sim\mathcal{N}(0,s)}[\sigma(z)] \tag{A260}$$

$$\rho := \bar{\rho} = (\frac{1}{s}\mathbb{E}_{z\sim\mathcal{N}(0,s)}[z(\sigma(z) - \mathbb{E}_{z\sim\mathcal{N}(0,s)}[\sigma(z)])])^2 = (\frac{1}{s}\mathbb{E}_{z\sim\mathcal{N}(0,s)}[z(\sigma(z)])^2 \tag{A261}$$

by using the definition of the centering. Effectively, this is equivalent to using the linearizations,

$$F \to \bar{F} \to \sqrt{\frac{\rho}{n_0}}WX + \sqrt{\eta - \zeta}\Theta \quad \text{and} \quad f \to \bar{f} \to \sqrt{\frac{\rho}{n_0}}W\mathbf{x} + \sqrt{\eta - \zeta}\theta \tag{A262}$$

directly to compute the test error.

Simply by definition, the teacher function is exactly linear,

$$Y = \sqrt{\frac{1}{n_0}}\beta^\top X + \varepsilon \quad \text{and} \quad y = \sqrt{\frac{1}{n_0}}\beta^\top \mathbf{x}. \tag{A263}$$

We emphasize that the restriction to linear teacher functions is simply a matter of convenience and simplicity — nonlinear teacher neural networks can be studied by means of an analogous linearization process, as discussed in [2], which merely requires introducing additional moment-matching constants and additional i.i.d. standard Gaussian terms $\theta$ and $\theta_y$.

In the high-dimensional limit the bulk statistics we compute defining the error, bias, and variance are invariant to the above replacements by linearized Gaussian equivalents. We remark that further intuition for these linearized information-plus-noise replacements can be gathered from the universality results of [5, 6]. To summarize briefly, the final expressions we compute are tracial (nonlinear) functions of several random matrices. A large body of universality results (e.g. [6, 17]) show that in the asymptotic limit it is sufficient to calculate with an equivalent tracial functional of a (linearized) rational function of random matrices, whose moments match their nonlinear counterparts.

More specifically, two basic trace objects arise in the calculations, which take the form

$$\text{tr}(AB) \quad \text{and} \quad \text{tr}\left(A\frac{1}{B - zI}\right). \tag{A264}$$

For the first case, $\text{tr}(AB)$, if the random matrices $A$ and $B$ are independent, its asymptotics can be understood via concentration of measure arguments. Conditionally on $A$, it is sufficient to replace $B$ with an equivalent matrix designed to match low-order moments, at the expense of an error that vanishes asymptotically. The second case, $\text{tr}\left(A\frac{1}{B-zI}\right)$, is a bit more involved. Our arguments proceed by 1) utilizing the linearized matrices in Eq. (A262) to express the trace object as a rational function of i.i.d. Gaussian matrices, and then 2) using the linear pencil method[10] [25, 45] to express it as the trace of a large (inverted) block matrix. The reason the linearization over $B$ preserves the asymptotic statistics even for general $A$ with correlated entries stems from the matrix Dyson equation [17].

### A9.3 Decomposition of the test loss

Recall the expression for the test loss in Eq. (A84),

$$E_{\Sigma^*} = E_1 + E_2 + E_3 \tag{A265}$$

---

[9]Including noise on the test labels merely shifts by the total error by an irreducible additive constant.

[10]This is essentially an iterative application of the Schur complement formula.

with

$$E_1 = \mathbb{E}_{(\mathbf{x},\beta,\varepsilon)} y(\mathbf{x})^2 = \mathbb{E}_{(\mathbf{x},\varepsilon)} \text{tr}(y(\mathbf{x})y(\mathbf{x})^\top) \tag{A266}$$

$$E_2 = -2\mathbb{E}_{(\mathbf{x},\beta,\varepsilon)}(K_\mathbf{x}^\top K^{-1} Y)y(\mathbf{x}) = -2\mathbb{E}_{(\mathbf{x},\varepsilon)} \text{tr}(K_\mathbf{x}^\top K^{-1} Y^\top y(\mathbf{x})) \tag{A267}$$

$$E_3 = \mathbb{E}_{(\mathbf{x},\beta,\varepsilon)}(K_\mathbf{x}^\top K^{-1} Y^\top)^2 = \mathbb{E}_{(\mathbf{x},\varepsilon)} \text{tr}(K_\mathbf{x}^\top K^{-1} Y^\top Y K^{-1} K_\mathbf{x}) \,, \tag{A268}$$

where the kernels $K = K(X,X)$ and $K_\mathbf{x} = K(X,\mathbf{x})$ are given by,

$$K = \frac{F^\top F}{n_1} + \gamma I_m \qquad \text{and} \qquad K_\mathbf{x} = \frac{1}{n_1} F^\top f \,. \tag{A269}$$

Using the cyclicity and linearity of the trace, the expectation over $\mathbf{x}$ requires the computation of

$$\mathbb{E}_\mathbf{x} K_\mathbf{x} K_\mathbf{x}^\top \,, \qquad \mathbb{E}_\mathbf{x} y(\mathbf{x}) K_\mathbf{x}^\top \,, \qquad \mathbb{E}_\mathbf{x} y(\mathbf{x}) y(\mathbf{x})^\top \,. \tag{A270}$$

As described in Sec. A9.2, without loss of generality we consider the case of a linear teacher, and Eqs. (A262) and (A263) read

$$y = \frac{1}{\sqrt{n_0}} \beta^\top \mathbf{x} \qquad \text{and} \qquad f \to f^\text{lin} = \frac{\sqrt{\rho}}{\sqrt{n_0}} W\mathbf{x} + \sqrt{\eta - \zeta}\,\theta \,. \tag{A271}$$

The expectations over $\mathbf{x}$ are now trivial and we readily find,

$$\mathbb{E}_\mathbf{x} K_\mathbf{x} K_\mathbf{x}^\top = \frac{1}{n_1^2} F^\top \left( \frac{\rho}{n_0} W \Sigma^* W^\top + (\eta - \zeta) I_{n_1} \right) F \tag{A272}$$

$$\mathbb{E}_\mathbf{x} y(\mathbf{x}) K_\mathbf{x}^\top = \frac{\sqrt{\rho}}{n_0 n_1} \beta^\top \Sigma^* W^\top F \tag{A273}$$

$$\mathbb{E}_\mathbf{x} y(\mathbf{x}) y(\mathbf{x})^\top = \frac{1}{n_0} \beta \Sigma^* \beta^\top \tag{A274}$$

One may interpret the substitution in Eq. (A271) as a tool to calculate the expectations above to leading order, which generates terms like Eq. (A264). Next, we recall the definition, $Y = \beta^\top X/\sqrt{n_0} + \varepsilon$, and, as above, we consider the leading order behavior with respect to the random variables $\beta$ to find

$$\mathbb{E}_{\beta,\varepsilon}\left[ Y^\top Y \right] = \frac{1}{n_0} X^\top X + \sigma_\varepsilon^2 I_m \tag{A275}$$

$$\mathbb{E}_{\beta,\varepsilon}\left[ Y^\top \mathbb{E}_\mathbf{x} y(\mathbf{x}) K_\mathbf{x}^\top \right] = \frac{\sqrt{\rho}}{n_0^{3/2} n_1} X^\top \Sigma^* W^\top F \,. \tag{A276}$$

Putting these pieces together, we have

$$E_1 = \frac{\text{tr}(\Sigma^*)}{n_0} \tag{A277}$$

$$E_2 = E_{21} \tag{A278}$$

$$E_3 = E_{31} + E_{32} \,, \tag{A279}$$

where,

$$E_{21} = -2\frac{\sqrt{\rho}}{n_0^{3/2} n_1} \mathbb{E}\text{tr}\left( X^\top \Sigma^* W^\top F K^{-1} \right) \tag{A280}$$

$$E_{31} = \sigma_\varepsilon^2 \mathbb{E}\text{tr}\left( K^{-1} \Sigma_3 K^{-1} \right) \tag{A281}$$

$$E_{32} = \frac{1}{n_0} \mathbb{E}\text{tr}\left( K^{-1} \Sigma_3 K^{-1} X^\top X \right) \tag{A282}$$

and,

$$\Sigma_3 = \frac{\rho}{n_0 n_1^2} F^\top W \Sigma^* W^\top F + \frac{\eta - \zeta}{n_1^2} F^\top F \,. \tag{A283}$$

## A9.4 Decomposition of the bias and total variance

Note that it is sufficient to calculate the bias term since given the total test loss, since the total variance can be obtained as $V_{\Sigma^*} = E_{\Sigma^*} - B_{\Sigma^*}$. Following the total multivariate bias-variance decomposition of [3], for each random variable in question we introduce an i.i.d. copy of it denoted by either the subscript 1 or 2. We can then write,

$$B_{\Sigma^*} = \mathbb{E}_{(\mathbf{x},y)}(y - \mathbb{E}_{(W,X,\varepsilon)}\hat{y}(\mathbf{x}; W, X, \varepsilon))^2 \tag{A284}$$

$$= \mathbb{E}_{(\mathbf{x},y)}\mathbb{E}_{(W_1,X_1,\varepsilon_1)}\mathbb{E}_{(W_2,X_2,\epsilon_2)}(y - \hat{y}(\mathbf{x}; W_1, X_1, \varepsilon_1))(y - \hat{y}(\mathbf{x}; W_2, X_2, \epsilon_2)) \tag{A285}$$

$$= \frac{\mathrm{tr}(\Sigma^*)}{n_0} + E_{21} + H_{000}\,, \tag{A286}$$

where an expression for $E_{21}$ was given previously and $H_{000}$ satisfies

$$H_{000} = \mathbb{E}\hat{y}(\mathbf{x}; W_1, X_1, \varepsilon_1)\hat{y}(\mathbf{x}; W_2, X_2, \epsilon_2)\,, \tag{A287}$$

where the expectations are over $\mathbf{x}, W_1, X_1, \varepsilon_1, W_2, X_2$, and $\epsilon_2$. Recalling the definition of $\hat{y}$,

$$\hat{y}(\mathbf{x}; W, X, \varepsilon) := Y(X, \epsilon)K(X, X; W)^{-1}K(X, \mathbf{x}; W) \tag{A288}$$

and the techniques described in the previous section, it is straightforward to analyze the above term. First note we can write,

$$\mathbb{E}_{\mathbf{x}}K(X_1, \mathbf{x}; W_1)K(\mathbf{x}, X_2; W_2) = \frac{\rho}{n_0 n_1^2}F_{11}^\top W_1 \Sigma^* W_2^\top F_{22} \tag{A289}$$

since the $f$ linearizations use different sources of auxiliary randomness. Here we have defined $F_{11} \equiv F(W_1, X_1)$ and $F_{22} \equiv F(W_2, X_2)$. Now we proceed to calculate $H_{000}$ as

$$H_{000} = \mathbb{E}\hat{y}(\mathbf{x}; W_1, X_1, \varepsilon)\hat{y}(\mathbf{x}; W_2, X_2, \epsilon_2) \tag{A290}$$

$$= \mathbb{E}K(\mathbf{x}, X_2; W_2)K(X_2, X_2; W_2)^{-1}Y(X_2, \epsilon_2)^\top Y(X_1, \varepsilon_1)K(X_1, X_1; W_1)^{-1}K(X_1, \mathbf{x}; W) \tag{A291}$$

$$= \mathbb{E}\mathrm{tr}\big(K(X_2, X_2; W_2)^{-1}X_2^\top X_1 K(X_1, X_1; W_1)^{-1}K(X_1, \mathbf{x}; W)K(\mathbf{x}, X_2; W_2)\big) \tag{A292}$$

$$= \frac{\rho}{n_0^2 n_1^2}\mathbb{E}\mathrm{tr}\big(K_{22}^{-1}X_2^\top X_1 K_{11}^{-1}F_{11}^\top W_1 \Sigma^* W_2^\top F_{22}\big) \tag{A293}$$

$$\equiv E_4\,, \tag{A294}$$

where in the second-to-last line we have defined $K_{11} \equiv K(X_1, X_1; W_1)$ and $K_{22} \equiv K(X_2, X_2; W_2)$.

## A9.5 Summary of linearized trace terms

We now summarize the requisite terms needed to compute the total test error, bias, and variance after using cyclicity of the trace to rearrange several of them. In the following, we slightly change notation in order to make explicit the dependence on the population covariance matrices $\Sigma$ and $\Sigma^*$. To be specific, whereas above we assumed that the columns of $X_1$ and $X_2$ were drawn from multivariate Gaussians with covariance $\Sigma$, below we assume that they are drawn from multivariate Gausssians with identity covariance. This change is equivalent to replaceing $X_1 \to \Sigma^{1/2}X_1$ and $X_2 \to \Sigma^{1/2}X_2$ in the above expressions. We utilize this definition so that $X_1$, $X_2$, $W_1$, $W_2$, and $\Theta$ all have i.i.d.

standard Gaussian entries. From the previous computations, we can now write the requisite terms as,

$$\Sigma_3 = \frac{\rho}{n_0 n_1^2} F_{11}^\top W_1 \Sigma^* W_1^\top F_{11} + \frac{\eta - \zeta}{n_1^2} F_{11}^\top F_{11} \tag{A295}$$

$$E_{21} = -2 \frac{\sqrt{\rho}}{n_0^{3/2} n_1} \mathrm{tr} \left( X_1^\top \Sigma^{1/2} \Sigma^* W_1^\top F_{11} K_{11}^{-1} \right) \tag{A296}$$

$$E_{31} = \sigma_\epsilon^2 \mathrm{tr} \left( K_{11}^{-1} \Sigma_3 K_{11}^{-1} \right) \tag{A297}$$

$$E_{32} = \frac{1}{n_0} \mathrm{tr} \left( K_{11}^{-1} \Sigma_3 K_{11}^{-1} X_1^\top \Sigma X_1 \right) \tag{A298}$$

$$E_4 = \frac{\rho}{n_0^2 n_1^2} \mathrm{tr} \left( F_{22} K_{22}^{-1} X_2^\top \Sigma X_1 K_{11}^{-1} F_{11}^\top W_1 \Sigma^* W_2^\top \right) \tag{A299}$$

$$E_{\Sigma^*} = \frac{\mathrm{tr}(\Sigma^*)}{n_0} + E_{21} + E_{31} + E_{32} \tag{A300}$$

$$B_{\Sigma^*} = \frac{\mathrm{tr}(\Sigma^*)}{n_0} + E_{21} + E_4 \tag{A301}$$

$$V_{\Sigma^*} = E_{\Sigma^*} - B_{\Sigma^*} \tag{A302}$$

In the remainder of this section we use the machinery of operator-valued free probability [45] and a series of lengthy algebraic computations to compute the asymptotic tracial expressions in $E_{21}, E_{31}, E_{32}, E_4$, from which the total test error, bias, and variance can be reconstructed.

## A9.6 Calculation of error terms

To compute the test error, bias, and total variance, we need to evaluate the asymptotic trace objects appearing in the expressions for $E_{21}, E_{31}, E_{32}$, and $E_4$, defined in the previous section. As these expressions are essentially rational functions of the random matrices $X$, $W$, $\Theta$, $\Sigma$, and $\Sigma^*$, these computations can be accomplished by constructing a linear pencil [19] and using the theory of operator-valued free probability [45]. These techniques and their application to problems of this type have been well-established elsewhere [1, 2, 3], we only sketch the mathematical details, referring the reader to the literature for a more pedagogical overview. Instead, we focus on presenting the details of the requisite calculations.

Relative to the prior work of [2, 3], the main challenge in the current setting is generalizing the calculations to include an arbitrary training covariance matrix $\Sigma$. This generalization is facilitated by the general theory of operator-valued free probability, and in particular through the subordinated form of the operator-valued self-consistent equations that we first present in Eq. (A322). The form of this equation enables the simple computation of the operator-valued R-transform of the remaining random matrices, $W$, $X$, and $\Theta$, which are all i.i.d. Gaussian and can therefore be obtained simply by using the methods of [19]. The remaining complication amounts to performing the trace in Eq. (A322), which becomes an integral over the LJSD $\mu$ in the limit. While this might in general lead to a complicated coupling of many transcendental equations, it turns out that the trascendentality can be entirely factored into a single scalar fixed-point equation, whose solution we denote by $x$ (see Eq. (A345)), and the remaining equations are purely algebraic given $x$. To facilitate this particular simplification, it is necessary to first compute all of the entries in the operator-valued Stieltjes transform of the kernel matrix $K$, which we do in Sec. A9.6.1. Using these results, we compute the remaining error terms in the subsequent sections.

As a matter of notation, note that throughout this entire section whenever a matrix $X$, $X_1$, or $X_2$ appears it is composed of i.i.d. standard Gaussian entries as in Sec. A9.5. This differs from the notation of the main paper, but we follow this prescription to ease the already cumbersome presentation. This definition of $X$ allows us to explicitly extract and represent the training covariance $\Sigma$ in our calculations.

## A9.6.1 $K^{-1}$

Define the block matrix $Q^{K^{-1}}$ as,

$$Q^{K^{-1}} = \begin{pmatrix} I_m & \frac{\sqrt{\eta-\zeta}\Theta^\top}{\gamma\sqrt{n_1}} & \frac{\sqrt{\rho}X^\top}{\gamma\sqrt{n_0}} & 0 & 0 & 0 \\ -\frac{\Theta\sqrt{\eta-\zeta}}{\sqrt{n_1}} & I_{n_1} & 0 & 0 & -\frac{\sqrt{\rho}W}{\sqrt{n_1}} & 0 \\ 0 & 0 & I_{n_0} & -\Sigma^{1/2} & 0 & 0 \\ 0 & -\frac{W^\top}{\sqrt{n_1}} & 0 & I_{n_0} & 0 & 0 \\ 0 & 0 & 0 & 0 & I_{n_0} & -\Sigma^{1/2} \\ -\frac{X}{\sqrt{n_0}} & 0 & 0 & 0 & 0 & I_{n_0} \end{pmatrix}. \tag{A303}$$

Then block matrix inversion (i.e. repeated applications of the Schur complement formula) shows that,

$$G_{1,1}^{K^{-1}} = \gamma \bar{\mathrm{tr}}(K^{-1}) \tag{A304}$$

$$G_{2,2}^{K^{-1}} = \gamma \bar{\mathrm{tr}}(\hat{K}^{-1}) \tag{A305}$$

$$G_{3,3}^{K^{-1}} = G_{6,6}^{K^{-1}} = 1 - \frac{\sqrt{\rho}\bar{\mathrm{tr}}\left(\Sigma^{1/2}W^\top F K^{-1} X^\top\right)}{\sqrt{n_0}n_1} \tag{A306}$$

$$G_{4,3}^{K^{-1}} = G_{6,5}^{K^{-1}} = \bar{\mathrm{tr}}(\Sigma^{1/2}) - \frac{\sqrt{\rho}\bar{\mathrm{tr}}\left(\Sigma W^\top F K^{-1} X^\top\right)}{\sqrt{n_0}n_1} \tag{A307}$$

$$G_{5,3}^{K^{-1}} = G_{6,4}^{K^{-1}} = \frac{\gamma\sqrt{\rho}\bar{\mathrm{tr}}\left(\Sigma^{1/2}W^\top \hat{K}^{-1} W\right)}{n_1} \tag{A308}$$

$$G_{6,3}^{K^{-1}} = \frac{\gamma\sqrt{\rho}\bar{\mathrm{tr}}\left(\Sigma W^\top \hat{K}^{-1} W\right)}{n_1} \tag{A309}$$

$$G_{3,4}^{K^{-1}} = G_{5,6}^{K^{-1}} = -\frac{\sqrt{\rho}\bar{\mathrm{tr}}\left(F K^{-1} X^\top W^\top\right)}{\sqrt{n_0}n_1\psi} \tag{A310}$$

$$G_{4,4}^{K^{-1}} = G_{5,5}^{K^{-1}} = 1 - \frac{\sqrt{\rho}\bar{\mathrm{tr}}\left(\Sigma^{1/2}W^\top F K^{-1} X^\top\right)}{\sqrt{n_0}n_1} \tag{A311}$$

$$G_{5,4}^{K^{-1}} = \frac{\gamma\sqrt{\rho}\bar{\mathrm{tr}}\left(\hat{K}^{-1}WW^\top\right)}{n_1\psi} \tag{A312}$$

$$G_{3,5}^{K^{-1}} = G_{4,6}^{K^{-1}} = -\frac{\sqrt{\rho}\bar{\mathrm{tr}}\left(\Sigma^{1/2} X K^{-1} X^\top\right)}{n_0} \tag{A313}$$

$$G_{4,5}^{K^{-1}} = -\frac{\sqrt{\rho}\bar{\mathrm{tr}}\left(\Sigma X K^{-1} X^\top\right)}{n_0} \tag{A314}$$

$$G_{3,6}^{K^{-1}} = -\frac{\sqrt{\rho}\bar{\mathrm{tr}}\left(K^{-1} X^\top X\right)}{n_0\phi}, \tag{A315}$$

where $G^{K^{-1}} := \mathrm{id}_6 \otimes \bar{\mathrm{tr}}\,[(Q^{K^{-1}})^\top]^{-1} \in M_6(\mathbb{C})$ is a scalar $6 \times 6$ matrix whose $i,j$ entry $G_{i,j}^{K^{-1}}$ is the normalized trace of the $(i,j)$-block of the inverse of $[Q^{K^{-1}}]^\top$, and we have defined $\hat{K} = \frac{1}{n_1}FF^\top + \gamma I_{n_1}$.

We aim to compute the limiting values of these trace terms as $n_0, n_1, m \to \infty$, as they will be related to the error terms of interest. Both here and in the sequel, to ease the already cumbersome presentation, we use $G$ to denote the limiting values as well as the non-limiting values and we refrain from explicitly denoting the limit operation itself, noting its existing can be inferred from context.

To proceed, recall that the asymptotic block-wise traces of the inverse of $Q^{K^{-1}}$ can be determined from its operator-valued Stieltjes transform [45]. The simplest way to apply the results of [19, 45] is to augment $Q^{K^{-1}}$ to form the the self-adjoint matrix $\bar{Q}^{K^{-1}}$,

$$\bar{Q}^{K^{-1}} = \begin{pmatrix} 0 & [Q^{K^{-1}}]^\top \\ Q^{K^{-1}} & 0 \end{pmatrix}, \tag{A316}$$

and observe that we can write $\bar{Q}^{K^{-1}}$ as,

$$
\bar{Q}^{K^{-1}} = \bar{Z} - \bar{Q}^{K^{-1}}_{W,X,\Theta} - \bar{Q}^{K^{-1}}_{\Sigma}
$$

$$
= \begin{pmatrix} 0 & Z^\top \\ Z & 0 \end{pmatrix} - \begin{pmatrix} 0 & [Q^{K^{-1}}_{W,X,\Theta}]^\top \\ Q^{K^{-1}}_{W,X,\Theta} & 0 \end{pmatrix} - \begin{pmatrix} 0 & [Q^{K^{-1}}_{\Sigma}]^\top \\ Q^{K^{-1}}_{\Sigma} & 0 \end{pmatrix} , \tag{A317}
$$

where $Z = I_{m+4n_0+n_1}$, and,

$$
Q^{K^{-1}}_{W,X,\Theta} = \begin{pmatrix}
0 & \frac{\sqrt{\eta-\zeta}\Theta^\top}{\gamma\sqrt{n_1}} & \frac{\sqrt{\rho}X^\top}{\gamma\sqrt{n_0}} & 0 & 0 & 0 \\
-\frac{\Theta\sqrt{\eta-\zeta}}{\sqrt{n_1}} & 0 & 0 & 0 & -\frac{\sqrt{\rho}W}{\sqrt{n_1}} & 0 \\
0 & 0 & 0 & 0 & 0 & 0 \\
0 & -\frac{W^\top}{\sqrt{n_1}} & 0 & 0 & 0 & 0 \\
0 & 0 & 0 & 0 & 0 & 0 \\
-\frac{X}{\sqrt{n_0}} & 0 & 0 & 0 & 0 & 0
\end{pmatrix} \tag{A318}
$$

$$
Q^{K^{-1}}_{\Sigma} = \begin{pmatrix}
0 & 0 & 0 & 0 & 0 & 0 \\
0 & 0 & 0 & 0 & 0 & 0 \\
0 & 0 & 0 & -\Sigma^{1/2} & 0 & 0 \\
0 & 0 & 0 & 0 & 0 & 0 \\
0 & 0 & 0 & 0 & 0 & -\Sigma^{1/2} \\
0 & 0 & 0 & 0 & 0 & 0
\end{pmatrix} . \tag{A319}
$$

Note that we have separated the i.i.d. Gaussian matrices $W, X, \Theta$ from the constant terms and from the $\Sigma$-dependent terms. Denote by $\bar{G}^{K^{-1}} \in M_{12}(\mathbb{C})$ the block matrix

$$
\bar{G}^{K^{-1}} = \begin{pmatrix} 0 & [G^{K^{-1}}]^\top \\ G^{K^{-1}} & 0 \end{pmatrix} = \mathrm{id}_{12} \otimes \bar{\mathrm{tr}}\left(\bar{Q}^{K^{-1}}\right)^{-1} , \tag{A320}
$$

and by $\bar{G}^{K^{-1}}_{\Sigma} \in M_{12}(\mathbb{C})$ the operator-valued Stieltjes transform of $\bar{Q}^{K^{-1}}_{\Sigma}$. Using Eq. (A317) and the definition of the operator-valued Stieltjes transform $G_{\bar{Q}^{K^{-1}}_{W,X,\Theta} + \bar{Q}^{K^{-1}}_{\Sigma}}$, we can write

$$
\bar{G}^{K^{-1}} = \mathrm{id}_{12} \otimes \bar{\mathrm{tr}}\left(\bar{Z} - \bar{Q}^{K^{-1}}_{W,X,\Theta} - \bar{Q}^{K^{-1}}_{\Sigma}\right)^{-1} = G_{\bar{Q}^{K^{-1}}_{W,X,\Theta} + \bar{Q}^{K^{-1}}_{\Sigma}}(\bar{Z}) . \tag{A321}
$$

Thus using the subordinated form of the equations for addition of free variables (45; section 9.2 Thm. 11), and the defining equation for $\bar{G}^{K^{-1}}_{\Sigma}$, the operator-valued theory of free probability shows that in the limit $n_0, n_1, m \to \infty$, the Stieltjes transform $\bar{G}^{K^{-1}}$ satisfies the following $12 \times 12$ matrix equation,

$$
\bar{G}^{K^{-1}} = \bar{G}^{K^{-1}}_{\Sigma}(\bar{Z} - \bar{R}^{K^{-1}}_{W,X,\Theta}(\bar{G}^{K^{-1}}))
$$

$$
= \mathrm{id} \otimes \bar{\mathrm{tr}}\left(\bar{Z} - \bar{R}^{K^{-1}}_{W,X,\Theta}(\bar{G}^{K^{-1}}) - \bar{Q}^{K^{-1}}_{\Sigma}\right)^{-1} , \tag{A322}
$$

where $\bar{R}^{K^{-1}}_{W,X,\Theta}(\bar{G}^{K^{-1}}) \in M_{12}(\mathbb{C})$ is the operator-valued R-transform of $\bar{Q}^{K^{-1}}_{W,X,\Theta}$. Here the normalized trace $\bar{\mathrm{tr}}$ acts on the constituent blocks, and the identity operator id acts on the space of $12 \times 12$ matrices. As described by [2, 3], since $\bar{Q}^{K^{-1}}_{W,X,\Theta}$ is a block matrix whose blocks are i.i.d. Gaussian matrices (and their transposes), an explicit expression for $\bar{R}^{K^{-1}}_{W,X,\Theta}(\bar{G}^{K^{-1}})$ can be obtained through a covariance map, denoted by $\eta$ [19]. In particular, $\eta : M_d(\mathbb{C}) \to M_d(\mathbb{C})$ is defined by,

$$
[\eta(D)]_{ij} = \sum_{kl} \sigma(i, k; l, j)\alpha_k D_{kl} , \tag{A323}
$$

where $\alpha_k$ is dimensionality of the $k$th block and $\sigma(i, k; l, k)$ denotes the covariance between the entries of the blocks $ij$ block of $\bar{Q}^{K^{-1}}_{W,X,\Theta}$ and entries of the $kl$ block of $\bar{Q}^{K^{-1}}_{W,X,\Theta}$. Here $d = 12$ is the number of blocks. When the constituent blocks are i.i.d. Gaussian matrices and their transposes, as is the case here, then $\bar{R}^{K^{-1}}_{W,X,\Theta} = \eta$ [45], and therefore the entries of $\bar{R}^{K^{-1}}_{W,X,\Theta}$ can be read off

from Eq. (A316). To simplify the presentation, we only report the entries of $\bar{R}_{W,X,\Theta}^{K^{-1}}(G^{K^{-1}})$ that are nonzero, given the specific sparsity pattern of $G^{K^{-1}}$. The latter follows from Eq. (A322) in the manner described in [45, 19]. Practically speaking, the sparsity pattern can be obtained by iterating an Eq. (A322), starting with an ansatz sparsity pattern determined by $\bar{Z}$, and stopping when the iteration converges to a fixed sparsity pattern. In this case (and all cases that follow in the subsequent sections), the number of necessary iterations is small and can be done explicitly. We omit the details and instead simply report the following results for the nonzero entries:

$$\bar{R}_{W,X,\Theta}^{K^{-1}}(\bar{G}^{K^{-1}}) = \begin{pmatrix} 0 & R_{W,X,\Theta}^{K^{-1}}(G^{K^{-1}})^{\top} \\ R_{W,X,\Theta}^{K^{-1}}(G^{K^{-1}}) & 0 \end{pmatrix}, \tag{A324}$$

where,

$$[R_{W,X,\Theta}^{K^{-1}}(G^{K^{-1}})]_{1,1} = \frac{G_{2,2}^{K^{-1}}(\zeta - \eta) - \sqrt{\rho}G_{6,3}^{K^{-1}}}{\gamma} \tag{A325}$$

$$[R_{W,X,\Theta}^{K^{-1}}(G^{K^{-1}})]_{2,2} = \frac{\psi G_{1,1}^{K^{-1}}(\zeta - \eta)}{\gamma\phi} + \sqrt{\rho}\psi G_{4,5}^{K^{-1}} \tag{A326}$$

$$[R_{W,X,\Theta}^{K^{-1}}(G^{K^{-1}})]_{4,5} = \sqrt{\rho}G_{2,2}^{K^{-1}} \tag{A327}$$

$$[R_{W,X,\Theta}^{K^{-1}}(G^{K^{-1}})]_{6,3} = -\frac{\sqrt{\rho}G_{1,1}^{K^{-1}}}{\gamma\phi}, \tag{A328}$$

and the remaining entries of $R_{W,X,\Theta}^{K^{-1}}(G^{K^{-1}})$ are zero. Owing to the large degree of sparsity, the matrix inverse in Eq. (A322) can be performed explicitly and yields relatively simple expressions that depend on the entries of $G^{K^{-1}}$ and the matrix $\Sigma$. For example, the $(9,6)$ entry of the self-consistent equation reads,

$$G_{3,6}^{K^{-1}} = \left[\mathrm{id} \otimes \bar{\mathrm{tr}}\left(\bar{Z} - \bar{R}_{W,X,\Theta}^{K^{-1}}(\bar{G}^{K^{-1}}) - \bar{Q}_{\Sigma}^{K^{-1}}\right)^{-1}\right]_{9,6} \tag{A329}$$

$$= \bar{\mathrm{tr}}\left[\sqrt{\rho}G_{1,1}^{K^{-1}}\left(-\Sigma\rho G_{1,1}^{K^{-1}}G_{2,2}^{K^{-1}} - \gamma\phi I_{n_0}\right)^{-1}\right] \tag{A330}$$

$$\overset{n_0 \to \infty}{=} \mathbb{E}_\mu\left[\frac{\sqrt{\rho}G_{1,1}^{K^{-1}}}{-\lambda\rho G_{1,1}^{K^{-1}}G_{2,2}^{K^{-1}} - \gamma\phi}\right], \tag{A331}$$

where to compute the asymptotic normalized trace we moved to an eigenbasis of $\Sigma$ and recalled the definition of the LJSD $\mu$. The remaining entries of the Eq. (A322) can be obtained in a similar manner and together yield the following set of coupled equations for the entries of $G^{K^{-1}}$,

$$G_{1,1}^{K^{-1}} = -\frac{\gamma}{-G_{2,2}^{K^{-1}}(-\zeta + \eta + \rho) + \rho G_{2,2}^{K^{-1}} - \sqrt{\rho}G_{6,3}^{K^{-1}} - \gamma} \tag{A332}$$

$$G_{2,2}^{K^{-1}} = \frac{\gamma\phi}{\psi G_{1,1}^{K^{-1}}(\eta - \zeta) - \gamma\phi\left(\sqrt{\rho}\psi G_{4,5}^{K^{-1}} - 1\right)} \tag{A333}$$

$$G_{3,6}^{K^{-1}} = \mathbb{E}_\mu\left[\frac{\sqrt{\rho}G_{1,1}^{K^{-1}}}{-\lambda\rho G_{1,1}^{K^{-1}}G_{2,2}^{K^{-1}} - \gamma\phi}\right] \tag{A334}$$

$$G_{4,5}^{K^{-1}} = \mathbb{E}_\mu\left[\frac{\lambda\sqrt{\rho}G_{1,1}^{K^{-1}}}{-\lambda\rho G_{1,1}^{K^{-1}}G_{2,2}^{K^{-1}} - \gamma\phi}\right] \tag{A335}$$

$$G_{5,4}^{K^{-1}} = \mathbb{E}_\mu\left[-\frac{\gamma\sqrt{\rho}\phi G_{2,2}^{K^{-1}}}{-\lambda\rho G_{1,1}^{K^{-1}}G_{2,2}^{K^{-1}} - \gamma\phi}\right] \tag{A336}$$

$$G_{6,3}^{K^{-1}} = \mathbb{E}_\mu\left[-\frac{\gamma\lambda\sqrt{\rho}\phi G_{2,2}^{K^{-1}}}{-\lambda\rho G_{1,1}^{K^{-1}}G_{2,2}^{K^{-1}} - \gamma\phi}\right] \tag{A337}$$

$$G_{3,4}^{K^{-1}} = G_{5,6}^{K^{-1}} = \mathbb{E}_\mu\left[\frac{\sqrt{\lambda}\rho G_{1,1}^{K^{-1}}G_{2,2}^{K^{-1}}}{-\lambda\rho G_{1,1}^{K^{-1}}G_{2,2}^{K^{-1}} - \gamma\phi}\right] \tag{A338}$$

$$G_{3,5}^{K^{-1}} \;=\; G_{4,6}^{K^{-1}} = \mathbb{E}_\mu\left[-\frac{G_{1,1}^{K^{-1}}\sqrt{\lambda\rho}}{\lambda\rho G_{1,1}^{K^{-1}}G_{2,2}^{K^{-1}} + \gamma\phi}\right] \tag{A339}$$

$$G_{4,3}^{K^{-1}} \;=\; G_{6,5}^{K^{-1}} = \mathbb{E}_\mu\left[-\frac{\gamma\sqrt{\lambda}\phi}{-\lambda\rho G_{1,1}^{K^{-1}}G_{2,2}^{K^{-1}} - \gamma\phi}\right] \tag{A340}$$

$$G_{5,3}^{K^{-1}} \;=\; G_{6,4}^{K^{-1}} = \mathbb{E}_\mu\left[\frac{\gamma\phi G_{2,2}^{K^{-1}}\sqrt{\lambda\rho}}{\lambda\rho G_{1,1}^{K^{-1}}G_{2,2}^{K^{-1}} + \gamma\phi}\right] \tag{A341}$$

$$G_{3,3}^{K^{-1}} \;=\; G_{4,4}^{K^{-1}} = G_{5,5}^{K^{-1}} = G_{6,6}^{K^{-1}} = \mathbb{E}_\mu\left[\frac{\gamma\phi}{\lambda\rho G_{1,1}^{K^{-1}}G_{2,2}^{K^{-1}} + \gamma\phi}\right], \tag{A342}$$

where we have used the fact that, asymptotically, the normalized trace becomes equivalent to an expectation over $\mu$. After eliminating $G_{6,3}^{K^{-1}}$ and $G_{4,5}^{K^{-1}}$ from the first two equations, it is straightforward to show that

$$\tau \equiv \bar{\mathrm{tr}}(K^{-1}) = \frac{1}{\gamma}G_{1,1}^{K^{-1}} = \frac{\sqrt{(\psi-\phi)^2 + 4x\psi\phi\gamma/\rho} + \psi - \phi}{2\psi\gamma} \tag{A343}$$

$$\bar{\tau} \equiv \bar{\mathrm{tr}}(\hat{K}^{-1}) = \frac{1}{\gamma}G_{2,2}^{K^{-1}} = \frac{1}{\gamma} + \frac{\psi}{\phi}\left(\tau - \frac{1}{\gamma}\right) \tag{A344}$$

where $\bar{\tau}$ is the companion transform of $\tau$, and where $x$ satisfies the self-consistent equation,

$$x = \frac{1 - \gamma\tau}{\omega + I_{1,1}} = \frac{1 - \frac{\sqrt{(\psi-\phi)^2 + 4x\psi\phi\gamma/\rho} + \psi - \phi}{2\psi}}{\omega + \mathcal{I}_{1,1}}. \tag{A345}$$

Here we used the two-index set of functionals of $\mu$, $\mathcal{I}_{a,b}$ defined in Eq. (14).

Note that the product $\tau\bar{\tau}$ is simply related to $x$,

$$x = \gamma\rho\tau\bar{\tau}, \tag{A346}$$

so that, given $x$, the equations for the remaining entries of $G^{K^{-1}}$ completely decouple. In particular,

$$G_{3,6}^{K^{-1}} \;=\; -\frac{\sqrt{\rho}\tau\mathcal{I}_{0,1}}{\phi} \tag{A347}$$

$$G_{4,5}^{K^{-1}} \;=\; -\frac{\sqrt{\rho}\tau\mathcal{I}_{1,1}}{\phi} \tag{A348}$$

$$G_{5,4}^{K^{-1}} \;=\; \gamma\sqrt{\rho}\bar{\tau}\mathcal{I}_{0,1} \tag{A349}$$

$$G_{6,3}^{K^{-1}} \;=\; \gamma\sqrt{\rho}\bar{\tau}\mathcal{I}_{1,1} \tag{A350}$$

$$G_{3,4}^{K^{-1}} \;=\; G_{5,6}^{K^{-1}} = -\frac{x\mathcal{I}_{\frac{1}{2},1}}{\phi} \tag{A351}$$

$$G_{3,5}^{K^{-1}} \;=\; G_{4,6}^{K^{-1}} = -\frac{\sqrt{\rho}\tau\mathcal{I}_{\frac{1}{2},1}}{\phi} \tag{A352}$$

$$G_{4,3}^{K^{-1}} \;=\; G_{6,5}^{K^{-1}} = \mathcal{I}_{\frac{1}{2},1} \tag{A353}$$

$$G_{5,3}^{K^{-1}} \;=\; G_{6,4}^{K^{-1}} = \gamma\sqrt{\rho}\bar{\tau}\mathcal{I}_{\frac{1}{2},1} \tag{A354}$$

$$G_{3,3}^{K^{-1}} \;=\; G_{4,4}^{K^{-1}} = G_{5,5}^{K^{-1}} = G_{6,6}^{K^{-1}} = \mathcal{I}_{0,1}, \tag{A355}$$

which will be important intermediate results for the subsequent sections.

## A9.6.2 $E_{21}$

Define the block matrix $Q^{E_{21}}$ as,

$$Q^{E_{21}} = \begin{pmatrix} I_{n_0} & 0 & -\Sigma^{1/2} & 0 & 0 & 0 & 0 & 0 & 0 \\ -\frac{X^\top}{\sqrt{n_0}} & I_m & 0 & 0 & 0 & 0 & 0 & 0 & 0 \\ 0 & 0 & I_{n_0} & -\Sigma^* & 0 & 0 & 0 & 0 & 0 \\ 0 & 0 & 0 & I_{n_0} & -\frac{W^\top}{\sqrt{n_1}} & 0 & 0 & 0 & 0 \\ 0 & 0 & 0 & 0 & I_{n_1} & -\frac{\Theta\sqrt{\eta-\zeta}}{\sqrt{n_1}} & -\frac{\sqrt{\rho}W}{\sqrt{n_1}} & 0 & 0 \\ 0 & 0 & 0 & 0 & \frac{\sqrt{\eta-\zeta}\Theta^\top}{\gamma\sqrt{n_1}} & I_m & 0 & 0 & \frac{\sqrt{\rho}X^\top}{\gamma\sqrt{n_0}} \\ 0 & 0 & 0 & 0 & 0 & 0 & I_{n_0} & -\Sigma^{1/2} & 0 \\ 0 & 0 & 0 & 0 & 0 & 0 & -\frac{X}{\sqrt{n_0}} & 0 & I_{n_0} & 0 \\ 0 & 0 & 0 & -\Sigma^{1/2} & 0 & 0 & 0 & 0 & I_{n_0} \end{pmatrix}. \tag{A356}$$

Then block matrix inversion (i.e. repeated applications of the Schur complement formula) shows that,

$$G_{1,1}^{E_{21}} = G_{2,2}^{E_{21}} = G_{3,3}^{E_{21}} = 1 \tag{A357}$$

$$G_{6,6}^{E_{21}} = G_{1,1}^{K^{-1}} \tag{A358}$$

$$G_{5,5}^{E_{21}} = G_{2,2}^{K^{-1}} \tag{A359}$$

$$G_{4,4}^{E_{21}} = G_{7,7}^{E_{21}} = G_{8,8}^{E_{21}} = G_{9,9}^{E_{21}} = G_{3,3}^{K^{-1}} \tag{A360}$$

$$G_{7,8}^{E_{21}} = G_{9,4}^{E_{21}} = G_{3,4}^{K^{-1}} \tag{A361}$$

$$G_{4,8}^{E_{21}} = G_{9,7}^{E_{21}} = G_{3,5}^{K^{-1}} \tag{A362}$$

$$G_{9,8}^{E_{21}} = G_{3,6}^{K^{-1}} \tag{A363}$$

$$G_{4,9}^{E_{21}} = G_{8,7}^{E_{21}} = G_{4,3}^{K^{-1}} \tag{A364}$$

$$G_{4,7}^{E_{21}} = G_{4,5}^{K^{-1}} \tag{A365}$$

$$G_{7,9}^{E_{21}} = G_{8,4}^{E_{21}} = G_{5,3}^{K^{-1}} \tag{A366}$$

$$G_{7,4}^{E_{21}} = G_{5,4}^{K^{-1}} \tag{A367}$$

$$G_{8,9}^{E_{21}} = G_{6,3}^{K^{-1}} \tag{A368}$$

$$G_{3,1}^{E_{21}} = \bar{\text{tr}}(\Sigma^{1/2}) \tag{A369}$$

$$G_{7,3}^{E_{21}} = \frac{\gamma\sqrt{\rho}\,\bar{\text{tr}}\left(\hat{K}^{-1}W\Sigma^*W^\top\right)}{n_1\psi} \tag{A370}$$

$$G_{7,1}^{E_{21}} = G_{8,3}^{E_{21}} = \frac{\gamma\sqrt{\rho}\,\bar{\text{tr}}\left(\Sigma^{1/2}W^\top\hat{K}^{-1}W\Sigma^*\right)}{n_1} \tag{A371}$$

$$G_{9,3}^{E_{21}} = -\frac{\sqrt{\rho}\,\bar{\text{tr}}\left(FK^{-1}X^\top\Sigma^*W^\top\right)}{\sqrt{n_0}n_1\psi} \tag{A372}$$

$$G_{8,1}^{E_{21}} = \frac{\gamma\sqrt{\rho}\,\bar{\text{tr}}\left(\Sigma W^\top\hat{K}^{-1}W\Sigma^*\right)}{n_1} \tag{A373}$$

$$G_{9,1}^{E_{21}} = -\frac{\sqrt{\rho}\,\bar{\text{tr}}\left(\Sigma^{1/2}XF^\top\hat{K}^{-1}W\Sigma^*\right)}{\sqrt{n_0}n_1} \tag{A374}$$

$$G_{6,2}^{E_{21}} = \frac{\gamma\phi\,\bar{\text{tr}}\left(\Sigma^{1/2}XF^\top\hat{K}^{-1}W\Sigma^*\right)}{\sqrt{n_0}n_1} \tag{A375}$$

$$G_{4,3}^{E_{21}} = \bar{\text{tr}}(\Sigma^*) - \frac{\sqrt{\rho}\,\bar{\text{tr}}\left(\Sigma^{1/2}XF^\top\hat{K}^{-1}W\Sigma^*\right)}{\sqrt{n_0}n_1} \tag{A376}$$

$$G_{4,1}^{E_{21}} = \bar{\text{tr}}(\Sigma^{1/2}\Sigma^*) - \frac{\sqrt{\rho}\,\bar{\text{tr}}\left(\Sigma X F^\top \hat{K}^{-1} W \Sigma^*\right)}{\sqrt{n_0}n_1}, \tag{A377}$$

where $G_{i,j}^{E_{21}}$ denotes the normalized trace of the $(i,j)$-block of the inverse of $\left(Q^{E_{21}}\right)^\top$. Comparing to Eq. (A280), we see that the error term $E_{21}$ is related to $G_{6,2}^{E_{21}}$ by

$$E_{21} = -\frac{2\sqrt{\rho}}{\gamma\phi} G_{6,2}^{E_{21}}. \tag{A378}$$

To compute the limiting values of these traces, we require the asymptotic block-wise traces of $Q^{E_{21}}$, which may be determined from the operator-valued Stieltjes transform. Proceeding as above, we first augment $\bar{Q}^{E_{21}}$ to form the the self-adjoint matrix $\bar{Q}^{E_{21}}$,

$$\bar{Q}^{E_{21}} = \begin{pmatrix} 0 & [Q^{E_{21}}]^\top \\ Q^{E_{21}} & 0 \end{pmatrix}. \tag{A379}$$

and observe that we can write $\bar{Q}^{E_{21}}$ as

$$\begin{aligned} \bar{Q}^{E_{21}} &= \bar{Z} - \bar{Q}_{W,X,\Theta}^{E_{21}} - \bar{Q}_\Sigma^{E_{21}} \\ &= \begin{pmatrix} 0 & Z^\top \\ Z & 0 \end{pmatrix} - \begin{pmatrix} 0 & [Q_{W,X,\Theta}^{E_{21}}]^\top \\ Q_{W,X,\Theta}^{E_{21}} & 0 \end{pmatrix} - \begin{pmatrix} 0 & [Q_\Sigma^{E_{21}}]^\top \\ Q_\Sigma^{E_{21}} & 0 \end{pmatrix}, \end{aligned} \tag{A380}$$

where $Z = I_{2m+6n_0+n_1}$, and,

$$Q_{W,X,\Theta}^{E_{21}} = \begin{pmatrix} 0 & 0 & 0 & 0 & 0 & 0 & 0 & 0 & 0 \\ -\frac{X^\top}{\sqrt{n_0}} & 0 & 0 & 0 & 0 & 0 & 0 & 0 & 0 \\ 0 & 0 & 0 & 0 & 0 & 0 & 0 & 0 & 0 \\ 0 & 0 & 0 & 0 & -\frac{W^\top}{\sqrt{n_1}} & 0 & 0 & 0 & 0 \\ 0 & 0 & 0 & 0 & 0 & -\frac{\Theta\sqrt{\eta-\zeta}}{\sqrt{n_1}} & -\frac{\sqrt{\rho}W}{\sqrt{n_1}} & 0 & 0 \\ 0 & 0 & 0 & 0 & \frac{\sqrt{\eta-\zeta}\Theta^\top}{\gamma\sqrt{n_1}} & 0 & 0 & 0 & \frac{\sqrt{\rho}X^\top}{\gamma\sqrt{n_0}} \\ 0 & 0 & 0 & 0 & 0 & 0 & 0 & 0 & 0 \\ 0 & 0 & 0 & 0 & 0 & -\frac{X}{\sqrt{n_0}} & 0 & 0 & 0 \\ 0 & 0 & 0 & 0 & 0 & 0 & 0 & 0 & 0 \end{pmatrix} \tag{A381}$$

$$Q_\Sigma^{E_{21}} = \begin{pmatrix} 0 & 0 & -\Sigma^{1/2} & 0 & 0 & 0 & 0 & 0 & 0 \\ 0 & 0 & 0 & 0 & 0 & 0 & 0 & 0 & 0 \\ 0 & 0 & 0 & -\Sigma^* & 0 & 0 & 0 & 0 & 0 \\ 0 & 0 & 0 & 0 & 0 & 0 & 0 & 0 & 0 \\ 0 & 0 & 0 & 0 & 0 & 0 & 0 & 0 & 0 \\ 0 & 0 & 0 & 0 & 0 & 0 & 0 & 0 & 0 \\ 0 & 0 & 0 & 0 & 0 & 0 & 0 & -\Sigma^{1/2} & 0 \\ 0 & 0 & 0 & 0 & 0 & 0 & 0 & 0 & 0 \\ 0 & 0 & 0 & -\Sigma^{1/2} & 0 & 0 & 0 & 0 & 0 \end{pmatrix}. \tag{A382}$$

The operator-valued Stieltjes transforms satisfy,

$$\begin{aligned} \bar{G}^{E_{21}} &= \bar{G}_\Sigma^{E_{21}}(\bar{Z} - \bar{R}_{W,X,\Theta}^{E_{21}}(\bar{G}^{E_{21}})) \\ &= \text{id} \otimes \bar{\text{tr}}\left(\bar{Z} - \bar{R}_{W,X,\Theta}^{E_{21}}(\bar{G}^{E_{21}}) - \bar{Q}_\Sigma^{E_{21}}\right)^{-1}, \end{aligned} \tag{A383}$$

where $\bar{R}_{W,X,\Theta}^{E_{21}}(\bar{G}^{E_{21}})$ is the operator-valued R-transform of $\bar{Q}_{W,X,\Theta}^{E_{21}}$. As discussed above, since $\bar{Q}_{W,X,\Theta}^{E_{21}}$ is a block matrix whose blocks are i.i.d. Gaussian matrices (and their transposes), an explicit expression for $\bar{R}_{W,X,\Theta}^{E_{21}}(\bar{G}^{E_{21}})$ can be obtained from the covariance map $\eta$, which can be read off from Eq. (A379). As above, we use the specific sparsity pattern for $G^{E_{21}}$ that is induced by Eq. (A383), to obtain,

$$\bar{R}_{W,X,\Theta}^{E_{21}}(\bar{G}^{E_{21}}) = \begin{pmatrix} 0 & R_{W,X,\Theta}^{E_{21}}(G^{E_{21}})^\top \\ R_{W,X,\Theta}^{E_{21}}(G^{E_{21}}) & 0 \end{pmatrix}, \tag{A384}$$

where,

$$[R_{W,X,\Theta}^{E_{21}}(G^{E_{21}})]_{2,6} \quad = \quad G_{8,1}^{E_{21}} \tag{A385}$$

$$[R_{W,X,\Theta}^{E_{21}}(G^{E_{21}})]_{4,7} \quad = \quad \sqrt{\rho}G_{5,5}^{E_{21}} \tag{A386}$$

$$[R_{W,X,\Theta}^{E_{21}}(G^{E_{21}})]_{5,5} \quad = \quad \frac{\psi G_{6,6}^{E_{21}}(\zeta - \eta)}{\gamma\phi} + \sqrt{\rho}\psi G_{4,7}^{E_{21}} \tag{A387}$$

$$[R_{W,X,\Theta}^{E_{21}}(G^{E_{21}})]_{6,6} \quad = \quad \frac{G_{5,5}^{E_{21}}(\zeta - \eta) - \sqrt{\rho}G_{8,9}^{E_{21}}}{\gamma} \tag{A388}$$

$$[R_{W,X,\Theta}^{E_{21}}(G^{E_{21}})]_{8,1} \quad = \quad \frac{G_{2,6}^{E_{21}}}{\phi} \tag{A389}$$

$$[R_{W,X,\Theta}^{E_{21}}(G^{E_{21}})]_{8,9} \quad = \quad -\frac{\sqrt{\rho}G_{6,6}^{E_{21}}}{\gamma\phi} \ , \tag{A390}$$

and the remaining entries of $R_{W,X,\Theta}^{E_{21}}(G^{E_{21}})$ are zero.

Owing to the large degree of sparsity, the matrix inverse in Eq. (A383) can be performed explicitly and yields relatively simple expressions that depend on the entries of $G^{E_{21}}$ and the matrices $\Sigma$ and $\Sigma^*$. For example, the $(13, 3)$ entry of the self-consistent equation reads,

$$G_{4,3}^{E_{21}} \quad = \quad \left[\mathrm{id} \otimes \bar{\mathrm{tr}}\left(\bar{Z} - \bar{R}_{W,X,\Theta}^{E_{21}}(\bar{G}^{E_{21}}) - \bar{Q}_{\Sigma}^{E_{21}}\right)^{-1}\right]_{13,3} \tag{A391}$$

$$= \quad \bar{\mathrm{tr}}\left[\Sigma^*\left(I_{n_0} + \frac{\rho}{\gamma\phi}G_{5,5}^{E_{21}}G_{6,6}^{E_{21}}\Sigma\right)^{-1}\right] \tag{A392}$$

$$\stackrel{n_0 \to \infty}{=} \quad \mathbb{E}_\mu\left[\frac{r}{1 + \frac{\rho}{\gamma\phi}\lambda G_{5,5}^{E_{21}}G_{6,6}^{E_{21}}}\right] \tag{A393}$$

$$= \quad \mathbb{E}_\mu\left[\frac{r}{1 + \frac{\rho}{\gamma\phi}\lambda G_{1,1}^{K^{-1}}G_{2,2}^{K^{-1}}}\right] \tag{A394}$$

$$= \quad \phi\mathbb{E}_\mu\left[\frac{r}{\phi + x\lambda}\right] \tag{A395}$$

$$= \quad \mathcal{I}_{1,1}^* \ . \tag{A396}$$

To obtain Eq. (A393), we computed the asymptotic normalized trace by moving to an eigenbasis of $\Sigma$ and recalling the definition of the LJSD $\mu$. We also used Eqs. (A359) and (A358) to obtain Eq. (A394) and Eqs. (A343), (A344), and (A346) to obtain Eq. (A395). The final line follows from the definition of $\mathcal{I}^*$ in Eq. (14). The remaining nonzero entries of Eq. (A383) can be obtained in a similar manner and together yield the following set of coupled equations for the entries of $G^{E_{21}}$,

$$G_{3,1}^{E_{21}} \quad = \quad \frac{\mathcal{I}_{\frac{1}{2},0}}{\phi} \tag{A397}$$

$$G_{4,1}^{E_{21}} \quad = \quad \mathcal{I}_{\frac{3}{2},1}^* \tag{A398}$$

$$G_{4,3}^{E_{21}} \quad = \quad \mathcal{I}_{1,1}^* \tag{A399}$$

$$G_{6,2}^{E_{21}} \quad = \quad \gamma\tau G_{8,1}^{E_{21}} \tag{A400}$$

$$G_{7,3}^{E_{21}} \quad = \quad \gamma\sqrt{\rho}\bar{\tau}\mathcal{I}_{1,1}^* \tag{A401}$$

$$G_{8,1}^{E_{21}} \quad = \quad \gamma\sqrt{\rho}\bar{\tau}\mathcal{I}_{2,1}^* \tag{A402}$$

$$G_{9,1}^{E_{21}} \quad = \quad -\frac{x\mathcal{I}_{2,1}^*}{\phi} \tag{A403}$$

$$G_{9,3}^{E_{21}} \quad = \quad -\frac{x\mathcal{I}_{\frac{3}{2},1}^*}{\phi} \tag{A404}$$

$$G_{7,1}^{E_{21}} \quad = \quad G_{8,3}^{E_{21}} = \gamma\sqrt{\rho}\bar{\tau}\mathcal{I}_{\frac{3}{2},1}^* \tag{A405}$$

$$G_{1,1}^{E_{21}} = G_{2,2}^{E_{21}} = G_{3,3}^{E_{21}} = 1\,, \tag{A406}$$

where we have again used the definition of the LJSD $\mu$, the relations in Eqs. (A357)-(A377), as well as the results in Sec. A9.6.1 to simplify the expressions. Note that in these equations and the above example for the $(13,3)$ entry, we have leveraged the simple manner in which $\Sigma^*$ enters in Eqs. (A357)-(A377), namely linearly in the numerator, to simplify the dependence on $\Sigma$ and $\Sigma^*$. In particular, by rewriting the arguments of the trace terms in an eigenbasis of $\Sigma$, the only dependence on $\Sigma$ and $\Sigma^*$ that remains is through the training eigenvalues $\lambda$ and the overlap coefficients $r$. As such, the $\bar{\mathrm{tr}}$ in (Eq. (A383)) can be written as an expectation over the LJSD $\mu$ in the limit, which leads to significant simplification through the introduction of the two-index set of functions of $\mu$, $\mathcal{I}_{a,b}^*$, defined in Eq. (14).

It is straightforward algebra to solve these equations for the undetermined entries of $G^{E_{21}}$ and thereby obtain the following expression for $E_{21}$,

$$E_{21} = -2\frac{x}{\phi}\mathcal{I}_{2,1}^*\,. \tag{A407}$$

### A9.6.3 $E_{31}$

Define the block matrix $Q^{E_{31}} \equiv [Q_1^{E_{31}}\ Q_2^{E_{31}}]$ by,

$$Q_1^{E_{31}} = \begin{pmatrix} I_m & \frac{\sqrt{\eta-\zeta}\Theta^\top}{\gamma\sqrt{n_1}} & \frac{\sqrt{\rho}X^\top}{\gamma\sqrt{n_0}} & 0 & 0 & 0 \\ -\frac{\Theta\sqrt{\eta-\zeta}}{\sqrt{n_1}} & I_{n_1} & 0 & 0 & -\frac{\sqrt{\rho}W}{\sqrt{n_1}} & 0 \\ 0 & 0 & I_{n_0} & -\Sigma^{1/2} & 0 & 0 \\ 0 & -\frac{W^\top}{\sqrt{n_1}} & 0 & I_{n_0} & 0 & 0 \\ 0 & 0 & 0 & 0 & I_{n_0} & -\Sigma^{1/2} \\ -\frac{X}{\sqrt{n_0}} & 0 & 0 & 0 & 0 & I_{n_0} \\ 0 & 0 & 0 & 0 & 0 & 0 \\ 0 & 0 & 0 & 0 & 0 & 0 \\ 0 & 0 & 0 & 0 & 0 & 0 \\ 0 & 0 & 0 & 0 & 0 & 0 \\ 0 & 0 & 0 & 0 & 0 & 0 \\ 0 & 0 & 0 & 0 & 0 & 0 \end{pmatrix}, \tag{A408}$$

and,

$$Q_2^{E_{31}} = \begin{pmatrix} \frac{\sqrt{\eta-\zeta}\Theta^\top(\zeta-\eta)}{\gamma\sqrt{n_1}} & \frac{\sqrt{\rho}X^\top(\zeta-\eta)}{\gamma\sqrt{n_0}} & 0 & 0 & 0 & 0 \\ 0 & 0 & 0 & 0 & 0 & 0 \\ 0 & 0 & 0 & 0 & 0 & 0 \\ 0 & 0 & 0 & 0 & 0 & 0 \\ 0 & 0 & \frac{n_1\Sigma^*\rho}{n_0\sqrt{\rho}} & 0 & 0 & 0 \\ 0 & 0 & 0 & 0 & 0 & 0 \\ I_{n_1} & 0 & 0 & -\frac{\Theta\sqrt{\eta-\zeta}}{\sqrt{n_1}} & -\frac{\sqrt{\rho}W}{\sqrt{n_1}} & 0 \\ 0 & I_{n_0} & -\Sigma^{1/2} & 0 & 0 & 0 \\ -\frac{W^\top}{\sqrt{n_1}} & 0 & I_{n_0} & 0 & 0 & 0 \\ \frac{\sqrt{\eta-\zeta}\Theta^\top}{\gamma\sqrt{n_1}} & \frac{\sqrt{\rho}X^\top}{\gamma\sqrt{n_0}} & 0 & I_m & 0 & 0 \\ 0 & 0 & 0 & 0 & I_{n_0} & -\Sigma^{1/2} \\ 0 & 0 & 0 & -\frac{X}{\sqrt{n_0}} & 0 & I_{n_0} \end{pmatrix}. \tag{A409}$$

Then block matrix inversion (i.e. repeated applications of the Schur complement formula) shows that,

$$G_{1,1}^{E_{31}} = G_{10,10}^{E_{31}} = G_{1,1}^{K^{-1}} \tag{A410}$$

$$G_{2,2}^{E_{31}} = G_{7,7}^{E_{31}} = G_{2,2}^{K^{-1}} \tag{A411}$$

$$G_{3,3}^{E_{31}} = G_{6,6}^{E_{31}} = G_{8,8}^{E_{31}} = G_{12,12}^{E_{31}} = G_{4,4}^{E_{31}} = G_{5,5}^{E_{31}} = G_{9,9}^{E_{31}} = G_{11,11}^{E_{31}} = G_{3,3}^{K^{-1}} \tag{A412}$$

$$G_{3,4}^{E_{31}} = G_{5,6}^{E_{31}} = G_{8,9}^{E_{31}} = G_{11,12}^{E_{31}} = G_{3,4}^{K^{-1}} \tag{A413}$$

$$G_{3,5}^{E_{31}} = G_{4,6}^{E_{31}} = G_{8,11}^{E_{31}} = G_{9,12}^{E_{31}} = G_{3,5}^{K^{-1}} \tag{A414}$$

$$G_{3,6}^{E_{31}} = G_{8,12}^{E_{31}} = G_{3,6}^{K^{-1}} \tag{A415}$$

$$G_{4,3}^{E_{31}} = G_{6,5}^{E_{31}} = G_{9,8}^{E_{31}} = G_{12,11}^{E_{31}} = G_{4,3}^{K^{-1}} \tag{A416}$$

$$G_{4,5}^{E_{31}} = G_{9,11}^{E_{31}} = G_{4,5}^{K^{-1}} \tag{A417}$$

$$G_{5,3}^{E_{31}} = G_{6,4}^{E_{31}} = G_{11,8}^{E_{31}} = G_{12,9}^{E_{31}} = G_{5,3}^{K^{-1}} \tag{A418}$$

$$G_{5,4}^{E_{31}} = G_{11,9}^{E_{31}} = G_{5,4}^{K^{-1}} \tag{A419}$$

$$G_{6,3}^{E_{31}} = G_{12,8}^{E_{31}} = G_{6,3}^{K^{-1}} \tag{A420}$$

$$G_{10,1}^{E_{31}} = \frac{\gamma\phi}{\psi\sigma_\varepsilon^2} E_{31}\,, \tag{A421}$$

where $G_{i,j}^{E_{31}}$ denotes the normalized trace of the $(i,j)$-block of the inverse of $\left(Q^{E_{31}}\right)^\top$. For brevity, we have suppressed the expressions for the other non-zero blocks.

To compute the limiting values of these traces, we require the asymptotic block-wise traces of $Q^{E_{31}}$, which may be determined from the operator-valued Stieltjes transform. Proceeding as above, we first augment $Q^{E_{31}}$ to form the the self-adjoint matrix $\bar{Q}^{E_{31}}$,

$$\bar{Q}^{E_{31}} = \begin{pmatrix} 0 & \left[Q^{E_{31}}\right]^\top \\ Q^{E_{31}} & 0 \end{pmatrix}. \tag{A422}$$

and observe that we can write $\bar{Q}^{E_{31}}$ as,

$$\begin{aligned}
\bar{Q}^{E_{31}} &= \bar{Z} - \bar{Q}_{W,X,\Theta}^{E_{31}} - \bar{Q}_\Sigma^{E_{31}} \\
&= \begin{pmatrix} 0 & Z^\top \\ Z & 0 \end{pmatrix} - \begin{pmatrix} 0 & [Q_{W,X,\Theta}^{E_{31}}]^\top \\ Q_{W,X,\Theta}^{E_{31}} & 0 \end{pmatrix} - \begin{pmatrix} 0 & [Q_\Sigma^{E_{31}}]^\top \\ Q_\Sigma^{E_{31}} & 0 \end{pmatrix},
\end{aligned} \tag{A423}$$

where $Z = I_{2m+8n_0+2n_1}$, $Q_{W,X,\Theta}^{E_{31}} \equiv [[Q_{W,X,\Theta}^{E_{31}}]_1\ [Q_{W,X,\Theta}^{E_{31}}]_2]$ and,

$$[Q_{W,X,\Theta}^{E_{31}}]_1 = \begin{pmatrix}
0 & \frac{\sqrt{\eta-\zeta}\Theta^\top}{\gamma\sqrt{n_1}} & \frac{\sqrt{\rho}X^\top}{\gamma\sqrt{n_0}} & 0 & 0 & 0 \\
-\frac{\Theta\sqrt{\eta-\zeta}}{\sqrt{n_1}} & 0 & 0 & 0 & -\frac{\sqrt{\rho}W}{\sqrt{n_1}} & 0 \\
0 & 0 & 0 & 0 & 0 & 0 \\
0 & -\frac{W^\top}{\sqrt{n_1}} & 0 & 0 & 0 & 0 \\
0 & 0 & 0 & 0 & 0 & 0 \\
-\frac{X}{\sqrt{n_0}} & 0 & 0 & 0 & 0 & 0 \\
0 & 0 & 0 & 0 & 0 & 0 \\
0 & 0 & 0 & 0 & 0 & 0 \\
0 & 0 & 0 & 0 & 0 & 0 \\
0 & 0 & 0 & 0 & 0 & 0 \\
0 & 0 & 0 & 0 & 0 & 0
\end{pmatrix} \tag{A424}$$

$$[Q_{W,X,\Theta}^{E_{31}}]_2 = \begin{pmatrix}
\frac{\sqrt{\eta-\zeta}\Theta^\top(\zeta-\eta)}{\gamma\sqrt{n_1}} & \frac{\sqrt{\rho}X^\top(\zeta-\eta)}{\gamma\sqrt{n_0}} & 0 & 0 & 0 & 0 \\
0 & 0 & 0 & 0 & 0 & 0 \\
0 & 0 & 0 & 0 & 0 & 0 \\
0 & 0 & 0 & 0 & 0 & 0 \\
0 & 0 & 0 & 0 & 0 & 0 \\
0 & 0 & 0 & 0 & 0 & 0 \\
0 & 0 & 0 & -\frac{\Theta\sqrt{\eta-\zeta}}{\sqrt{n_1}} & -\frac{\sqrt{\rho}W}{\sqrt{n_1}} & 0 \\
0 & 0 & 0 & 0 & 0 & 0 \\
-\frac{W^\top}{\sqrt{n_1}} & 0 & 0 & 0 & 0 & 0 \\
\frac{\sqrt{\eta-\zeta}\Theta^\top}{\gamma\sqrt{n_1}} & \frac{\sqrt{\rho}X^\top}{\gamma\sqrt{n_0}} & 0 & 0 & 0 & 0 \\
0 & 0 & 0 & 0 & 0 & 0 \\
0 & 0 & 0 & -\frac{X}{\sqrt{n_0}} & 0 & 0
\end{pmatrix} \tag{A425}$$

$$Q_\Sigma^{E_{31}} = \begin{pmatrix} 0 & 0 & 0 & 0 & 0 & 0 & 0 & 0 & 0 & 0 & 0 & 0 \\ 0 & 0 & 0 & 0 & 0 & 0 & 0 & 0 & 0 & 0 & 0 & 0 \\ 0 & 0 & 0 & -\Sigma^{1/2} & 0 & 0 & 0 & 0 & 0 & 0 & 0 & 0 \\ 0 & 0 & 0 & 0 & 0 & 0 & 0 & 0 & 0 & 0 & 0 & 0 \\ 0 & 0 & 0 & 0 & 0 & -\Sigma^{1/2} & 0 & 0 & \frac{n_1 \Sigma^* \rho}{n_0 \sqrt{\rho}} & 0 & 0 & 0 \\ 0 & 0 & 0 & 0 & 0 & 0 & 0 & 0 & 0 & 0 & 0 & 0 \\ 0 & 0 & 0 & 0 & 0 & 0 & 0 & 0 & 0 & 0 & 0 & 0 \\ 0 & 0 & 0 & 0 & 0 & 0 & 0 & 0 & -\Sigma^{1/2} & 0 & 0 & 0 \\ 0 & 0 & 0 & 0 & 0 & 0 & 0 & 0 & 0 & 0 & 0 & 0 \\ 0 & 0 & 0 & 0 & 0 & 0 & 0 & 0 & 0 & 0 & 0 & 0 \\ 0 & 0 & 0 & 0 & 0 & 0 & 0 & 0 & 0 & 0 & 0 & -\Sigma^{1/2} \\ 0 & 0 & 0 & 0 & 0 & 0 & 0 & 0 & 0 & 0 & 0 & 0 \end{pmatrix} . \quad \text{(A426)}$$

The operator-valued Stieltjes transforms satisfy,

$$\begin{aligned} \bar{G}^{E_{31}} &= \bar{G}_\Sigma^{E_{31}} (\bar{Z} - \bar{R}_{W,X,\Theta}^{E_{31}}(\bar{G}^{E_{31}})) \\ &= \text{id} \otimes \bar{\text{tr}} \left( \bar{Z} - \bar{R}_{W,X,\Theta}^{E_{31}}(\bar{G}^{E_{31}}) - \bar{Q}_\Sigma^{E_{31}} \right)^{-1} , \end{aligned} \quad \text{(A427)}$$

where $\bar{R}_{W,X,\Theta}^{E_{31}}(\bar{G}^{E_{31}})$ is the operator-valued R-transform of $\bar{Q}_{W,X,\Theta}^{E_{31}}$. As discussed above, since $\bar{Q}_{W,X,\Theta}^{E_{31}}$ is a block matrix whose blocks are i.i.d. Gaussian matrices (and their transposes), an explicit expression for $\bar{R}_{W,X,\Theta}^{E_{31}}(\bar{G}^{E_{31}})$ can be obtained from the covariance map $\eta$, which can be read off from Eq. (A422). As above, we use the specific sparsity pattern for $G^{E_{31}}$ that is induced by Eq. (A427), to obtain,

$$\bar{R}_{W,X,\Theta}^{E_{31}}(\bar{G}^{E_{31}}) = \begin{pmatrix} 0 & R_{W,X,\Theta}^{E_{31}}(G^{E_{31}})^\top \\ R_{W,X,\Theta}^{E_{31}}(G^{E_{31}}) & 0 \end{pmatrix} , \quad \text{(A428)}$$

where,

$$[R_{W,X,\Theta}^{E_{31}}(G^{E_{31}})]_{1,1} = \frac{G_{2,2}^{E_{31}}(\zeta - \eta)}{\gamma} - \frac{\sqrt{\rho} G_{6,3}^{E_{31}}}{\gamma} + \frac{\sqrt{\rho} G_{6,8}^{E_{31}}(\eta - \zeta)}{\gamma} \\ + \frac{G_{2,7}^{E_{31}}(\zeta - \eta)(\zeta - \eta)}{\gamma} \quad \text{(A429)}$$

$$[R_{W,X,\Theta}^{E_{31}}(G^{E_{31}})]_{1,10} = \frac{G_{7,2}^{E_{31}}(\zeta - \eta)}{\gamma} - \frac{\sqrt{\rho} G_{12,3}^{E_{31}}}{\gamma} + \frac{\sqrt{\rho} G_{12,8}^{E_{31}}(\eta - \zeta)}{\gamma} \\ + \frac{G_{7,7}^{E_{31}}(\zeta - \eta)(\zeta - \eta)}{\gamma} \quad \text{(A430)}$$

$$[R_{W,X,\Theta}^{E_{31}}(G^{E_{31}})]_{2,2} = \frac{\psi G_{1,1}^{E_{31}}(\zeta - \eta)}{\gamma \phi} + \sqrt{\rho} \psi G_{4,5}^{E_{31}} \quad \text{(A431)}$$

$$[R_{W,X,\Theta}^{E_{31}}(G^{E_{31}})]_{2,7} = \frac{\psi G_{10,1}^{E_{31}}(\zeta - \eta)}{\gamma \phi} + \sqrt{\rho} \psi G_{9,5}^{E_{31}} + \frac{\psi G_{1,1}^{E_{31}}(\zeta - \eta)(\zeta - \eta)}{\gamma \phi} \quad \text{(A432)}$$

$$[R_{W,X,\Theta}^{E_{31}}(G^{E_{31}})]_{4,5} = \sqrt{\rho} G_{2,2}^{E_{31}} \quad \text{(A433)}$$

$$[R_{W,X,\Theta}^{E_{31}}(G^{E_{31}})]_{4,11} = \sqrt{\rho} G_{7,2}^{E_{31}} \quad \text{(A434)}$$

$$[R_{W,X,\Theta}^{E_{31}}(G^{E_{31}})]_{6,3} = -\frac{\sqrt{\rho} G_{1,1}^{E_{31}}}{\gamma \phi} \quad \text{(A435)}$$

$$[R_{W,X,\Theta}^{E_{31}}(G^{E_{31}})]_{6,8} = \frac{\sqrt{\rho} G_{1,1}^{E_{31}}(\eta - \zeta)}{\gamma \phi} - \frac{\sqrt{\rho} G_{10,1}^{E_{31}}}{\gamma \phi} \quad \text{(A436)}$$

$$[R_{W,X,\Theta}^{E_{31}}(G^{E_{31}})]_{7,2} = \frac{\psi G_{1,10}^{E_{31}}(\zeta - \eta)}{\gamma \phi} + \sqrt{\rho} \psi G_{4,11}^{E_{31}} \quad \text{(A437)}$$

$$[R_{W,X,\Theta}^{E_{31}}(G^{E_{31}})]_{7,7} = \frac{\psi G_{10,10}^{E_{31}}(\zeta - \eta)}{\gamma \phi} + \sqrt{\rho} \psi G_{9,11}^{E_{31}} + \frac{\psi G_{1,10}^{E_{31}}(\zeta - \eta)(\zeta - \eta)}{\gamma \phi} \quad \text{(A438)}$$

$$[R_{W,X,\Theta}^{E_{31}}(G^{E_{31}})]_{9,5} = \sqrt{\rho} G_{2,7}^{E_{31}} \quad \text{(A439)}$$

$$[R_{W,X,\Theta}^{E_{31}}(G^{E_{31}})]_{9,11} = \sqrt{\rho} G_{7,7}^{E_{31}} \quad \text{(A440)}$$

$$[R^{E_{31}}_{W,X,\Theta}(G^{E_{31}})]_{10,1} = \frac{G^{E_{31}}_{2,7}(\zeta - \eta)}{\gamma} - \frac{\sqrt{\rho}G^{E_{31}}_{6,8}}{\gamma} \tag{A441}$$

$$[R^{E_{31}}_{W,X,\Theta}(G^{E_{31}})]_{10,10} = \frac{G^{E_{31}}_{7,7}(\zeta - \eta)}{\gamma} - \frac{\sqrt{\rho}G^{E_{31}}_{12,8}}{\gamma} \tag{A442}$$

$$[R^{E_{31}}_{W,X,\Theta}(G^{E_{31}})]_{12,3} = -\frac{\sqrt{\rho}G^{E_{31}}_{1,10}}{\gamma\phi} \tag{A443}$$

$$[R^{E_{31}}_{W,X,\Theta}(G^{E_{31}})]_{12,8} = \frac{\sqrt{\rho}G^{E_{31}}_{1,10}(\eta - \zeta)}{\gamma\phi} - \frac{\sqrt{\rho}G^{E_{31}}_{10,10}}{\gamma\phi}, \tag{A444}$$

and the remaining entries of $R^{E_{31}}_{W,X,\Theta}(G^{E_{31}})$ are zero. Similarly, following the example from $G^{E_{21}}$ above, plugging these expressions into Eq. (A427) and explicitly performing the block-matrix inverse yields the following set of coupled equations,

$$G^{E_{31}}_{7,2} = \gamma^2\sqrt{\rho}\bar\tau^2\psi G^{E_{31}}_{9,5} + \frac{\gamma\bar\tau^2\psi G^{E_{31}}_{10,1}(\zeta - \eta)}{\phi} + \frac{\gamma^2\tau\bar\tau^2\psi(\zeta - \eta)(\zeta - \eta)}{\phi} \tag{A445}$$

$$G^{E_{31}}_{8,6} = \frac{\rho\tau\psi\phi\mathcal{I}_{0,2}(\eta - \zeta) - x^2\mathcal{I}^*_{2,2}\rho}{\sqrt{\rho}\psi\phi} - \frac{\sqrt{\rho}G^{E_{31}}_{10,1}\mathcal{I}_{0,2}}{\gamma} + \frac{\rho^{3/2}\tau^2 G^{E_{31}}_{7,2}\mathcal{I}_{1,2}}{\phi} \tag{A446}$$

$$G^{E_{31}}_{9,5} = \frac{\rho\tau\mathcal{I}_{1,2}(\eta - \zeta) - \frac{\phi\mathcal{I}^*_{1,2}\rho}{\psi}}{\sqrt{\rho}} - \frac{\sqrt{\rho}G^{E_{31}}_{10,1}\mathcal{I}_{1,2}}{\gamma} + \frac{\rho^{3/2}\tau^2 G^{E_{31}}_{7,2}\mathcal{I}_{2,2}}{\phi} \tag{A447}$$

$$G^{E_{31}}_{10,1} = \gamma\tau^2 G^{E_{31}}_{7,2}(\zeta - \eta) - \gamma\sqrt{\rho}\tau^2 G^{E_{31}}_{12,3} + \gamma\tau(\gamma\tau - 1)(\zeta - \eta) \tag{A448}$$

$$G^{E_{31}}_{11,4} = -\frac{\gamma^2\sqrt{\rho}\bar\tau^2\left(\phi\mathcal{I}^*_{1,2}\rho + \rho\tau\psi\mathcal{I}_{1,2}(\zeta - \eta)\right)}{\psi} + \sqrt{\rho}\phi G^{E_{31}}_{7,2}\mathcal{I}_{0,2} - \gamma\rho^{3/2}\bar\tau^2 G^{E_{31}}_{10,1}\mathcal{I}_{1,2} \tag{A449}$$

$$G^{E_{31}}_{12,3} = -\frac{\gamma^2\sqrt{\rho}\bar\tau^2\left(\phi\mathcal{I}^*_{2,2}\rho + \rho\tau\psi\mathcal{I}_{2,2}(\zeta - \eta)\right)}{\psi} + \sqrt{\rho}\phi G^{E_{31}}_{7,2}\mathcal{I}_{1,2} - \gamma\rho^{3/2}\bar\tau^2 G^{E_{31}}_{10,1}\mathcal{I}_{2,2} \tag{A450}$$

$$G^{E_{31}}_{8,3} = G^{E_{31}}_{12,6} = \frac{\gamma^2\rho\tau\bar\tau^2\mathcal{I}^*_{2,2}\rho}{\psi} - \rho\tau G^{E_{31}}_{7,2}\mathcal{I}_{1,2} - \rho\bar\tau G^{E_{31}}_{10,1}\mathcal{I}_{1,2} + x\mathcal{I}_{1,2}(\eta - \zeta) \tag{A451}$$

$$G^{E_{31}}_{8,4} = G^{E_{31}}_{11,6} = \frac{\gamma^2\rho\tau\bar\tau^2\mathcal{I}^*_{\frac{3}{2},2}\rho}{\psi} - \rho\tau G^{E_{31}}_{7,2}\mathcal{I}_{\frac{1}{2},2} - \rho\bar\tau G^{E_{31}}_{10,1}\mathcal{I}_{\frac{1}{2},2} + x\mathcal{I}_{\frac{1}{2},2}(\eta - \zeta) \tag{A452}$$

$$G^{E_{31}}_{8,5} = G^{E_{31}}_{9,6} = \frac{\frac{x}{\psi}\mathcal{I}^*_{\frac{3}{2},2}\rho + \rho\tau\mathcal{I}_{\frac{1}{2},2}(\eta - \zeta)}{\sqrt{\rho}} - \frac{\sqrt{\rho}G^{E_{31}}_{10,1}\mathcal{I}_{\frac{1}{2},2}}{\gamma} + \frac{\rho^{3/2}\tau^2 G^{E_{31}}_{7,2}\mathcal{I}_{\frac{3}{2},2}}{\phi} \tag{A453}$$

$$G^{E_{31}}_{9,3} = G^{E_{31}}_{12,5} = -\frac{\gamma\bar\tau\phi\mathcal{I}^*_{\frac{3}{2},2}\rho}{\psi} - \rho\tau G^{E_{31}}_{7,2}\mathcal{I}_{\frac{3}{2},2} - \rho\bar\tau G^{E_{31}}_{10,1}\mathcal{I}_{\frac{3}{2},2} + x\mathcal{I}_{\frac{3}{2},2}(\eta - \zeta) \tag{A454}$$

$$G^{E_{31}}_{9,4} = G^{E_{31}}_{11,5} = -\frac{\gamma\bar\tau\phi\mathcal{I}^*_{1,2}\rho}{\psi} - \rho\tau G^{E_{31}}_{7,2}\mathcal{I}_{1,2} - \rho\bar\tau G^{E_{31}}_{10,1}\mathcal{I}_{1,2} + x\mathcal{I}_{1,2}(\eta - \zeta) \tag{A455}$$

$$\begin{aligned}G^{E_{31}}_{11,3} = G^{E_{31}}_{12,4} = &-\frac{\gamma^2\sqrt{\rho}\bar\tau^2\left(\phi\mathcal{I}^*_{\frac{3}{2},2}\rho + \rho\tau\psi\mathcal{I}_{\frac{3}{2},2}(\zeta - \eta)\right)}{\psi} + \sqrt{\rho}\phi G^{E_{31}}_{7,2}\mathcal{I}_{\frac{1}{2},2}\\ &- \gamma\rho^{3/2}\bar\tau^2 G^{E_{31}}_{10,1}\mathcal{I}_{\frac{3}{2},2},\end{aligned} \tag{A456}$$

Here we have used the definition of the LJSD $\mu$, the relations in Eqs. (A410)-(A421), the definition of $\mathcal{I}^*_{a,b}$, and the results in Sec. A9.6.1 to simplify the expressions. It is straightforward algebra to solve these equations for the undetermined entries of $G^{E_{31}}$ and thereby obtain the following expression for $E_{31}$,

$$E_{31} = \sigma^2_\varepsilon \frac{(\eta - \zeta)A_{31} + \rho B_{31}}{D_{31}}, \tag{A457}$$

where,

$$\begin{aligned}A_{31} = \ &\rho^2\tau\psi x^2\mathcal{I}_{2,2}(-\gamma\tau\psi + \psi + \phi) + 2\rho\tau\psi^2 x^2\phi(\eta - \zeta)\mathcal{I}_{1,2} + \rho^2\tau\psi^2 x^2\phi^2\mathcal{I}^2_{1,2}\\ &+ \tau\psi\left(x^2\psi(\zeta - \eta)^2 - \rho^2(\phi - \gamma\tau\phi)\right) - \rho^2\tau\psi^2 x^4\mathcal{I}^2_{2,2}\end{aligned} \tag{A458}$$

$$B_{31} = \rho\psi x^4 \phi \mathcal{I}_{2,2}^* \mathcal{I}_{2,2} - \rho\psi x^2 \phi^3 \mathcal{I}_{1,2}^* \mathcal{I}_{1,2} + \psi x^2 \phi^2 \mathcal{I}_{1,2}^*(\zeta - \eta) - \rho x^2 \phi^2 \mathcal{I}_{2,2}^* \tag{A459}$$

$$D_{31} = -\rho^2 \psi x^4 \phi \mathcal{I}_{2,2}^2 + 2\rho\psi x^2 \phi^2 \mathcal{I}_{1,2}(\eta - \zeta) + \rho^2 \psi x^2 \phi^3 \mathcal{I}_{1,2}^2 + \rho^2 x^2 \phi \mathcal{I}_{2,2}(\psi + \phi)$$
$$+ \phi \left( x^2 \psi(\zeta - \eta)^2 - \rho^2 \phi \right) . \tag{A460}$$

Further simplifications are possible using the raising and lowering identities in Eq. (A5), as well as the results in Sec. A9.6.1, to obtain,

$$E_{31} = \sigma_\varepsilon^2 \frac{\tau \bar{\tau} x \left( \rho \frac{\psi}{\phi}(\phi \mathcal{I}_{1,2} + \omega)(\omega + \mathcal{I}_{1,1}^*) + \frac{x}{\tau} \mathcal{I}_{2,2}^* \right)}{\tau + \bar{\tau} x(\omega + \phi \mathcal{I}_{1,2}) - \tau x^2 \frac{\psi}{\phi} \mathcal{I}_{2,2}} \tag{A461}$$

$$= -\sigma_\varepsilon^2 \frac{\partial x}{\partial \gamma} \left( \rho \frac{\psi}{\phi}(\phi \mathcal{I}_{1,2} + \omega)(\omega + \mathcal{I}_{1,1}^*) + \frac{x}{\tau} \mathcal{I}_{2,2}^* \right), \tag{A462}$$

where we have used,

$$\frac{\partial x}{\partial \gamma} = -\frac{x}{\gamma + \rho\gamma(\frac{\psi}{\phi}\tau + \bar{\tau})(\omega + \phi \mathcal{I}_{1,2})}, \tag{A463}$$

which follows from Eq. (A345) via implicit differentiation.

### A9.6.4 $E_{32}$

Define the block matrix $Q^{E_{32}} \equiv [Q_1^{E_{32}} \ Q_2^{E_{32}}]$ by,

$$Q_1^{E_{32}} = \begin{pmatrix}
I_m & \frac{\sqrt{\eta-\zeta}\Theta^\top}{\gamma\sqrt{n_1}} & \frac{\sqrt{\rho}X^\top}{\gamma\sqrt{n_0}} & 0 & 0 & 0 & \frac{\sqrt{\eta-\zeta}\Theta^\top(\zeta-\eta)}{\gamma\sqrt{n_1}} \\
-\frac{\Theta\sqrt{\eta-\zeta}}{\sqrt{n_1}} & I_{n_1} & 0 & 0 & -\frac{\sqrt{\rho}W}{\sqrt{n_1}} & 0 & 0 \\
0 & 0 & I_{n_0} & -\Sigma^{1/2} & 0 & 0 & 0 \\
0 & -\frac{W^\top}{\sqrt{n_1}} & 0 & I_{n_0} & 0 & 0 & 0 \\
0 & 0 & 0 & 0 & I_{n_0} & -\Sigma^{1/2} & 0 \\
-\frac{X}{\sqrt{n_0}} & 0 & 0 & 0 & 0 & I_{n_0} & 0 \\
0 & 0 & 0 & 0 & 0 & 0 & I_{n_1} \\
0 & 0 & 0 & 0 & 0 & 0 & -\frac{W^\top}{\sqrt{n_1}} \\
0 & 0 & 0 & 0 & 0 & 0 & \frac{\sqrt{\eta-\zeta}\Theta^\top}{\gamma\sqrt{n_1}} \\
0 & 0 & 0 & 0 & 0 & 0 & 0 \\
0 & 0 & 0 & 0 & 0 & 0 & 0 \\
0 & 0 & 0 & 0 & 0 & 0 & -\frac{W^\top}{\sqrt{n_1}} \\
0 & 0 & 0 & 0 & 0 & 0 & 0 \\
0 & 0 & 0 & 0 & 0 & 0 & 0
\end{pmatrix}, \tag{A464}$$

and,

$$Q_2^{E_{32}} = \begin{pmatrix}
0 & 0 & 0 & 0 & 0 & 0 & 0 & 0 \\
0 & 0 & 0 & 0 & 0 & 0 & 0 & 0 \\
\Sigma^{1/2}(\eta-\zeta) & 0 & 0 & 0 & 0 & 0 & 0 & 0 \\
0 & 0 & 0 & 0 & 0 & 0 & 0 & 0 \\
\frac{n_1\Sigma^*\rho}{n_0\sqrt{\rho}} & 0 & 0 & 0 & 0 & 0 & 0 & 0 \\
0 & 0 & 0 & 0 & 0 & 0 & 0 & 0 \\
0 & -\frac{\Theta\sqrt{\eta-\zeta}}{\sqrt{n_1}} & -\frac{\sqrt{\rho}W}{\sqrt{n_1}} & 0 & 0 & 0 & 0 & 0 \\
I_{n_0} & 0 & 0 & 0 & 0 & 0 & 0 & 0 \\
0 & I_m & 0 & 0 & \frac{\sqrt{\rho}X^\top}{\gamma\sqrt{n_0}} & 0 & 0 & 0 \\
0 & 0 & I_{n_0} & -\Sigma^{1/2} & 0 & 0 & 0 & 0 \\
0 & -\frac{X}{\sqrt{n_0}} & 0 & I_{n_0} & 0 & 0 & 0 & 0 \\
0 & 0 & 0 & 0 & I_{n_0} & -\Sigma^{1/2} & 0 & 0 \\
0 & 0 & 0 & 0 & 0 & I_{n_0} & \frac{\Sigma^{1/2}}{\sqrt{\rho}} & 0 \\
0 & 0 & 0 & 0 & 0 & 0 & I_{n_0} & -\frac{X}{\sqrt{n_0}} \\
0 & 0 & 0 & 0 & 0 & 0 & 0 & I_m
\end{pmatrix}. \tag{A465}$$

Then block matrix inversion (i.e. repeated applications of the Schur complement formula) shows that,

$$G_{8,8}^{E_{32}} = G_{14,14}^{E_{32}} = G_{15,15}^{E_{32}} = 1 \tag{A466}$$

$$G_{1,1}^{E_{32}} = G_{9,9}^{E_{32}} = G_{1,1}^{K^{-1}} \tag{A467}$$

$$G_{2,2}^{E_{32}} = G_{7,7}^{E_{32}} = G_{2,2}^{K^{-1}} \tag{A468}$$

$$G_{3,3}^{E_{32}} = G_{6,6}^{E_{32}} = G_{11,11}^{E_{32}} = G_{12,12}^{E_{32}} = G_{4,4}^{E_{32}} = G_{5,5}^{E_{32}} = G_{10,10}^{E_{32}} = G_{13,13}^{E_{32}} = G_{3,3}^{K^{-1}} \tag{A469}$$

$$G_{3,4}^{E_{32}} = G_{5,6}^{E_{32}} = G_{10,11}^{E_{32}} = G_{12,8}^{E_{32}} = G_{12,13}^{E_{32}} = G_{3,4}^{K^{-1}} \tag{A470}$$

$$G_{3,5}^{E_{32}} = G_{4,6}^{E_{32}} = G_{12,10}^{E_{32}} = G_{13,11}^{E_{32}} = G_{3,5}^{K^{-1}} \tag{A471}$$

$$G_{3,6}^{E_{32}} = G_{12,11}^{E_{32}} = G_{3,6}^{K^{-1}} \tag{A472}$$

$$G_{4,3}^{E_{32}} = G_{6,5}^{E_{32}} = G_{11,10}^{E_{32}} = G_{13,12}^{E_{32}} = G_{4,3}^{K^{-1}} \tag{A473}$$

$$G_{4,5}^{E_{32}} = G_{13,10}^{E_{32}} = G_{4,5}^{K^{-1}} \tag{A474}$$

$$G_{5,3}^{E_{32}} = G_{6,4}^{E_{32}} = G_{10,12}^{E_{32}} = G_{11,8}^{E_{32}} = G_{11,13}^{E_{32}} = G_{5,3}^{K^{-1}} \tag{A475}$$

$$G_{5,4}^{E_{32}} = G_{10,8}^{E_{32}} = G_{10,13}^{E_{32}} = G_{5,4}^{K^{-1}} \tag{A476}$$

$$G_{6,3}^{E_{32}} = G_{11,12}^{E_{32}} = G_{6,3}^{K^{-1}} \tag{A477}$$

$$G_{15,1}^{E_{32}} = \frac{\phi}{\psi} E_{32} , \tag{A478}$$

where $G_{i,j}^{E_{32}}$ denotes the normalized trace of the $(i,j)$-block of the inverse of $\left(Q^{E_{32}}\right)^{\top}$. For brevity, we have suppressed the expressions for the other non-zero blocks.

To compute the limiting values of these traces, we require the asymptotic block-wise traces of $Q^{E_{32}}$, which may be determined from the operator-valued Stieltjes transform. To proceed, we first augment $Q^{E_{32}}$ to form the the self-adjoint matrix $\bar{Q}^{E_{32}}$,

$$\bar{Q}^{E_{32}} = \begin{pmatrix} 0 & [Q^{E_{32}}]^{\top} \\ Q^{E_{32}} & 0 \end{pmatrix} . \tag{A479}$$

and observe that we can write $\bar{Q}^{E_{32}}$ as,

$$
\begin{aligned}
\bar{Q}^{E_{32}} &= \bar{Z} - \bar{Q}_{W,X,\Theta}^{E_{32}} - \bar{Q}_{\Sigma}^{E_{32}} \\
&= \begin{pmatrix} 0 & Z^{\top} \\ Z & 0 \end{pmatrix} - \begin{pmatrix} 0 & [Q_{W,X,\Theta}^{E_{32}}]^{\top} \\ Q_{W,X,\Theta}^{E_{32}} & 0 \end{pmatrix} - \begin{pmatrix} 0 & [Q_{\Sigma}^{E_{32}}]^{\top} \\ Q_{\Sigma}^{E_{32}} & 0 \end{pmatrix} ,
\end{aligned}
\tag{A480}
$$

where $Z = I_{3m+10n_0+2n_1}$, $Q_{W,X,\Theta}^{E_{32}} \equiv [[Q_{W,X,\Theta}^{E_{32}}]_1 \ [Q_{W,X,\Theta}^{E_{32}}]_2]$ and,

$$[Q_{W,X,\Theta}^{E_{32}}]_1 = \begin{pmatrix}
0 & \frac{\sqrt{\eta-\zeta}\Theta^{\top}}{\gamma\sqrt{n_1}} & \frac{\sqrt{\rho}X^{\top}}{\gamma\sqrt{n_0}} & 0 & 0 & 0 & \frac{\sqrt{\eta-\zeta}\Theta^{\top}(\zeta-\eta)}{\gamma\sqrt{n_1}} \\
-\frac{\Theta\sqrt{\eta-\zeta}}{\sqrt{n_1}} & 0 & 0 & 0 & -\frac{\sqrt{\rho}W}{\sqrt{n_1}} & 0 & 0 \\
0 & 0 & 0 & 0 & 0 & 0 & 0 \\
0 & -\frac{W^{\top}}{\sqrt{n_1}} & 0 & 0 & 0 & 0 & 0 \\
0 & 0 & 0 & 0 & 0 & 0 & 0 \\
-\frac{X}{\sqrt{n_0}} & 0 & 0 & 0 & 0 & 0 & 0 \\
0 & 0 & 0 & 0 & 0 & 0 & 0 \\
0 & 0 & 0 & 0 & 0 & 0 & -\frac{W^{\top}}{\sqrt{n_1}} \\
0 & 0 & 0 & 0 & 0 & 0 & \frac{\sqrt{\eta-\zeta}\Theta^{\top}}{\gamma\sqrt{n_1}} \\
0 & 0 & 0 & 0 & 0 & 0 & 0 \\
0 & 0 & 0 & 0 & 0 & 0 & 0 \\
0 & 0 & 0 & 0 & 0 & 0 & -\frac{W^{\top}}{\sqrt{n_1}} \\
0 & 0 & 0 & 0 & 0 & 0 & 0 \\
0 & 0 & 0 & 0 & 0 & 0 & 0
\end{pmatrix} \tag{A481}$$

$$[Q_{W,X,\Theta}^{E_{32}}]_2 = \begin{pmatrix}
0 & 0 & 0 & 0 & 0 & 0 & 0 & 0 \\
0 & 0 & 0 & 0 & 0 & 0 & 0 & 0 \\
0 & 0 & 0 & 0 & 0 & 0 & 0 & 0 \\
0 & 0 & 0 & 0 & 0 & 0 & 0 & 0 \\
0 & 0 & 0 & 0 & 0 & 0 & 0 & 0 \\
0 & 0 & 0 & 0 & 0 & 0 & 0 & 0 \\
0 & -\frac{\Theta\sqrt{\eta-\zeta}}{\sqrt{n_1}} & -\frac{\sqrt{\rho}W}{\sqrt{n_1}} & 0 & 0 & 0 & 0 & 0 \\
0 & 0 & 0 & 0 & 0 & 0 & 0 & 0 \\
0 & 0 & 0 & 0 & \frac{\sqrt{\rho}X^{\top}}{\gamma\sqrt{n_0}} & 0 & 0 & 0 \\
0 & 0 & 0 & 0 & 0 & 0 & 0 & 0 \\
0 & -\frac{X}{\sqrt{n_0}} & 0 & 0 & 0 & 0 & 0 & 0 \\
0 & 0 & 0 & 0 & 0 & 0 & 0 & 0 \\
0 & 0 & 0 & 0 & 0 & 0 & 0 & 0 \\
0 & 0 & 0 & 0 & 0 & 0 & 0 & -\frac{X}{\sqrt{n_0}} \\
0 & 0 & 0 & 0 & 0 & 0 & 0 & 0
\end{pmatrix} \tag{A482}$$

$$
Q_\Sigma^{E_{32}} = \begin{pmatrix}
0 & 0 & 0 & 0 & 0 & 0 & 0 & 0 & 0 & 0 & 0 & 0 & 0 & 0 & 0 \\
0 & 0 & 0 & 0 & 0 & 0 & 0 & 0 & 0 & 0 & 0 & 0 & 0 & 0 & 0 \\
0 & 0 & 0 & -\Sigma^{1/2} & 0 & 0 & 0 & \Sigma^{1/2}\left(\eta - \zeta\right) & 0 & 0 & 0 & 0 & 0 & 0 & 0 \\
0 & 0 & 0 & 0 & 0 & 0 & 0 & 0 & 0 & 0 & 0 & 0 & 0 & 0 & 0 \\
0 & 0 & 0 & 0 & 0 & -\Sigma^{1/2} & 0 & \frac{n_1 \Sigma^* \rho}{n_0 \sqrt{\rho}} & 0 & 0 & 0 & 0 & 0 & 0 & 0 \\
0 & 0 & 0 & 0 & 0 & 0 & 0 & 0 & 0 & 0 & 0 & 0 & 0 & 0 & 0 \\
0 & 0 & 0 & 0 & 0 & 0 & 0 & 0 & 0 & 0 & 0 & 0 & 0 & 0 & 0 \\
0 & 0 & 0 & 0 & 0 & 0 & 0 & 0 & 0 & 0 & 0 & 0 & 0 & 0 & 0 \\
0 & 0 & 0 & 0 & 0 & 0 & 0 & 0 & 0 & 0 & 0 & 0 & 0 & 0 & 0 \\
0 & 0 & 0 & 0 & 0 & 0 & 0 & 0 & 0 & 0 & -\Sigma^{1/2} & 0 & 0 & 0 & 0 \\
0 & 0 & 0 & 0 & 0 & 0 & 0 & 0 & 0 & 0 & 0 & 0 & 0 & 0 & 0 \\
0 & 0 & 0 & 0 & 0 & 0 & 0 & 0 & 0 & 0 & 0 & 0 & -\Sigma^{1/2} & 0 & 0 \\
0 & 0 & 0 & 0 & 0 & 0 & 0 & 0 & 0 & 0 & 0 & 0 & 0 & \frac{\Sigma^{1/2}}{\sqrt{\rho}} & 0 \\
0 & 0 & 0 & 0 & 0 & 0 & 0 & 0 & 0 & 0 & 0 & 0 & 0 & 0 & 0 \\
0 & 0 & 0 & 0 & 0 & 0 & 0 & 0 & 0 & 0 & 0 & 0 & 0 & 0 & 0 \\
\end{pmatrix} .
$$

(A483)

The operator-valued Stieltjes transforms satisfy,

$$
\begin{aligned}
\bar{G}^{E_{32}} &= \bar{G}_\Sigma^{E_{32}}(\bar{Z} - \bar{R}_{W,X,\Theta}^{E_{32}}(\bar{G}^{E_{32}})) \\
&= \mathrm{id} \otimes \bar{\mathrm{tr}}\left(\bar{Z} - \bar{R}_{W,X,\Theta}^{E_{32}}(\bar{G}^{E_{32}}) - \bar{Q}_\Sigma^{E_{32}}\right)^{-1},
\end{aligned}
$$

(A484)

where $\bar{R}_{W,X,\Theta}^{E_{32}}(\bar{G}^{E_{32}})$ is the operator-valued R-transform of $\bar{Q}_{W,X,\Theta}^{E_{32}}$. As discussed above, since $\bar{Q}_{W,X,\Theta}^{E_{32}}$ is a block matrix whose blocks are i.i.d. Gaussian matrices (and their transposes), an explicit expression for $\bar{R}_{W,X,\Theta}^{E_{32}}(\bar{G}^{E_{32}})$ can be obtained from the covariance map $\eta$, which can be read off from Eq. (A479). As above, we use the specific sparsity pattern for $G^{E_{32}}$ that is induced by Eq. (A484), to obtain,

$$
\bar{R}_{W,X,\Theta}^{E_{32}}(\bar{G}^{E_{32}}) = \begin{pmatrix} 0 & R_{W,X,\Theta}^{E_{32}}(G^{E_{32}})^\top \\ R_{W,X,\Theta}^{E_{32}}(G^{E_{32}}) & 0 \end{pmatrix},
$$

(A485)

where,

$$
[R_{W,X,\Theta}^{E_{32}}(G^{E_{32}})]_{1,1} = \frac{G_{2,2}^{E_{32}}\left(\zeta - \eta\right)}{\gamma} - \frac{\sqrt{\rho}G_{6,3}^{E_{32}}}{\gamma} + \frac{G_{2,7}^{E_{32}}\left(\zeta - \eta\right)\left(\zeta - \eta\right)}{\gamma}
$$

(A486)

$$
[R_{W,X,\Theta}^{E_{32}}(G^{E_{32}})]_{1,9} = \frac{G_{7,2}^{E_{32}}\left(\zeta - \eta\right)}{\gamma} - \frac{\sqrt{\rho}G_{11,3}^{E_{32}}}{\gamma} + \frac{G_{7,7}^{E_{32}}\left(\zeta - \eta\right)\left(\zeta - \eta\right)}{\gamma}
$$

(A487)

$$
[R_{W,X,\Theta}^{E_{32}}(G^{E_{32}})]_{1,15} = -\frac{\sqrt{\rho}G_{14,3}^{E_{32}}}{\gamma}
$$

(A488)

$$
[R_{W,X,\Theta}^{E_{32}}(G^{E_{32}})]_{2,2} = \frac{\psi G_{1,1}^{E_{32}}\left(\zeta - \eta\right)}{\gamma\phi} + \sqrt{\rho}\psi G_{4,5}^{E_{32}}
$$

(A489)

$$
\begin{aligned}
[R_{W,X,\Theta}^{E_{32}}(G^{E_{32}})]_{2,7} &= \frac{\psi G_{9,1}^{E_{32}}\left(\zeta - \eta\right)}{\gamma\phi} + \sqrt{\rho}\psi G_{8,5}^{E_{32}} + \sqrt{\rho}\psi G_{13,5}^{E_{32}} \\
&\quad + \frac{\psi G_{1,1}^{E_{32}}\left(\zeta - \eta\right)\left(\zeta - \eta\right)}{\gamma\phi}
\end{aligned}
$$

(A490)

$$
[R_{W,X,\Theta}^{E_{32}}(G^{E_{32}})]_{4,5} = \sqrt{\rho}G_{2,2}^{E_{32}}
$$

(A491)

$$
[R_{W,X,\Theta}^{E_{32}}(G^{E_{32}})]_{4,10} = \sqrt{\rho}G_{7,2}^{E_{32}}
$$

(A492)

$$
[R_{W,X,\Theta}^{E_{32}}(G^{E_{32}})]_{6,3} = -\frac{\sqrt{\rho}G_{1,1}^{E_{32}}}{\gamma\phi}
$$

(A493)

$$
[R_{W,X,\Theta}^{E_{32}}(G^{E_{32}})]_{6,12} = -\frac{\sqrt{\rho}G_{9,1}^{E_{32}}}{\gamma\phi}
$$

(A494)

$$
[R_{W,X,\Theta}^{E_{32}}(G^{E_{32}})]_{7,2} = \frac{\psi G_{1,9}^{E_{32}}\left(\zeta - \eta\right)}{\gamma\phi} + \sqrt{\rho}\psi G_{4,10}^{E_{32}}
$$

(A495)

$$
[R_{W,X,\Theta}^{E_{32}}(G^{E_{32}})]_{7,7} = \frac{\psi G_{9,9}^{E_{32}}\left(\zeta - \eta\right)}{\gamma\phi} + \sqrt{\rho}\psi G_{8,10}^{E_{32}} + \sqrt{\rho}\psi G_{13,10}^{E_{32}}
$$

$$+ \frac{\psi G_{1,9}^{E_{32}} (\zeta - \eta) (\zeta - \eta)}{\gamma \phi} \tag{A496}$$

$$[R_{W,X,\Theta}^{E_{32}}(G^{E_{32}})]_{8,5} = \sqrt{\rho} G_{2,7}^{E_{32}} \tag{A497}$$

$$[R_{W,X,\Theta}^{E_{32}}(G^{E_{32}})]_{8,10} = \sqrt{\rho} G_{7,7}^{E_{32}} \tag{A498}$$

$$[R_{W,X,\Theta}^{E_{32}}(G^{E_{32}})]_{9,1} = \frac{G_{2,7}^{E_{32}} (\zeta - \eta)}{\gamma} - \frac{\sqrt{\rho} G_{6,12}^{E_{32}}}{\gamma} \tag{A499}$$

$$[R_{W,X,\Theta}^{E_{32}}(G^{E_{32}})]_{9,9} = \frac{G_{7,7}^{E_{32}} (\zeta - \eta)}{\gamma} - \frac{\sqrt{\rho} G_{11,12}^{E_{32}}}{\gamma} \tag{A500}$$

$$[R_{W,X,\Theta}^{E_{32}}(G^{E_{32}})]_{9,15} = -\frac{\sqrt{\rho} G_{14,12}^{E_{32}}}{\gamma} \tag{A501}$$

$$[R_{W,X,\Theta}^{E_{32}}(G^{E_{32}})]_{11,3} = -\frac{\sqrt{\rho} G_{1,9}^{E_{32}}}{\gamma \phi} \tag{A502}$$

$$[R_{W,X,\Theta}^{E_{32}}(G^{E_{32}})]_{11,12} = -\frac{\sqrt{\rho} G_{9,9}^{E_{32}}}{\gamma \phi} \tag{A503}$$

$$[R_{W,X,\Theta}^{E_{32}}(G^{E_{32}})]_{13,5} = \sqrt{\rho} G_{2,7}^{E_{32}} \tag{A504}$$

$$[R_{W,X,\Theta}^{E_{32}}(G^{E_{32}})]_{13,10} = \sqrt{\rho} G_{7,7}^{E_{32}} \tag{A505}$$

$$[R_{W,X,\Theta}^{E_{32}}(G^{E_{32}})]_{14,3} = -\frac{\sqrt{\rho} G_{1,15}^{E_{32}}}{\gamma \phi} \tag{A506}$$

$$[R_{W,X,\Theta}^{E_{32}}(G^{E_{32}})]_{14,12} = -\frac{\sqrt{\rho} G_{9,15}^{E_{32}}}{\gamma \phi} \,, \tag{A507}$$

and the remaining entries of $R_{W,X,\Theta}^{E_{32}}(G^{E_{32}})$ are zero. Similarly, following the example from $G^{E_{21}}$ above, plugging these expressions into Eq. (A484) and explicitly performing the block-matrix inverse yields the following set of coupled equations,

$$G_{7,2}^{E_{32}} = \gamma^2 \sqrt{\rho} \bar{\tau}^2 \psi G_{8,5}^{E_{32}} + \gamma^2 \sqrt{\rho} \bar{\tau}^2 \psi G_{13,5}^{E_{32}} + \frac{\gamma \bar{\tau}^2 \psi G_{9,1}^{E_{32}} (\zeta - \eta)}{\phi}$$
$$+ \frac{\gamma^2 \tau \bar{\tau}^2 \psi (\zeta - \eta) (\zeta - \eta)}{\phi} \tag{A508}$$

$$G_{8,3}^{E_{32}} = \mathcal{I}_{\frac{1}{2},1} (\zeta - \eta) - \frac{\gamma \bar{\tau} \mathcal{I}_{\frac{3}{2},1}^* \rho}{\psi} \tag{A509}$$

$$G_{8,4}^{E_{32}} = \gamma(-\bar{\tau}) \left( \frac{\mathcal{I}_{1,1}^* \rho}{\psi} + \frac{\rho \tau \mathcal{I}_{1,1} (\zeta - \eta)}{\phi} \right) \tag{A510}$$

$$G_{8,5}^{E_{32}} = \frac{\rho \tau \psi \mathcal{I}_{1,1} (\eta - \zeta) - \phi \mathcal{I}_{1,1}^* \rho}{\sqrt{\rho} \psi \phi} \tag{A511}$$

$$G_{8,6}^{E_{32}} = \frac{\sqrt{\rho} \tau \left( \gamma \bar{\tau} \mathcal{I}_{\frac{3}{2},1}^* \rho + \psi \mathcal{I}_{\frac{1}{2},1} (\eta - \zeta) \right)}{\psi \phi} \tag{A512}$$

$$G_{9,1}^{E_{32}} = \gamma \tau^2 G_{7,2}^{E_{32}} (\zeta - \eta) - \gamma \sqrt{\rho} \tau^2 G_{11,3}^{E_{32}} + \gamma^2 \tau^2 \bar{\tau} (\zeta - \eta) (\zeta - \eta) \tag{A513}$$

$$G_{10,3}^{E_{32}} = -\frac{\gamma \sqrt{\rho} \bar{\tau} \phi \left( \gamma \bar{\tau} \mathcal{I}_{\frac{3}{2},2}^* \rho + \psi \mathcal{I}_{\frac{1}{2},2} (\eta - \zeta) \right)}{\psi} + \sqrt{\rho} \phi G_{7,2}^{E_{32}} \mathcal{I}_{\frac{1}{2},2}$$
$$- \gamma \rho^{3/2} \bar{\tau}^2 G_{9,1}^{E_{32}} \mathcal{I}_{\frac{3}{2},2} \tag{A514}$$

$$G_{10,4}^{E_{32}} = -\frac{\gamma^2 \sqrt{\rho} \bar{\tau}^2 \left( \phi \mathcal{I}_{1,2}^* \rho + \rho \tau \psi \mathcal{I}_{1,2} (\zeta - \eta) \right)}{\psi} + \sqrt{\rho} \phi G_{7,2}^{E_{32}} \mathcal{I}_{0,2}$$
$$- \gamma \rho^{3/2} \bar{\tau}^2 G_{9,1}^{E_{32}} \mathcal{I}_{1,2} \tag{A515}$$

$$G_{10,5}^{E_{32}} = -\frac{\gamma\bar{\tau}\phi\mathcal{I}_{1,2}^*\rho}{\psi} - \rho\tau G_{7,2}^{E_{32}}\mathcal{I}_{1,2} - \rho\bar{\tau}G_{9,1}^{E_{32}}\mathcal{I}_{1,2} + x\mathcal{I}_{1,2}\left(\eta - \zeta\right) \tag{A516}$$

$$G_{10,6}^{E_{32}} = \frac{\gamma^2\rho\tau\bar{\tau}^2\mathcal{I}_{\frac{3}{2},2}^*\rho}{\psi} - \rho\tau G_{7,2}^{E_{32}}\mathcal{I}_{\frac{1}{2},2} - \rho\bar{\tau}G_{9,1}^{E_{32}}\mathcal{I}_{\frac{1}{2},2} + x\mathcal{I}_{\frac{1}{2},2}\left(\eta - \zeta\right) \tag{A517}$$

$$G_{11,3}^{E_{32}} = -\frac{\gamma\sqrt{\rho}\bar{\tau}\phi\left(\gamma\bar{\tau}\mathcal{I}_{2,2}^*\rho + \psi\mathcal{I}_{1,2}\left(\eta - \zeta\right)\right)}{\psi} + \sqrt{\rho}\phi G_{7,2}^{E_{32}}\mathcal{I}_{1,2} - \gamma\rho^{3/2}\bar{\tau}^2 G_{9,1}^{E_{32}}\mathcal{I}_{2,2} \tag{A518}$$

$$G_{11,4}^{E_{32}} = -\frac{\gamma^2\sqrt{\rho}\bar{\tau}^2\left(\phi\mathcal{I}_{\frac{3}{2},2}^*\rho + \rho\tau\psi\mathcal{I}_{\frac{3}{2},2}\left(\zeta - \eta\right)\right)}{\psi} + \sqrt{\rho}\phi G_{7,2}^{E_{32}}\mathcal{I}_{\frac{1}{2},2}$$
$$- \gamma\rho^{3/2}\bar{\tau}^2 G_{9,1}^{E_{32}}\mathcal{I}_{\frac{3}{2},2} \tag{A519}$$

$$G_{11,5}^{E_{32}} = -\frac{\gamma\bar{\tau}\phi\mathcal{I}_{\frac{3}{2},2}^*\rho}{\psi} - \rho\tau G_{7,2}^{E_{32}}\mathcal{I}_{\frac{3}{2},2} - \rho\bar{\tau}G_{9,1}^{E_{32}}\mathcal{I}_{\frac{3}{2},2} + x\mathcal{I}_{\frac{3}{2},2}\left(\eta - \zeta\right) \tag{A520}$$

$$G_{12,4}^{E_{32}} = \frac{\gamma^2\rho\tau\bar{\tau}^2\mathcal{I}_{\frac{3}{2},2}^*\rho}{\psi} - \rho\tau G_{7,2}^{E_{32}}\mathcal{I}_{\frac{1}{2},2} - \rho\bar{\tau}G_{9,1}^{E_{32}}\mathcal{I}_{\frac{1}{2},2} + \frac{x^2\mathcal{I}_{\frac{3}{2},2}\left(\zeta - \eta\right)}{\phi} \tag{A521}$$

$$G_{12,5}^{E_{32}} = \frac{\frac{x\mathcal{I}_{\frac{3}{2},2}^*\rho}{\psi} + \frac{\gamma\rho^2\tau^2\bar{\tau}\mathcal{I}_{\frac{3}{2},2}\left(\zeta - \eta\right)}{\phi}}{\sqrt{\rho}} - \frac{\sqrt{\rho}G_{9,1}^{E_{32}}\mathcal{I}_{\frac{1}{2},2}}{\gamma} + \frac{\rho^{3/2}\tau^2 G_{7,2}^{E_{32}}\mathcal{I}_{\frac{3}{2},2}}{\phi} \tag{A522}$$

$$G_{12,6}^{E_{32}} = \frac{\gamma\rho^2\tau^2\bar{\tau}\psi\mathcal{I}_{1,2}\left(\zeta - \eta\right) - x^2\mathcal{I}_{2,2}^*\rho}{\sqrt{\rho}\psi\phi} - \frac{\sqrt{\rho}G_{9,1}^{E_{32}}\mathcal{I}_{0,2}}{\gamma} + \frac{\rho^{3/2}\tau^2 G_{7,2}^{E_{32}}\mathcal{I}_{1,2}}{\phi} \tag{A523}$$

$$G_{13,3}^{E_{32}} = \frac{\gamma^2\rho\tau\bar{\tau}^2\mathcal{I}_{\frac{5}{2},2}^*\rho}{\psi} - \rho\tau G_{7,2}^{E_{32}}\mathcal{I}_{\frac{3}{2},2} - \rho\bar{\tau}G_{9,1}^{E_{32}}\mathcal{I}_{\frac{3}{2},2} + x\mathcal{I}_{\frac{3}{2},2}\left(\eta - \zeta\right) \tag{A524}$$

$$G_{13,4}^{E_{32}} = \frac{\gamma^2\rho\tau\bar{\tau}^2\mathcal{I}_{2,2}^*\rho}{\psi} - \rho\tau G_{7,2}^{E_{32}}\mathcal{I}_{1,2} - \rho\bar{\tau}G_{9,1}^{E_{32}}\mathcal{I}_{1,2} + \frac{x^2\mathcal{I}_{2,2}\left(\zeta - \eta\right)}{\phi} \tag{A525}$$

$$G_{13,5}^{E_{32}} = \frac{\frac{x\mathcal{I}_{2,2}^*\rho}{\psi} + \frac{\gamma\rho^2\tau^2\bar{\tau}\mathcal{I}_{2,2}\left(\zeta - \eta\right)}{\phi}}{\sqrt{\rho}} - \frac{\sqrt{\rho}G_{9,1}^{E_{32}}\mathcal{I}_{1,2}}{\gamma} + \frac{\rho^{3/2}\tau^2 G_{7,2}^{E_{32}}\mathcal{I}_{2,2}}{\phi} \tag{A526}$$

$$G_{13,6}^{E_{32}} = \frac{\gamma\rho^2\tau^2\bar{\tau}\psi\mathcal{I}_{\frac{3}{2},2}\left(\zeta - \eta\right) - x^2\mathcal{I}_{\frac{5}{2},2}^*\rho}{\sqrt{\rho}\psi\phi} - \frac{\sqrt{\rho}G_{9,1}^{E_{32}}\mathcal{I}_{\frac{1}{2},2}}{\gamma} + \frac{\rho^{3/2}\tau^2 G_{7,2}^{E_{32}}\mathcal{I}_{\frac{3}{2},2}}{\phi} \tag{A527}$$

$$G_{13,8}^{E_{32}} = -\frac{x\mathcal{I}_{1,1}}{\phi} \tag{A528}$$

$$G_{14,3}^{E_{32}} = \frac{x\psi\mathcal{I}_{2,2}\left(\zeta - \eta\right) - \gamma^2\rho\tau\bar{\tau}^2\mathcal{I}_{3,2}^*\rho}{\sqrt{\rho}\psi} + \sqrt{\rho}\tau G_{7,2}^{E_{32}}\mathcal{I}_{2,2} + \sqrt{\rho}\bar{\tau}G_{9,1}^{E_{32}}\mathcal{I}_{2,2} \tag{A529}$$

$$G_{14,4}^{E_{32}} = \frac{x^2\psi\mathcal{I}_{\frac{5}{2},2}\left(\eta - \zeta\right) - \gamma^2\rho\tau\bar{\tau}^2\phi\mathcal{I}_{\frac{5}{2},2}^*\rho}{\sqrt{\rho}\psi\phi} + \sqrt{\rho}\tau G_{7,2}^{E_{32}}\mathcal{I}_{\frac{3}{2},2} + \sqrt{\rho}\bar{\tau}G_{9,1}^{E_{32}}\mathcal{I}_{\frac{3}{2},2} \tag{A530}$$

$$G_{14,5}^{E_{32}} = -\frac{x\mathcal{I}_{\frac{5}{2},2}^*\rho}{\rho\psi} + \frac{G_{9,1}^{E_{32}}\mathcal{I}_{\frac{3}{2},2}}{\gamma} - \frac{\rho\tau^2 G_{7,2}^{E_{32}}\mathcal{I}_{\frac{5}{2},2}}{\phi} + \frac{\gamma\rho\tau^2\bar{\tau}\mathcal{I}_{\frac{5}{2},2}\left(\eta - \zeta\right)}{\phi} \tag{A531}$$

$$G_{14,6}^{E_{32}} = \frac{\frac{x^2\mathcal{I}_{3,2}^*\rho}{\psi} + \gamma\rho^2\tau^2\bar{\tau}\mathcal{I}_{2,2}\left(\eta - \zeta\right)}{\rho\phi} + \frac{G_{9,1}^{E_{32}}\mathcal{I}_{1,2}}{\gamma} - \frac{\rho\tau^2 G_{7,2}^{E_{32}}\mathcal{I}_{2,2}}{\phi} \tag{A532}$$

$$G_{14,8}^{E_{32}} = \frac{x\mathcal{I}_{\frac{3}{2},1}}{\sqrt{\rho}\phi} \tag{A533}$$

$$G_{14,10}^{E_{32}} = \frac{\tau\mathcal{I}_{\frac{3}{2},1}}{\phi} \tag{A534}$$

$$G_{14,11}^{E_{32}} = \frac{\tau\mathcal{I}_{1,1}}{\phi} \tag{A535}$$

$$G_{14,12}^{E_{32}} = -\frac{\mathcal{I}_{1,1}}{\sqrt{\rho}} \tag{A536}$$

$$G_{14,13}^{E_{32}} = -\frac{\mathcal{I}_{\frac{1}{2},1}}{\sqrt{\rho}} \tag{A537}$$

$$G_{15,1}^{E_{32}} = G_{14,12}^{E_{32}} \left( \sqrt{\rho}\tau^2 G_{7,2}^{E_{32}}(\eta - \zeta) + \rho\tau^2 G_{11,3}^{E_{32}} + \gamma\sqrt{\rho}\tau^2(-\bar{\tau})(\zeta - \eta)(\zeta - \eta) \right)$$
$$- \sqrt{\rho}\tau G_{14,3}^{E_{32}} \tag{A538}$$

$$G_{15,9}^{E_{32}} = -\sqrt{\rho}\tau G_{14,12}^{E_{32}} \tag{A539}$$

$$G_{11,6}^{E_{32}} = G_{12,3}^{E_{32}} = \frac{\gamma^2 \rho\tau\bar{\tau}^2 \mathcal{I}_{2,2}^* \rho}{\psi} - \rho\tau G_{7,2}^{E_{32}} \mathcal{I}_{1,2} - \rho\bar{\tau} G_{9,1}^{E_{32}} \mathcal{I}_{1,2} + x\mathcal{I}_{1,2}(\eta - \zeta) \tag{A540}$$

$$G_{8,8}^{E_{32}} = G_{14,14}^{E_{32}} = G_{15,15}^{E_{32}} = 1, \tag{A541}$$

Here we have used the definition of the LJSD $\mu$, the relations in Eqs. (A466)-(A478), the definition of $\mathcal{I}_{a,b}^*$, and the results in Sec. A9.6.1 to simplify the expressions. It is straightforward algebra to solve these equations for the undetermined entries of $G^{E_{32}}$ and thereby obtain the following expression for $E_{32}$,

$$E_{32} = \frac{(\eta - \zeta)A_{32} + \rho B_{32}}{D_{32}}, \tag{A542}$$

where,

$$\begin{aligned}
A_{32} = &-\rho^3 \tau\psi^2 x^4 \mathcal{I}_{1,1}\mathcal{I}_{2,2}^2 + \rho^2 \tau\psi x^3 \mathcal{I}_{2,2}^2(\rho\phi + x\psi(\zeta - \eta)) - \rho^3 \tau\psi^2 x^3 \phi\mathcal{I}_{1,1}\mathcal{I}_{1,2}\mathcal{I}_{2,2} \\
&+ \rho^2 \tau\psi^2 x^2 \mathcal{I}_{1,1}^2(\eta - \zeta) + \rho^2 \tau\psi^2 x^2 \mathcal{I}_{1,1}\mathcal{I}_{2,2}(\rho + x(\zeta - \eta)) \\
&+ \rho^2 \tau\psi x^2 \phi\mathcal{I}_{1,2}\mathcal{I}_{2,2}(\rho\phi + x\psi(\zeta - \eta)) + \rho^3 \tau\psi^2 x^2 \phi\mathcal{I}_{1,1}^2\mathcal{I}_{1,2} \\
&- \rho^2 \tau\psi x\phi\mathcal{I}_{1,1}\mathcal{I}_{1,2}(\rho\phi + x\psi(\zeta - \eta)) + \rho\tau\psi x\mathcal{I}_{1,1}(\zeta - \eta)(\rho\phi + x\psi(\zeta - \eta)) \\
&- \rho\tau\psi x\mathcal{I}_{2,2}(\rho + x(\zeta - \eta))(\rho\phi + x\psi(\zeta - \eta)) \tag{A543}
\end{aligned}$$

$$\begin{aligned}
B_{32} = &-\rho^2 \psi x^6 \mathcal{I}_{3,2}^* \mathcal{I}_{2,2}^2 - 2\rho^2 \psi x^5 \phi\mathcal{I}_{2,2}^* \mathcal{I}_{2,2}^2 + 2\rho\psi x^4 \phi\mathcal{I}_{3,2}^* \mathcal{I}_{1,2}(\eta - \zeta) + \rho^2 \psi x^4 \phi^2 \mathcal{I}_{3,2}^* \mathcal{I}_{1,2}^2 \\
&- 2\rho^2 \psi x^4 \phi^2 \mathcal{I}_{2,2}^* \mathcal{I}_{1,2}\mathcal{I}_{2,2} + \rho^2 \psi x^4 \phi\mathcal{I}_{1,1}^* \mathcal{I}_{2,2}^2 + \rho^2 x^4 \mathcal{I}_{3,2}^* \mathcal{I}_{2,2}(\psi + \phi) \\
&+ \rho^2 \psi x^4 \phi\mathcal{I}_{3,2}^* \mathcal{I}_{1,1}\mathcal{I}_{2,2} + \rho x^3 \phi\mathcal{I}_{2,2}^* \mathcal{I}_{2,2}(\rho(\psi + \phi) + 2x\psi(\zeta - \eta)) \\
&+ \rho^2 \psi x^3 \phi^2 \mathcal{I}_{2,2}^* \mathcal{I}_{1,1}\mathcal{I}_{1,2} + \rho^2 \psi x^3 \phi^2 \mathcal{I}_{1,1}^* \mathcal{I}_{1,2}\mathcal{I}_{2,2} + \rho\psi x^2 \phi\mathcal{I}_{1,1}^* \mathcal{I}_{1,1}(\zeta - \eta) \\
&- \rho x^2 \phi\mathcal{I}_{2,2}^* \mathcal{I}_{1,1}(\rho\phi + x\psi(\zeta - \eta)) - \rho\psi x^2 \phi\mathcal{I}_{1,1}^* \mathcal{I}_{2,2}(\rho + x(\zeta - \eta)) \\
&- \rho^2 \psi x^2 \phi^2 \mathcal{I}_{1,1}^* \mathcal{I}_{1,1}\mathcal{I}_{1,2} + \mathcal{I}_{3,2}^* \left( x^4 \psi(\zeta - \eta)^2 - \rho^2 x^2 \phi \right) \tag{A544}
\end{aligned}$$

$$\begin{aligned}
D_{32} = &-\rho^3 \psi x^4 \phi\mathcal{I}_{2,2}^2 + 2\rho^2 \psi x^2 \phi^2 \mathcal{I}_{1,2}(\eta - \zeta) + \rho^3 \psi x^2 \phi^3 \mathcal{I}_{1,2}^2 + \rho^3 x^2 \phi\mathcal{I}_{2,2}(\psi + \phi) \\
&+ \rho\phi \left( x^2 \psi(\zeta - \eta)^2 - \rho^2 \phi \right). \tag{A545}
\end{aligned}$$

Further simplifications are possible using the raising and lowering identities in Eq. (A5), as well as the results in Sec. A9.6.1, to obtain,

$$E_{32} = -\frac{\partial x}{\partial \gamma} \left( (\phi\mathcal{I}_{1,2} + \omega) \left( \frac{\mathcal{I}_{1,1}^*}{\tau} + \frac{\psi\rho}{\phi}\mathcal{I}_{1,1}(\omega + \mathcal{I}_{1,1}^*) \right) + \frac{\omega\psi x}{\phi\bar{\tau}}\mathcal{I}_{2,2} + \frac{\phi}{x\bar{\tau}}\mathcal{I}_{1,2}^* - \frac{\omega x}{\tau}\mathcal{I}_{2,2}^* \right), \tag{A546}$$

where

$$\frac{\partial x}{\partial \gamma} = -\frac{x}{\gamma + \rho\gamma(\tau\psi/\phi + \bar{\tau})(\omega + \phi\mathcal{I}_{1,2})}. \tag{A547}$$

**A9.6.5** $E_4$

Define the block matrix $Q^{E_4} \equiv [Q_1^{E_4} \ Q_2^{E_4}]$ by,

$$Q_1^{E_4} = \begin{pmatrix} I_m & \frac{\sqrt{\eta-\zeta}\Theta_{22}^\top}{\gamma\sqrt{n_1}} & 0 & 0 & \frac{\sqrt{\rho}X_2^\top}{\gamma\sqrt{n_0}} & 0 & 0 \\ -\frac{\Theta_{22}\sqrt{\eta-\zeta}}{\sqrt{n_1}} & I_{n_1} & -\frac{\sqrt{\rho}W_2}{\sqrt{n_1}} & 0 & 0 & 0 & 0 \\ 0 & 0 & I_{n_0} & -\Sigma^{1/2} & 0 & 0 & 0 \\ -\frac{X_2}{\sqrt{n_0}} & 0 & 0 & I_{n_0} & 0 & 0 & 0 \\ 0 & 0 & 0 & 0 & I_{n_0} & -\Sigma^{1/2} & 0 \\ 0 & -\frac{W_2^\top}{\sqrt{n_1}} & 0 & 0 & 0 & I_{n_0} & \frac{\Sigma^{1/2}}{\sqrt{\rho}} \\ 0 & 0 & 0 & 0 & 0 & 0 & I_{n_0} \\ 0 & 0 & 0 & 0 & 0 & 0 & 0 \\ 0 & 0 & 0 & 0 & 0 & 0 & 0 \\ 0 & 0 & 0 & 0 & 0 & 0 & 0 \\ 0 & 0 & 0 & 0 & 0 & 0 & -\Sigma^{1/2} \\ 0 & 0 & 0 & 0 & 0 & 0 & 0 \\ 0 & 0 & 0 & 0 & 0 & 0 & 0 \end{pmatrix}, \quad \text{(A548)}$$

and,

$$Q_2^{E_4} = \begin{pmatrix} 0 & 0 & 0 & 0 & 0 & 0 & 0 \\ 0 & 0 & 0 & 0 & 0 & 0 & 0 \\ 0 & 0 & 0 & 0 & 0 & 0 & 0 \\ 0 & 0 & 0 & 0 & 0 & 0 & 0 \\ 0 & 0 & 0 & 0 & 0 & 0 & 0 \\ 0 & 0 & 0 & 0 & 0 & 0 & 0 \\ -\frac{X_1}{\sqrt{n_0}} & 0 & 0 & 0 & 0 & 0 & 0 \\ I_m & \frac{\sqrt{\eta-\zeta}\Theta_{11}^\top}{\gamma\sqrt{n_1}} & \frac{\sqrt{\rho}X_1^\top}{\gamma\sqrt{n_0}} & 0 & 0 & 0 & 0 \\ -\frac{\Theta_{11}\sqrt{\eta-\zeta}}{\sqrt{n_1}} & I_{n_1} & 0 & 0 & -\frac{\sqrt{\rho}W_1}{\sqrt{n_1}} & 0 & 0 \\ 0 & 0 & I_{n_0} & -\Sigma^{1/2} & 0 & 0 & 0 \\ 0 & -\frac{W_1^\top}{\sqrt{n_1}} & 0 & I_{n_0} & 0 & 0 & 0 \\ 0 & 0 & 0 & 0 & I_{n_0} & \frac{\Sigma^*}{\sqrt{\rho}} & 0 \\ 0 & 0 & 0 & 0 & 0 & I_{n_0} & -\frac{W_2^\top}{\sqrt{n_1}} \\ 0 & 0 & 0 & 0 & 0 & 0 & I_{n_1} \end{pmatrix}. \quad \text{(A549)}$$

Then block matrix inversion (i.e. repeated applications of the Schur complement formula) shows that,

$$G_{13,13}^{E_4} = G_{14,14}^{E_4} = 1 \tag{A550}$$

$$G_{1,1}^{E_4} = G_{8,8}^{E_4} = G_{1,1}^{K^{-1}} \tag{A551}$$

$$G_{2,2}^{E_4} = G_{9,9}^{E_4} = G_{2,2}^{K^{-1}} \tag{A552}$$

$$G_{3,3}^{E_4} = G_{6,6}^{E_4} = G_{11,11}^{E_4} = G_{12,12}^{E_4} = G_{4,4}^{E_4} = G_{5,5}^{E_4} = G_{7,7}^{E_4} = G_{10,10}^{E_4} = G_{3,3}^{K^{-1}} \tag{A553}$$

$$G_{3,4}^{E_4} = G_{5,6}^{E_4} = G_{10,11}^{E_4} = G_{12,7}^{E_4} = G_{3,4}^{K^{-1}} \tag{A554}$$

$$G_{5,3}^{E_4} = G_{6,4}^{E_4} = G_{10,12}^{E_4} = G_{11,7}^{E_4} = G_{3,5}^{K^{-1}} \tag{A555}$$

$$G_{5,4}^{E_4} = G_{10,7}^{E_4} = G_{3,6}^{K^{-1}} \tag{A556}$$

$$G_{4,3}^{E_4} = G_{6,5}^{E_4} = G_{7,12}^{E_4} = G_{11,10}^{E_4} = G_{4,3}^{K^{-1}} \tag{A557}$$

$$G_{6,3}^{E_4} = G_{11,12}^{E_4} = G_{4,5}^{K^{-1}} \tag{A558}$$

$$G_{3,5}^{E_4} = G_{4,6}^{E_4} = G_{7,11}^{E_4} = G_{12,10}^{E_4} = G_{5,3}^{K^{-1}} \tag{A559}$$

$$G_{3,6}^{E_4} = G_{12,11}^{E_4} = G_{5,4}^{K^{-1}} \tag{A560}$$

$$G_{4,5}^{E_4} = G_{7,10}^{E_4} = G_{6,3}^{K^{-1}} \tag{A561}$$

$$G_{14,2}^{E_4} = \frac{\psi}{\rho} E_4, \tag{A562}$$

where $G_{i,j}^{E_4}$ denotes the normalized trace of the $(i,j)$-block of the inverse of $\left(Q^{E_4}\right)^\top$. For brevity, we have suppressed the expressions for the other non-zero blocks.

To compute the limiting values of these traces, we require the asymptotic block-wise traces of $Q^{E_4}$, which may be determined from the operator-valued Stieltjes transform. To proceed, we first augment $Q^{E_4}$ to form the the self-adjoint matrix $\bar{Q}^{E_4}$,

$$\bar{Q}^{E_4} = \begin{pmatrix} 0 & [Q^{E_4}]^\top \\ Q^{E_4} & 0 \end{pmatrix}. \tag{A563}$$

and observe that we can write $\bar{Q}^{E_4}$ as,

$$\begin{aligned}
\bar{Q}^{E_4} &= \bar{Z} - \bar{Q}_{W,X,\Theta}^{E_4} - \bar{Q}_\Sigma^{E_4} \\
&= \begin{pmatrix} 0 & Z^\top \\ Z & 0 \end{pmatrix} - \begin{pmatrix} 0 & [Q_{W,X,\Theta}^{E_4}]^\top \\ Q_{W,X,\Theta}^{E_4} & 0 \end{pmatrix} - \begin{pmatrix} 0 & [Q_\Sigma^{E_4}]^\top \\ Q_\Sigma^{E_4} & 0 \end{pmatrix},
\end{aligned} \tag{A564}$$

where $Z = I_{2m+9n_0+3n_1}$, $Q_{W,X,\Theta}^{E_4} \equiv [[Q_{W,X,\Theta}^{E_4}]_1 \ [Q_{W,X,\Theta}^{E_4}]_2]$ and,

$$[Q_{W,X,\Theta}^{E_4}]_1 = \begin{pmatrix}
I_m & \frac{\sqrt{\eta-\zeta}\Theta_{22}^\top}{\gamma\sqrt{n_1}} & 0 & 0 & \frac{\sqrt{\rho}X_2^\top}{\gamma\sqrt{n_0}} & 0 & 0 \\
-\frac{\Theta_{22}\sqrt{\eta-\zeta}}{\sqrt{n_1}} & I_{n_1} & -\frac{\sqrt{\rho}W_2}{\sqrt{n_1}} & 0 & 0 & 0 & 0 \\
0 & 0 & I_{n_0} & 0 & 0 & 0 & 0 \\
-\frac{X_2}{\sqrt{n_0}} & 0 & 0 & I_{n_0} & 0 & 0 & 0 \\
0 & 0 & 0 & 0 & I_{n_0} & 0 & 0 \\
0 & -\frac{W_2^\top}{\sqrt{n_1}} & 0 & 0 & 0 & I_{n_0} & 0 \\
0 & 0 & 0 & 0 & 0 & 0 & I_{n_0} \\
0 & 0 & 0 & 0 & 0 & 0 & 0 \\
0 & 0 & 0 & 0 & 0 & 0 & 0 \\
0 & 0 & 0 & 0 & 0 & 0 & 0 \\
0 & 0 & 0 & 0 & 0 & 0 & 0 \\
0 & 0 & 0 & 0 & 0 & 0 & 0 \\
0 & 0 & 0 & 0 & 0 & 0 & 0
\end{pmatrix} \tag{A565}$$

$$[Q_{W,X,\Theta}^{E_4}]_2 = \begin{pmatrix}
0 & 0 & 0 & 0 & 0 & 0 & 0 \\
0 & 0 & 0 & 0 & 0 & 0 & 0 \\
0 & 0 & 0 & 0 & 0 & 0 & 0 \\
0 & 0 & 0 & 0 & 0 & 0 & 0 \\
0 & 0 & 0 & 0 & 0 & 0 & 0 \\
0 & 0 & 0 & 0 & 0 & 0 & 0 \\
-\frac{X_1}{\sqrt{n_0}} & 0 & 0 & 0 & 0 & 0 & 0 \\
I_m & \frac{\sqrt{\eta-\zeta}\Theta_{11}^\top}{\gamma\sqrt{n_1}} & \frac{\sqrt{\rho}X_1^\top}{\gamma\sqrt{n_0}} & 0 & 0 & 0 & 0 \\
-\frac{\Theta_{11}\sqrt{\eta-\zeta}}{\sqrt{n_1}} & I_{n_1} & 0 & 0 & -\frac{\sqrt{\rho}W_1}{\sqrt{n_1}} & 0 & 0 \\
0 & 0 & I_{n_0} & 0 & 0 & 0 & 0 \\
0 & -\frac{W_1^\top}{\sqrt{n_1}} & 0 & I_{n_0} & 0 & 0 & 0 \\
0 & 0 & 0 & 0 & I_{n_0} & 0 & 0 \\
0 & 0 & 0 & 0 & 0 & I_{n_0} & -\frac{W_2^\top}{\sqrt{n_1}} \\
0 & 0 & 0 & 0 & 0 & 0 & I_{n_1}
\end{pmatrix} \tag{A566}$$

$$Q_\Sigma^{E_4} = \begin{pmatrix}
0 & 0 & 0 & 0 & 0 & 0 & 0 & 0 & 0 & 0 & 0 & 0 & 0 & 0 \\
0 & 0 & 0 & 0 & 0 & 0 & 0 & 0 & 0 & 0 & 0 & 0 & 0 & 0 \\
0 & 0 & 0 & -\Sigma^{1/2} & 0 & 0 & 0 & 0 & 0 & 0 & 0 & 0 & 0 & 0 \\
0 & 0 & 0 & 0 & 0 & 0 & 0 & 0 & 0 & 0 & 0 & 0 & 0 & 0 \\
0 & 0 & 0 & 0 & 0 & -\Sigma^{1/2} & 0 & 0 & 0 & 0 & 0 & 0 & 0 & 0 \\
0 & 0 & 0 & 0 & 0 & 0 & \frac{\Sigma^{1/2}}{\sqrt{\rho}} & 0 & 0 & 0 & 0 & 0 & 0 & 0 \\
0 & 0 & 0 & 0 & 0 & 0 & 0 & 0 & 0 & 0 & 0 & 0 & 0 & 0 \\
0 & 0 & 0 & 0 & 0 & 0 & 0 & 0 & 0 & 0 & 0 & 0 & 0 & 0 \\
0 & 0 & 0 & 0 & 0 & 0 & 0 & 0 & 0 & 0 & 0 & 0 & 0 & 0 \\
0 & 0 & 0 & 0 & 0 & 0 & 0 & 0 & 0 & 0 & -\Sigma^{1/2} & 0 & 0 & 0 \\
0 & 0 & 0 & 0 & 0 & 0 & 0 & 0 & 0 & 0 & 0 & 0 & 0 & 0 \\
0 & 0 & 0 & 0 & 0 & 0 & -\Sigma^{1/2} & 0 & 0 & 0 & 0 & 0 & \frac{\Sigma^*}{\sqrt{\rho}} & 0 \\
0 & 0 & 0 & 0 & 0 & 0 & 0 & 0 & 0 & 0 & 0 & 0 & 0 & 0 \\
0 & 0 & 0 & 0 & 0 & 0 & 0 & 0 & 0 & 0 & 0 & 0 & 0 & 0
\end{pmatrix}. \tag{A567}$$

The operator-valued Stieltjes transforms satisfy,

$$\bar{G}^{E_4} = \bar{G}^{E_4}_\Sigma (\bar{Z} - \bar{R}^{E_4}_{W,X,\Theta}(\bar{G}^{E_4}))$$

$$= \mathrm{id} \otimes \bar{\mathrm{tr}} \left( \bar{Z} - \bar{R}^{E_4}_{W,X,\Theta}(\bar{G}^{E_4}) - \bar{Q}^{E_4}_\Sigma \right)^{-1} , \tag{A568}$$

where $\bar{R}^{E_4}_{W,X,\Theta}(\bar{G}^{E_4})$ is the operator-valued R-transform of $\bar{Q}^{E_4}_{W,X,\Theta}$. As discussed above, since $\bar{Q}^{E_4}_{W,X,\Theta}$ is a block matrix whose blocks are i.i.d. Gaussian matrices (and their transposes), an explicit expression for $\bar{R}^{E_4}_{W,X,\Theta}(\bar{G}^{E_4})$ can be obtained from the covariance map $\eta$, which can be read off from Eq. (A563). As above, we use the specific sparsity pattern for $G^{E_4}$ that is induced by Eq. (A568), to obtain,

$$\bar{R}^{E_4}_{W,X,\Theta}(\bar{G}^{E_4}) = \begin{pmatrix} 0 & R^{E_4}_{W,X,\Theta}(G^{E_4})^\top \\ R^{E_4}_{W,X,\Theta}(G^{E_4}) & 0 \end{pmatrix} , \tag{A569}$$

where,

$$[R^{E_4}_{W,X,\Theta}(G^{E_4})]_{1,1} = \frac{G^{E_4}_{2,2}(\zeta - \eta)}{\gamma} - \frac{\sqrt{\rho} G^{E_4}_{4,5}}{\gamma} \tag{A570}$$

$$[R^{E_4}_{W,X,\Theta}(G^{E_4})]_{2,2} = \frac{\psi G^{E_4}_{1,1}(\zeta - \eta)}{\gamma \phi} + \sqrt{\rho}\psi G^{E_4}_{6,3} \tag{A571}$$

$$[R^{E_4}_{W,X,\Theta}(G^{E_4})]_{2,14} = \sqrt{\rho}\psi G^{E_4}_{13,3} \tag{A572}$$

$$[R^{E_4}_{W,X,\Theta}(G^{E_4})]_{4,5} = -\frac{\sqrt{\rho} G^{E_4}_{1,1}}{\gamma \phi} \tag{A573}$$

$$[R^{E_4}_{W,X,\Theta}(G^{E_4})]_{6,3} = \sqrt{\rho} G^{E_4}_{2,2} \tag{A574}$$

$$[R^{E_4}_{W,X,\Theta}(G^{E_4})]_{7,10} = -\frac{\sqrt{\rho} G^{E_4}_{8,8}}{\gamma \phi} \tag{A575}$$

$$[R^{E_4}_{W,X,\Theta}(G^{E_4})]_{8,8} = \frac{G^{E_4}_{9,9}(\zeta - \eta)}{\gamma} - \frac{\sqrt{\rho} G^{E_4}_{7,10}}{\gamma} \tag{A576}$$

$$[R^{E_4}_{W,X,\Theta}(G^{E_4})]_{9,9} = \frac{\psi G^{E_4}_{8,8}(\zeta - \eta)}{\gamma \phi} + \sqrt{\rho}\psi G^{E_4}_{11,12} \tag{A577}$$

$$[R^{E_4}_{W,X,\Theta}(G^{E_4})]_{11,12} = \sqrt{\rho} G^{E_4}_{9,9} \tag{A578}$$

$$[R^{E_4}_{W,X,\Theta}(G^{E_4})]_{13,3} = \sqrt{\rho} G^{E_4}_{2,14} , \tag{A579}$$

and the remaining entries of $R^{E_4}_{W,X,\Theta}(G^{E_4})$ are zero. Similarly, following the example from $G^{E_{21}}$ above, plugging these expressions into Eq. (A568) and explicitly performing the block-matrix inverse yields the following set of coupled equations,

$$G^{E_4}_{7,5} = -\frac{\phi \mathcal{I}_{1,2}}{\sqrt{\rho}} \tag{A580}$$

$$G^{E_4}_{7,6} = -\frac{\phi \mathcal{I}_{\frac{1}{2},2}}{\sqrt{\rho}} \tag{A581}$$

$$G^{E_4}_{10,4} = -\frac{\sqrt{\rho}\tau^2 \mathcal{I}_{1,2}}{\phi} \tag{A582}$$

$$G^{E_4}_{10,6} = \tau \mathcal{I}_{\frac{1}{2},2} \tag{A583}$$

$$G^{E_4}_{11,3} = -\frac{\sqrt{\rho}\tau^2 \mathcal{I}_{2,2}}{\phi} \tag{A584}$$

$$G^{E_4}_{12,3} = -\frac{\gamma\rho\tau^2 \bar{\tau} \mathcal{I}_{2,2}}{\phi} \tag{A585}$$

$$G^{E_4}_{12,4} = -\frac{\gamma\rho\tau^2 \bar{\tau} \mathcal{I}_{\frac{3}{2},2}}{\phi} \tag{A586}$$

$$G_{12,5}^{E_4} = \frac{x\mathcal{I}_{\frac{3}{2},2}}{\sqrt{\rho}} \tag{A587}$$

$$G_{12,6}^{E_4} = \frac{x\mathcal{I}_{1,2}}{\sqrt{\rho}} \tag{A588}$$

$$G_{13,3}^{E_4} = \frac{\gamma\sqrt{\rho}\tau^2\bar{\tau}\mathcal{I}_{3,2}^*}{\phi} \tag{A589}$$

$$G_{13,4}^{E_4} = \frac{\gamma\sqrt{\rho}\tau^2\bar{\tau}\mathcal{I}_{\frac{5}{2},2}^*}{\phi} \tag{A590}$$

$$G_{13,5}^{E_4} = -\frac{x\mathcal{I}_{\frac{5}{2},2}^*}{\rho} \tag{A591}$$

$$G_{13,6}^{E_4} = -\frac{x\mathcal{I}_{2,2}^*}{\rho} \tag{A592}$$

$$G_{13,7}^{E_4} = \frac{x\mathcal{I}_{\frac{3}{2},1}^*}{\sqrt{\rho}\phi} \tag{A593}$$

$$G_{13,10}^{E_4} = \gamma(-\bar{\tau})\mathcal{I}_{\frac{3}{2},1}^* \tag{A594}$$

$$G_{13,11}^{E_4} = \gamma(-\bar{\tau})\mathcal{I}_{1,1}^* \tag{A595}$$

$$G_{13,12}^{E_4} = -\frac{\mathcal{I}_{1,1}^*}{\sqrt{\rho}} \tag{A596}$$

$$G_{14,2}^{E_4} = \gamma\sqrt{\rho}\bar{\tau}\psi G_{13,3}^{E_4} \tag{A597}$$

$$G_{7,3}^{E_4} = G_{11,5}^{E_4} = \tau\mathcal{I}_{\frac{3}{2},2} \tag{A598}$$

$$G_{10,3}^{E_4} = G_{11,4}^{E_4} = -\frac{\sqrt{\rho}\tau^2\mathcal{I}_{\frac{3}{2},2}}{\phi} \tag{A599}$$

$$G_{13,13}^{E_4} = G_{14,14}^{E_4} = 1 \tag{A600}$$

$$G_{7,4}^{E_4} = G_{10,5}^{E_4} = G_{11,6}^{E_4} = \tau\mathcal{I}_{1,2}\,, \tag{A601}$$

Here we have used the definition of the LJSD $\mu$, the relations in Eqs. (A550)-(A562), the definition of $\mathcal{I}_{a,b}^*$, and the results in Sec. A9.6.1, to simplify the expressions. It is straightforward algebra to solve these equations for the undetermined entries of $G^{E_4}$ and thereby obtain the following expression for $E_4$,

$$E_4 = \frac{x^2}{\phi}\mathcal{I}_{3,2}^*\,. \tag{A602}$$

### A9.6.6 Results for total error, bias, and variance

General expressions for the total error, bias and variance in terms of $E_{21}, E_{31}, E_{32}, E_4$ can be found in Sec. A9.5. Combining the results from Sec. A9.5 and Eqs. (A407), (A461) and (A546) yields expressions for the bias and total error. Using $E_\mu = B_\mu + V_\mu$, these results also determine the variance. Putting all the pieces together we find,

$$B_\mu = \phi\mathcal{I}_{1,2}^* \tag{A603}$$

$$V_\mu = -\rho\frac{\psi}{\phi}\frac{\partial x}{\partial \gamma}\left(\mathcal{I}_{1,1}(\omega + \phi\mathcal{I}_{1,2})(\omega + \mathcal{I}_{1,1}^*) + \frac{\phi^2}{\psi}\gamma\bar{\tau}\mathcal{I}_{1,2}\mathcal{I}_{2,2}^* + \gamma\tau\mathcal{I}_{2,2}(\omega + \phi\mathcal{I}_{1,2}^*)\right.$$

$$\left. + \sigma_\varepsilon^2\left((\omega + \phi\mathcal{I}_{1,2})(\omega + \mathcal{I}_{1,1}^*) + \frac{\phi}{\psi}\gamma\bar{\tau}\mathcal{I}_{2,2}^*\right)\right) \tag{A604}$$

$$\tag{A605}$$

where $x$ is the unique positive real root of $x = \frac{1-\gamma\tau}{\omega+\mathcal{I}_{1,1}}$, as in Eq. (A345). The derivative $\frac{\partial x}{\partial \gamma}$ follows from implicit differentiation and was given in Eq. (A463),

$$\frac{\partial x}{\partial \gamma} = -\frac{x}{\gamma + \rho\gamma(\tau\psi/\phi + \bar{\tau})(\omega + \phi\mathcal{I}_{1,2})}\,. \tag{A606}$$

The asymptotic trace objects $\tau$ and $\bar{\tau}$ were defined in Eq. (A343) and Eq. (A344) and are given by,

$$\tau = \frac{\sqrt{(\psi - \phi)^2 + 4x\psi\phi\gamma/\rho} + \psi - \phi}{2\psi\gamma} \quad \text{and} \quad \bar{\tau} = \frac{1}{\gamma} + \frac{\psi}{\phi}\left(\tau - \frac{1}{\gamma}\right). \qquad \text{(A607)}$$

All together, these results prove Thm. 5.1.