# OpenReview forum: "Overparameterization Improves Robustness to Covariate Shift in High Dimensions"
_NeurIPS.cc/2021/Conference — NeurIPS 2021 Poster_

### Official Review · Reviewer_oyKe · 2021-07-04

**Rating:** 7
**Confidence:** 3

**Summary:**

This paper presents a theoretical study of covariate shift under the setting of kernel ridge regression in the asymptotic regime. They charaterize the limiting test error, bias, and variance in this setting. They relates these with a quantity named hardness. They claim that their findings agree with several interesting empirical phenomena.


**Limitations And Societal Impact:**

I feel some assumptions of the papers are not sufficiently justified/explained. Please see specific comment 1.


**Main Review:**

Originality: The paper is original. They proposed a way to quantify the hardness of covariate shift and related this to the Bayes risk in kernel ridge regression.

Quality: The quality is good. Theoretical results have solid proof as far as I can tell.

Clarity: There is some room for improvement. Specifically, the paper is a little hard to read because there is not enough explanation/intuition for the presented theorems, especially for thm2 and coro1. It is unclear what those many terms in the expressions mean. It might be better to add some explanation or point out to the readers where/which terms/what they should focus on.

Significance: This is a good contribution to an important problem.


Specific comments:
1. Is there any literature/computation/examples that can justify the spectral assumptions? Specifically, I find the assumptions in the part from line 125 to line 151 not sufficiently justified or explained. For example, it might be better to provide some nontrivial cases in which the limit LJSD exists, or r_2/r_1 is monotonic, etc.
2. There are several closely related works regarding generalization. First, the kernel regression setting studied in this paper seems closely related to arxiv.org/abs/1908.10292. It would be good to see a discussion about the relation/comparison with that work. Also, the study of the Bayes risk is close to that in https://arxiv.org/abs/2008.01036.
3. This paper consider the Bayes risk (i.e. expectation w.r.t. \beta). Is it possible to consider the case here each dimension of \beta is pre-fixed?


**Time Spent Reviewing:**

4

---

> ### Author Response · Authors · 2021-08-11
> **Response to Reviewer oyKe**
>
> We are appreciative of the reviewer’s time and effort in reviewing our manuscript. We agree that our paper makes original contributions and is of good quality, and we answer the specific comments below. We hope this persuades the reviewer to update the overall rating.
>
> - “Spectral Assumptions”/“Assumption 1”: As is characteristic of asymptotic results in RMT, some specification of the limiting behavior of the relevant spectral objects is necessary. An assumption equivalent to Assumption 1 has been used recently in the study of overparameterized linear regression with weighted regularizers (https://arxiv.org/pdf/2006.05800.pdf), where it was also necessary to specify the limiting behavior of matrices that do not have the same eigenspaces. When the eigenspaces of the train and test covariances are aligned, Assumption 1 reduces to a standard assumption that is very common in prior literature using RMT to study the asymptotic test error of ridge regression and random feature models (e.g. the cited references in the related work section).
> - Theorem 2:  We recognize the complexity of Theorem 2 and have further streamlined its presentation and included additional motivation for the terms that arise. However, we note the lack of an explicit closed-form solution for the test error is present even in the case of *ridge regression without any covariate shift*--see for example Theorem 2.1.2 in https://arxiv.org/abs/1507.03003. The main value of Theorem 2 is its ability to make precise predictions of the test error, bias, and variance that are not loose upper bounds. We hope the later propositions help elucidate the phenomenology that can be explored using Theorem 2. The term $\mathcal{I}^\star$ is the high dimensional limit of functionals arising in the random feature test error similar to how $\mathrm{tr}(\Sigma^* \Sigma) \to \mathbb{E}_{\mu}[r/\lambda]$ arises in the test error of linear regression.
> - Additional references: We thank Reviewer oyKe for pointing out these additional references. One principal difference between these works and ours is that we explicitly study the setting of covariate shift in the random design setting, which is not addressed in these previous works.
> - “Bayes Risk wrt to $\beta$”: Our results rely on an expectation over an isotropic $\beta$. Recent work in the setting of ridge regression has provided asymptotic computations of the test error for more general (but random) anisotropic $\beta$ (https://arxiv.org/pdf/2006.05800.pdf). We believe it may be possible to generalize our results to either fixed or nonanistropic $\beta$ but this requires additional technical work.

---

> > ### Comment · Reviewer_oyKe · 2021-08-18
> > **recommend accept**
> >
> > Dear authors,
> >
> > Thank you very much for your effort in the response. It addressed my questions and I raised my score and recommended accept.

---

### Official Review · Reviewer_rcgs · 2021-07-11

**Rating:** 7
**Confidence:** 3

**Summary:**

This work provides theoretical insight for why overparameterized models perform well when there is some covariate shift between the training and test sets. The authors' characterize their results by introducing a notion of "relative hardness". They present empirical evidence to match their intuition, and corroborate the main results in the paper using several empirical benchmarks.

**Limitations And Societal Impact:**

The authors have pointed out some limitations of their analysis in the conclusion section of this submission.

**Main Review:**

### Originality
I am not too familiar with works related to analysing test error when there is covariate shift. Nonetheless, I find the results and the technique of working with random features interesting.

### Quality
Introduction is well written, and the motivation of the problem is sufficiently detailed. The notion of a class of covariate shifts is an interesting generalization theoretically, and subsequent results are empirically verified using simply experimental setups. The submission is technically sound, and complete, with a few future directions that are beyond the scope of this submission.

### Clarity
Section 2.2 could generally be simplified, as there are plenty of equations or perhaps it would be better if the authors introduced these equations in a sequential manner and explain the purpose or intuition behind them. For instance the authors claim that $\eta, \eta\_{\*}$ capture the effect of activation non-linearity (these seem to be $\text{Var}(\sigma(\sqrt{s} z)), \text{Var}(\sigma(\sqrt{s\_{\*}} z))$ respectively). There is also some undefined notation: $\mathbb{C}^{+}$ (complex numbers with real positive parts?), $\gamma$, $\mathfrak{R}$ (real part map?). Additionally what does it mean for $E\_{\Sigma^{\*}} \to E\_{\mu}$ to be equal to another limit? Unfortunately, a majority of Page 6 seems to be hard to understand: while Theorem 2 is general, it is not immediately clear what the apparent purpose of the Theorem is, and the authors could have added a prologue regarding this.

### Significance
While mathematically interesting, the asymptotic setting is not very useful practically. For instance, it would be better to obtain non-asymptotic rates, and this might also help corroborate empirical phenomena. The authors also highlight this in the final section of the paper.

-----

Some additional detailed comments / questions:

- The model being discussed here is covariate shift where distribution of the $X_{train} \neq$ distribution of $X_{test}$, but distribution $y | X_{train} =$ distribution $y | X_{test}$ . Specifically, here the authors are looking at the following model: $X_{train} \overset{i.i.d}\sim \mathcal{N}(0, \Sigma)$ whereas $X^{\star} \sim \mathcal{N}(0, \Sigma^{\star})$. Why not study: $X_{train} \overset{i.i.d}\sim \mathcal{N}(\mu, \Sigma)$ whereas $X^{\star} \sim \mathcal{N}(\mu^{\star}, \Sigma^{\star})$. Is it because centering the data effectively reduces the aforementioned scenario to the one studied in the paper?

- The notion of a class of covariate shifts is an interesting generalization theoretically. However, in definition 1, it is not particularly clear how monotonicity with respect to a random variable is defined (in this case, w.r.t. $\lambda$) [Line 175/176].

- Line 115 / 150: $\sigma : \mathbb{R}^{n_{0}} \to \mathbb{R}^{n_{0}}$ or $\sigma : \mathbb{R} \to \mathbb{R}$ since it's an elementwise operation.

- Line 132: dimension-normalized trace: this appears to be $tr(A) / n$ if $A \in \mathbb{R}^{n \times n}$ but could help to specify directly early on?

- Equation 7: $\delta_{a,b}$ is undefined. Is this $\infty$ if $a = b$ and $0$ otherwise?

- In assumption 1, are we considering asymptotic convergence of the empirical measures as the dimension tends to $\infty$, because $n_{0}$ is also the size of the covariance in the model. If so, any convergence results also would only make sense if the dimensionality of the inputs increased towards $\infty$, correct?

- The remarks regarding the implications of Proposition 1 are interesting and make sense, and it is also encouraging to see this partial ordering being satisfied empirical for the identity covariances (and scaled versions) in Fig 1a. Naturally this gives rise to Definition 1 which characterizes difficulty. The family of power-law distributional shifts is well motivated.

- Figure 1b seems to be in agreement with Proposition 3 (similar to Figure 1a with the scaling), and it is again encouraging to see that such patterns can be quantified.

- Proposition 4 is very interesting and non-trivial as the authors remark, but what does the hardness of $\mu$ mean? Is it with respect to $\mu_{\emptyset}$?

- Proposition 5 also provides an explanation of phenomena observed in Figures 3a and 3b. This proposition is also useful and practically applicable with the proliferation of over parameterized methods, again as noted by the authors.

- In conclusion, drawing from the random feature model, the authors have presented an explanation for why overparameterized models perform well in covariate shift. Large scale experiments on CIFAR-10 corroborate their theory: larger shift is hard to work with, but can be partially alleviated with the use of higher number of parameters. While this is not an accurate representation of the input scenario (neural networks), the random feature regression model seems to produce conclusions that are also applicable to such large scale models.

**Time Spent Reviewing:**

4

---

> ### Author Response · Authors · 2021-08-11
> **Response to Reviewer rcgs**
>
> We thank the reviewer for the many comments, questions, and suggestions to improve our paper. We have used them to make the manuscript clearer and more concise, and we attempt to address any outstanding issues below.
>
> - Notation and clarity: $E_{\mu}$ is the limit of $E_{\Sigma^{*}}$ as the dimension parameters go to infinity. Theorem 2 states that this limiting object, $E_{\mu}$, can also be obtained as the limit in Eqn. (12), i.e. the two limits are the same. However, Eqn. (12) can be solved numerically, allowing for $E_{\mu}$ to be calculated. We will also clarify that $\mathbb{C}^{+}$ is the complex plane with real positive part, $\gamma$ denotes the ridge regularization parameter as defined in the problem setup and notation, and $\mathfrak{R}$, $\mathfrak{I}$ denotes the real and imaginary parts of a number respectively.
> - “Mean Shift/Centering”: Yes, we chose not to explicitly include a mean shift since the centering condition is often explicitly enforced as a pre-processing step in most practical settings. Moreover, a mean shift adds complexity to the model without resulting in interesting new behavior. However, we note that adding a mean shift is a trivial extension of our techniques, and there is no technical reason preventing us from doing so.
>  - “Monotonicity with respect to $\lambda$”: Here, we mean that we are conditioning on the event that $\lambda=x$ for some real number $x$, and that as a function of $x$, the ratio of asymptotic overlap coefficients in Definition 1 is monotonic. We have clarified this in a new version of the paper.
> - Line 115: We thank Reviewer rcgs for pointing out this typo and have corrected it.
> - Line 132: Yes, we will formally define the dimension-normalized trace as $\mathrm{tr}(A) = \frac{1}{n} \mathrm{tr}(A)$ for $A \in \mathbb{R}^{n \times n}$.
> - “$\delta_{(a,b)}$”: We will clarify this means the standard measure-theoretic definition of an atom or point mass at the point $(a,b)$ in $\mathbb{R}^2$.
> - “Asymptotic Convergence”: Yes, the theoretical results here assume the proportional asymptotics where the size of the covariance $n_0$ tends to infinity as do the sample size $m$, and number of random features $n_1$.
> - “Proposition 4/hard”: Here we intend hard to be defined in the formal sense of Definition 1. That is, $\mu$ is hard, if $\mu$ is harder then $\mu_{\emptyset}$ (with hardness defined in Definition 1).

---

> > ### Comment · Reviewer_rcgs · 2021-08-17
> > **Thank you for your response**
> >
> > Thank you authors for your response. I have gone through them, and will stick with my current rating.

---

### Official Review · Reviewer_9Mq1 · 2021-07-15

**Rating:** 6
**Confidence:** 1

**Summary:**

The paper studies the test error, as well as the bias and variance, of random feature regression in the asymptotic regime, in a setting with covariate shift — i.e., when the covariance matrix of the features is different at testing time than at training time. The paper defines a partial ordering over covariance shifts, and it proves that this corresponds to increased or decreased performance compared to the case of no shift. It also obtains a few results that relate these notions to overparameterization.

**Limitations And Societal Impact:**

The paper seems fine to me in this regard.

**Main Review:**

One concern is whether the partial ordering in Definition 1 can capture interesting shifts. Most of the results depend on the partial ordering, so it is crucial for this to be the case. However, at first sight, the definition looks very restrictive: it requires ratios of overlap coefficients and eigenvalues to form a non-increasing sequence. It’s not obvious at all when this should be the case. The only example given, Example 1, is unfortunately trivial in this respect: the covariance matrices are diagonal (so the eigenspaces are aligned, which makes the definition much easier) and they have only two different eigenvalues. I feel that it would be crucial to give examples of more interesting families of covariate shifts for which the definition is meaningful; in particular, ones where the eigenspaces are not aligned and where the eigenvalues are more different. I feel that it would also be important to explicitly mention common shifts that cannot be captured by the definition; e.g., unless I am mistaken, this is the case with rotations, which seems an important limitation, given how common such transformations are.

Another concern is that Theorem 2 and even Corollary 1 are extremely complicated. I understand that the advantage of presenting them like this is that they are exact. But it also makes the results very difficult to understand and interpret.

In fact, the paper does not give any explanation of the intuition or of the proof techniques for the result in the random feature regression setting. The only explanation, given in Section 2.1, is about linear regression — but, based on the related work section, it seems that this setting has already been studied. The contribution of this paper is supposed to be to extend such results to the random feature regression setting. So, I can only conclude that the paper gives no explanation at all for its actual contribution — i.e., for the extension from linear regression to random feature regression.

The lack of an explanation also makes it more difficult to trust the results. This is particularly true because the proofs are very calculation-heavy; e.g. section F consists of 20 pages of pure calculations. A good explanation of the intuition or of the proof techniques would help alleviate some of these worries.

The results in propositions 3-6 do seem actually interesting, assuming that the partial ordering is actually interesting.

In my opinion the experiments included in the main text are not particularly interesting (and similar results have been known about overparameterization for some time). I would have been more interested to see experiments that illuminate aspects of the theory, even if only synthetic (e.g., a check of proposition 3 for common covariance shifts, some of which may respect the partial ordering and some of which may not).

Other comments:

It is also unclear to me how standard Assumption 1 is. If it is standard, or if it applies to a wide range of common distributions, this could be stated. If not, then it should definitely be stated.

At line 78 there’s an unnecessary comma.

----

UPDATE: My concerns about the partial ordering are addressed. For some of the other concerns, the answer boils down to "this is what other papers in this area do too". Unfortunately, I am less familiar with this area than I initially thought. I increased my rating to 6 while decreasing my confidence to 1.

**Time Spent Reviewing:**

4

---

> ### Author Response · Authors · 2021-08-11
> **Response to Reviewer 9Mq1**
>
> We thank the reviewer for the time spent reviewing and the helpful feedback. In the comments below, we address the main points raised in the review. Specifically, we believe our response adequately addresses the reviewer's concerns about Definition 1 and Theorem 2. As such, we would kindly ask the reviewer to consider revising the overall score and recommendation.
>
> **Definition 1:**
>
> While at first glance the monotonicity of ratios of overlap coefficients in Definition 1 may seem restrictive, in fact many shifts satisfy the condition. To see this, for simplicity focus on the finite-dimensional case and consider two shifted covariance matrices $\Sigma_1^*$ and $\Sigma_2^*$. The overlap coefficients can equivalently be written as $r_{i,1}=v_i^\top \Sigma_1^* v_i$ and $r_{i,2}=v_i^\top \Sigma_2^* v_i$ where $v_i$ denote the eigenvectors of the training covariance. Without loss of generality, we can take the vectors $v_i$ to be the canonical basis vectors $e_i$, since we are always free to choose the coordinate system. In this case, the monotonicity condition only concerns the diagonal entries of $\Sigma_1^*$ and $\Sigma_2^*$. The remaining off-diagonal elements are completely unconstrained (so long as the matrices are PSD). Clearly these $\mathcal{O}(n_0^2)$ degrees of freedom allow for nontrivial eigenstructure and rich classes of covariate shifts that satisfy Definition 1.
>
> Additionally, the monotonicity condition in Definition 1 is as weak as possible, in the sense that without it there is no way to ensure a model-independent definition of shift strength. We have added to the SM a simple counterexample emphasizing this point, with a concrete realization of a pair of LJSDs which have a single out-of-order overlap coefficient, and show that this is sufficient to cause the relative ordering of errors in Proposition 3 to depend on the details of the random feature model, i.e. on its nonlinearity and width.
>
> As the reviewer points out, many real-world covariate shifts may violate these monotonicity conditions (though usually it is impossible to verify this since we do not have access to the population covariance matrices). However, we strongly disagree with the reviewer's perspective that this is a limitation of our analysis. To the contrary, our results provide a proof of the nontrivial and practically relevant observation that characterizing the ``strength" of such shifts cannot be done without reference to the model under consideration.
>
> - “Theorem 2 is complicated”: We agree that simplicity is important, and we have gone to great lengths to provide a simplified presentation of Theorem 2 in our revised draft. However, we would kindly remind the reviewer that we are presenting an exact expression for the test error of a nontrivial learning problem that exhibits intricate behavior and significant nuance. We do not believe it is reasonable to expect an overly simple formula to suffice. Indeed, although exact results of this type are relatively rare in the literature, the existing results exhibit similar levels of complexity. See for example Theorem 1/Definition 1 in https://arxiv.org/pdf/1908.05355.pdf, which expresses the test error for random feature regression (without covariate shift) as a complicated function of the solutions to a fixed point equation.
>
> - “Explanation of the intuition or of the proof techniques": The focus of this paper is on the consequences of the Theorem, as outlined in the various Propositions, not on its derivation or specific structure. The proof techniques have been developed elsewhere (see refs. [1,2,10,24, 32]) and were the primary focus of some of those prior works. The high-level structure of the Theorem is quite standard for asymptotic random matrix theory calculations.
>     Moreover, prior work in this vein has adopted a similar philosophy to ours, focusing on the implication of the Theorems rather than the proof techniques or the form of the result. For example, the structure of Theorems 1 and 4 of https://papers.nips.cc/paper/2019/file/c133fb1bb634af68c5088f3438848bfd-Paper.pdf is quite similar to ours, expressing the results in terms of $\psi$, a solution to a self-consistent equation, analogous to our $x$ in Eq. (11). In that work, which appeared at this same conference in a prior year, no effort is made to explain the intuition behind $\psi$ (or most of the quantities appearing in the Theorems) for the same reasons as us --- these types of expressions are standard in RMT, are derived in the proofs, and have little relevance for the main points in paper. For these reasons, we do not believe it is necessary or appropriate to include additional discussion of the structure of the Theorem or its proof in the main text. Having said that, for completeness, in Sections E and F of the SM we do actually include a lengthy discussion of the proof techniques as well as details of the mathematical machinery needed to move from linear regression to the nonlinear random feature setting.
>
> - “The lack of an explanation also makes it more difficult to trust the results.": We believe that that detailed proof in Section F, the thorough explanation of that proof in Sections E and F, the extensive numerical corroboration of our results in Figs. 1-3, and the agreement in special cases with prior work in Section C.5 together provide more than sufficient evidence to support the correctness of our results.
>
>  - “It is also unclear to me how standard Assumption 1 is":  It is a standard assumption. As is characteristic of asymptotic results in RMT, some specification of the high-dimensional limiting behavior of the relevant spectral objects is necessary. In fact, an assumption equivalent to Assumption 1 appears recently in https://arxiv.org/pdf/2006.05800.pdf in the study of overparameterized linear regression with weighted regularizers.

---

### Official Review · Reviewer_X5kM · 2021-07-16

**Rating:** 8
**Confidence:** 3

**Summary:**

The authors study covariate shift in over-parameterized random features models.  They provide an analysis of the test error, as well as an analysis of the decomposition of the test error into the bias and variance components.  This results in a partial ordering over test errors WRT the "hardness" of the covariate shift.  Further, the theoretical results provide a concrete explanation of the linear trend that has been observed in past work between the test error on clean and corrupted error.

**Limitations And Societal Impact:**

I don't think that this applies to this paper.

**Main Review:**


### Positive aspects

**Problem setting.**  The setting is very interesting here.  I haven't seen any theoretical work that has done much of anything with this setting.  As covariate shift is known to occur quite often in ML datasets, it's extremely important that we understand how covariate shift impacts test error.  So I think that any theoretical insights toward understanding the impact of covariate shift are quite valuable.

**Results in the linear setting.**  The result of Cor. 1 seems quite interesting.  In particular, it seems that the test error depends on the ratios of the "projected eigenvalues" $r_i$ and the $\lambda_i$.  This is cool, since it gives us a quantitive way (in the admittedly simple case of LR) to think about different magnitudes of covariate shift and their impact on test error.  And it seems to have the nice property that if there is no shift, we get the intuitive asymptotics $\sigma_\epsilon^2 \phi/(1-\phi)$.

**Corroboration of the linear trend between clean and corrupted test error.**  There has indeed been a growing recent discussion about the relationship between the clean and corrupted test error of deep models.  Along with [25], this is the first paper I have seen that has sought to theoretically explain the empirically-observed linear relationship between these quantities.  This is a significant contribution.

**A formal way of quantifying how "hard" a distribution shift is.**  I liked this notion of "hardness" of distribution shifts, which allows us to explicitly quantify the relationship between two distributions, where for example one distribution is the clean data and another is the shifted data.

**Experiments.**  The experiments were a nice touch.  I think that even without the CIFAR-10/CIFAR-10-C experiments, this paper would merit acceptance.  It was interesting (although perhaps unsurprising to see that as the corruption level increased, test accuracy (resp. error) went up (resp. down), and that over-parameterization seemed to improve the accuracy/error.  A small point of confusion was the notation involving $c$.  Is $c$ is the so-called "challenge-level" for these datasets?  As far as I can tell, $c$ is never defined.

---

### Notational confusion and lack of intuition

In my opinion, the notational section has a number of imprecise statements/definitions that make the rest of the paper harder to follow.  In what follows, I'll list the things that tripped me up:

**On the definition of $\sigma$.**  The activation function $\sigma$ seems to be defined differently in two separate locations, and I think that both of them are incorrect.  In line 115, we have $\sigma:\mathbb{R}^{n_0}\to\mathbb{R}$ and in line 150, we have $\sigma:\mathbb{R}\to\mathbb{R}$.  However, neither of these really make sense.  In the line 115 definition, it is said that $\sigma$ is applied elementwise.  How can a function mapping to $\mathbb{R}$ be applied elementwise?  Furthermore, if this is the definition, then we don't need a transpose in the definition of the kernel (3), since $\sigma(W_1 x_1/\sqrt{n_0})$ will already be real-valued.  And earlier on, the features $F$ and $f$ would reduce to a single scalar.  But following the standard literature (e.g. [35]), the feature map should be $\sigma: x\mapsto \sigma(Wx)$, which $\sigma(W\cdot)$ is an embedding of $x$ into some higher-dimensional feature space.  On the other hand, following the line 150 definition, we place a norm on $\sigma(x)$, i.e. we write $||\sigma(x)||$.  If $\sigma$ is real-valued, why would we need a norm?  The absolute value should suffice.  Similarly, if $\sigma:\mathbb{R}\to\mathbb{R}$, the gradient will be a scalar, so we also don't need a norm on the derivative $||\sigma'(x)||$.

From what I can tell, $\sigma$ should be such that $\sigma:\mathbb{R}^{n_1}\to\mathbb{R}^{n_1}$.  In this way, it's argument is an element of $\mathbb{R}^{n_1}$, which couples with the fact that $W_1x_1\in\mathbb{R}^{n_1}$.  Further, this $\sigma$ maps to $\mathbb{R}^{n_1}$, so its operation (e.g. ReLU, tanh, sigmoid, etc.) can be applied "elementwise" in the expected way.  However, if this is incorrect and this is all completely incorrect (which is possible), we can definitely clear things up in the discussion.

**On expectation notation.**  I find it extremely confusing when I cannot tell what the randomness in an expectation is over.  This is the case in the definitions in (6).  First of all, the authors at first say that the randomness in the innermost expectation in (5) is over $W$, $X$, and $\epsilon$.  But the randomness in $\epsilon$ should not matter here, because the authors pulled out the label noise, which is evinced by the trailing $\sigma_{\epsilon}^2$ term.  Then in (6), I'm unsure of what the randomness in $\mathbb{E}[\hat{y}(\mathbf{x})]$ and $\mathbb{V}[\hat{y}(\mathbf{x})]$ is over.  I thought it might be $\beta$, but the authors say that we supress the randomness in $\beta$ due to the assumption that $\beta$ is sub-Gaussian and therefore concentrates super fast around its mean.  So it's not $\beta$.  And the middle expectation is apparently over $W_1$, $X$, and $\epsilon$ (the latter of which is now apparently included despite the fact that it was explicitly factored out in $E_{\Sigma^\star}$).  If this is the case, then (as far as I can tell) there is no randomness left!  To summarize, it would help a lot if in these notational sections the authors could explicitly write which expectations correspond to which sources of randomness.

**On the trace notation.**  The authors say that $\bar{\text{tr}}$ denotes the "normalized" trace.  For an $n\times n$ matrix, does this mean we normalize by $n$ or by $\sqrt{n}$?

**Intuition for the overlap coefficients.**  I had a bit of trouble interpreting the overlap coefficients $r_i$.  To me, it seems that for each $i \in\{1, \dots, n_0\}$, we are projecting the eigenvector $v_i$ of $\Sigma$ onto the eigenspace spanned by the eigenvectors of $\Sigma^\star$.  So $r_i$ is kind of like the an "eigenvalue" for $v_i$ in this projected space induced by $\Sigma^\star$.  This doesn't agree with the statement that we should consider the spectra of $\Sigma$ and $\Sigma^\star$ in joint eigenbasis of $\Sigma$ (rather than $\Sigma^\star$), as stated on line 134.  Is this the right way to think about it, or have I misunderstood?  This seems like a fundamental quantity in this paper, so I think it would be helpful to build some intuition for the reader.

As a side note, it's a bit confusing to use $\nu$ and $v$ here.  With the wide world of characters open to you, perhaps there is a better choice of symbols that do not look so similar?

**Figure 1.**  It's hard to disentangle the relationship between the plots in Figure 1.  Two things are varying between (a) and (b): the activation and the composition of the covariance matrix.  It would be easier to understand if tanh was used for both, or there were four plots.  In essence, I can't disentangle the impact of the activation and the data mechanism based on this figure.  (Although putting that aside, it is quite interesting that we see this kind of "double descent" phenomena with over-parameterization!)

**The behemoth that is Theorem 2.** I was with the authors until around theorem 2, which hit me like a truck.  I think I could spend a full day trying to unpack and intuitively understand this result and the associated quantities in full generality, but reviewing 7 papers in 3 weeks makes such a commitment impossible.  To this end, I think that a gaping hole in the paper is an intuitive explanation of theorem 2.  What is the main takeaway?  While Corollary 1 illustrates this for the ridgeless linear regression (LR) setting, I am still left feeling that I don't understand what most of the associated quantities represent.  For instance, the authors tell us that

"The structural form of Theorem 2 implies the generalization error under covariate shift exclusively factors through $\mathcal{I}^\star$."

But what **is** $\mathcal{I}^\star$.  How can I interpret the definition

$$\mathcal{I}^\star_{a,b} \triangleq \phi\mathbb{E}_\mu \left[ r\lambda^a(\phi + x\lambda)^{-b}\right]$$

in an intuitive way?  I've stared at this for some time and still have very little idea.  Overall, I think that this lack of intuition is what makes this paper just a "good" paper instead of a "great" paper.

---

### Final Thoughts

I think that this a good paper that should be accepted.  The results addressing a crucial setting.  And the corollaries of theorem 2 tell us about (1) an ordering on test errors as a function of how "hard" the covariate shift is, and (2) an explanation of why over-parameterized models display a linear trend in test error under distribution shift.  The experiments on CIFAR-10 are also quite informative.  I think that as written, this paper is a "good" paper that could be made a "great" paper if more intuition was added throughout so that a broader class of readers could understand the results.  Also, I think some notational bugs should be sorted out.  But to summarize, I liked the paper a lot, and enjoyed reading it!


**Time Spent Reviewing:**

8

---

> ### Author Response · Authors · 2021-08-11
> **Response to Reviewer X5kM**
>
> We are grateful to the reviewer for the time and effort spent reading our paper and for the enthusiasm about our results. The many comments and suggestions have helped us improve the paper.
>
> - “Definition of $\sigma$”: We thank RX5kM for pointing out and correcting this typo. The correct definition should be $\sigma(x) : \mathbb{R} \to \mathbb{R}$ and the norm should be written as $|\sigma(x)|$; so $\sigma(W x)$ is a shorthand implying that the function is applied elementwise to the vector $W x$.
> - “Expectation notation”: In Eqns. (5) and (6), the outer expectation is over the *test* point $(\mathbf{x},y)$ and the inner expectation is over all randomness from the training process $W_1$, $X$, and $Y$ (leaving aside $\beta$). One point of confusion is that there are two sources of randomness in the label noise: label noise on the training data $(X,Y)$ and label noise on the *new* test data point $(\mathbf{x},y)$. The expectations over $\mathbb{E}$ (and $\mathbb{V}$) in Eqns. (5) and (6) are over all the randomness in the training process, $W_1, X, \mathbf{\epsilon}$, where $\mathbf{\epsilon}$ corresponds to label noise for the training data. This randomness is implicitly contained in the kernel prediction rule $\hat{y}(x)$ in Eqn. (5). The $-\sigma_{\epsilon}^2$ in Eqn. (5) subtracts the contribution of an i.i.d. additive noise contribution to the*new* test datapoint $\mathbf{x}$ which arises from $y(\mathbf{x})$. This test point noise is a constant, irreducible error term for any regression method. We have also clarified these points in an updated version of the paper.
> - “Normalized trace”: We have clarified that the normalized trace of a matrix $A \in \mathbb{R}^{n \times n}$ is $\frac{1}{n} \mathrm{tr}(A)$ in a new version of the paper.
> - “Interpretation of $r_i$/Overlap Coefficients”: The overlap coefficients can equivalently be written as $r_i=v_i^\top \Sigma^* v_i$ where $v_i$ denote the eigenvectors of the training covariance. Thus, $r_i$ is the induced norm of $v_i$ with respect to $\Sigma^*$. As such, $r_i$ capture the alignment of the training covariance eigendirections with the test covariance and help quantify the contributions to the total error, bias, and variance arising from different directions in the space.
> - Line 134: We were trying to communicate it is not sufficient to only specify the limiting behavior of the eigenvalues of $\Sigma^*$ and $\Sigma$ separately in Assumption 1: since the eigenspaces of $\Sigma^*$ and $\Sigma$ may not align, but their interaction is a relevant component in the test error. Hence, we must specify both the limiting behavior of the eigenvalues and overlap coefficients (i.e. in the same basis).
> - Figure 1: We thank the reviewer for the suggestion and have used the same activation function for both displays in an improved version of Fig. 1.
> - “Complexity of Theorem 2”:  We recognize the complexity of Theorem 2 and have further streamlined its presentation. However, we note the lack of an explicit closed-form solution for the test error is present even in the case of *ridge regression without any covariate shift*--see for example Theorem 2.1.2 in https://arxiv.org/abs/1507.03003. The main value of Theorem 2 is its ability to make precise predictions of the test error, bias, and variance that are not loose upper bounds. We hope the later propositions help elucidate the phenomenology that can be explored using Theorem 2. The term $\mathcal{I}^\star$ is the high dimensional limit of functionals arising in the random feature test error similar to how $\mathrm{tr}(\Sigma^* \Sigma) \to \mathbb{E}_{\mu}[r/\lambda]$ arises in the test error of linear regression.
> - Corruption Strength: We have now clarified in the experimental section that increasing values of $c$ corresponds to increasing corruption strengths on CIFAR-10-C.

---

> > ### Comment · Reviewer_X5kM · 2021-08-17
> > **Thanks for the clarifications + I'll stay at my original score**
> >
> > Thanks for clearing all of these things up.  I still feel that this is a strong paper, and I will keep my original score.

---

### Official Review · Reviewer_i3Dq · 2021-07-17

**Rating:** 6
**Confidence:** 3

**Summary:**

This paper studies random feature regression in high dimensional setting with the presence of covariate shift between train and test data. The authors compute the high-dimensional asymptotic limit of the test error for Gaussian covariates with a shift in the covariance matrix. They also provide results characterizing the bias and variance decomposition.

**Limitations And Societal Impact:**

See comments above for suggestions.

**Main Review:**

The authors did hard work to derive analytically the asymptotic limit of the test error for random feature regression with covariate shift, and the experimental results seem to agree with the theory well. However, I found the paper hard to follow and the writing of the paper needs to be improved. Even after several attempts, I still cannot parse the theoretical results in Section 2 and gain useful intuitions from them. This prevents me from evaluating the usefulness and significance of the work. I have no choice but vote rejection.

Below are several comments/questions I have that may help improve the readability of the paper:
1. The term "naturally occurring shift" is not explained in the paper. Also, I would like to know why the shift considered in the paper only happens in the covariance matrix. Is the shift in mean not natural?
1. Do you have intuition that could help understand $r_i$, i.e., overlap coefficients, defined on line 131?
1. Theorem 2 and Corollary 1 are too complicated. It is nice to have the exact (although implicit) form of the test error, but perhaps a more important question is: what intuition/knowledge can we gain from them besides the complicated expressions? It would be largely appreciated if a paragraph parsing these results was provided. Also, the symbol under the limit in (12) is not defined.
1. On line 225, the authors claim that "Proposition 3 corroborate the intuitions obtained from linear regression". The intuition obtained from linear regression is in Definition 1 and the paragraph above it. It is unclear to me how the condition $\xi \le 1$ is connected to that.
1. In Proposition 4, It is said that "When $\mu$ is hard, SLOPE $\ge 1$". What do you mean by $\mu$ is hard?

Other comments:
1. I would like to see references supporting the claim on line 89.
1. On line 115: should $R^{n_0}$ be $R$?
1. In Assumption 2, should $\lVert \sigma(x) \rVert$ be $|\sigma(x)|$?


----------------------------------------------------------------------------

Updates: after reading the authors' response and other reviewers' comments, most of my questions are answered. The partial ordering  characterizing the hardness of distribution shifts is indeed an interesting contribution. However, I do think Th. 2 and Cor. 1 are too complicated and lack explanations. Based on these considerations, I have increased my score to 6.

**Time Spent Reviewing:**

7

---

> ### Author Response · Authors · 2021-08-11
> **Response to Reviewer i3Dq**
>
> We thank the reviewer for reading our paper and giving us the opportunity to clarify some points that were raised. We hope this will help the reviewer reconsider the rating of our paper.
>
> - “Naturally occurring shift”: The term naturally occurring shift has been used in the empirical literature informally to describe shifts that are non-adversarial (and was our intended meaning). Examples include shifts arising from variation in data-collection procedures across different rounds or blur/fog corrupting an image. By contrast, poisoning attacks on a model--such as adversarial examples--are not naturally occurring. This point has been clarified in an update to the paper.
> - “No mean shift”: We chose not to explicitly include a mean shift since the centering condition is often explicitly enforced as a pre-processing step in most practical settings. Moreover, a mean shift adds complexity to the model without resulting in interesting new behavior.
> - “Interpretation of $r_i$/Overlap Coefficients": The overlap coefficients can equivalently be written as $r_i=v_i^\top \Sigma^* v_i$ where $v_i$ denote the eigenvectors of the training covariance. Thus, $r_i$ is the induced norm of $v_i$ with respect to $\Sigma^*$. As such, $r_i$ capture the alignment of the training covariance eigendirections with the test covariance and help quantify the contributions to the total error, bias, and variance arising from different directions in the space.
> - “Proposition 3 corroborate intuitions from LR”: The condition $\xi \leq 1$ does not have a direct analog in the simple setting of linear regression, and is a unique feature of the nonlinearity induced by random feature regression. However, the condition is relatively mild as $\xi=1$ when equalizing the train/test covariate distributions to have the same scale ($s=s_*$). We believe it is interesting (and surprising) that under a mild condition, a single *model-independent* definition shift can provide an ordering across the class of random feature models.
> - “$\mu$ is hard": Here we intend hard to be defined in the formal sense of Definition 1. That is, $\mu$ is hard, if $\mu$ is harder then $\mu_{\emptyset}$ (with hardness defined in Definition 1).
> - Line 115: We thank Ri3Dq for pointing out this typo, it should be $\sigma(x) : \mathbb{R} \to \mathbb{R}$ and the norm should be written as $|\sigma(x)|$.
> - Line 89: We have added a relevant citation for this claim to https://arxiv.org/abs/2012.07421.

---

### Decision · Program_Chairs · 2021-09-27

**Decision:**

Accept (Poster)

**Comment:**

The paper studies the robustness of overparametrized models to covariate shift between train and test data. The authors study this in the sandbox of random feature regression, by computing the high-dimensional asymptotic limit of the test error in the presence of a shift in the covariance matrix. In this model, they provide analytic solutions, and they corroborate the theory with empirical evidence.

The reviewers concluded the paper makes a value contribution and should be accepted, but noted that the paper could use improvements to the presentation. In particular, Theorem 2 is rather hard to parse/interpret, and some of the proofs are essentially long calculations that are difficult to follow. This was partially addressed in the discussions, and some of these techniques are standard in the literature. We strongly suggest that the authors address these concerns for the camera ready version.